# LEARNING HAMILTONIAN DYNAMICS AT SCALE: A DIFFERENTIAL-GEOMETRIC APPROACH

## ABSTRACT

Embedding physical intuition into network architectures allows the learning of dynamics that enforce fundamental properties, such as energy conservation laws, thereby leading to physically-plausible predictions. Yet, scaling these models to intrinsically high-dimensional dynamical systems remains a significant challenge. This paper introduces Geometric Reduced-order Hamiltonian Neural Network (RO-HNN), a novel physics-inspired neural network that combines the conservation laws of Hamiltonian mechanics with the scalability of model order reduction. RO-HNN is built on two core components: a novel geometrically-constrained symplectic autoencoder that learns a low-dimensional, structure-preserving symplectic submanifold, and a geometric Hamiltonian neural network that models the dynamics on the submanifold. Our experiments demonstrate that RO-HNN provides physically-consistent, stable, and generalizable predictions of complex high-dimensional dynamics, thereby effectively extending the scope of Hamiltonian neural networks to high-dimensional physical systems.

## 1 INTRODUCTION

Learning the unknown governing equations of dynamical systems is of fundamental importance to model physical processes. In this context, generic neural models lack built-in physical intuition, thus resulting in limited explainability and poor generalization beyond the data support. Embedding fundamental physical properties, such as conservation laws and boundary conditions, into neural networks has been shown to drastically improve their performance. Various models incorporate physical intuition as soft constraints via penalty terms in the loss function (Saqlain et al., 2023). This often leads to suboptimal enforcement of physical properties and to stiff optimization (Wang et al., 2021), motivating the embedding of domain-specific priors as hard constraints in specialized neural architectures. This allowed recent methods to learn dynamics that preserve energy (Greydanus et al., 2019; Cranmer et al., 2020; Lutter & Peters, 2023), conserve mass and momentum (Jnini et al., 2025), and strictly enforce general conservation laws (Liu et al., 2024b), thereby improving performances, generalization, and stability while yielding physically-consistent predictions.

Hamiltonian mechanics, introduced by Hamilton (1834) as a reformulation of Lagrangian mechanics, describe the evolution of a broad range of dynamical systems in robotics (Duong & Atanasov, 2021), fluid dynamics (Salmon, 1988), quantum mechanics (Schrödinger, 1926), and biology (Duarte et al., 1998), among others. Hamiltonian systems evolve on a phase space with symplectic structure, naturally enforcing energy conservation (Abraham & Marsden, 1987). Compared to Lagrangian mechanics, Hamiltonian mechanics provide a first-order formulation of dynamics that describes a broader range of physical systems. Hamiltonian neural networks (HNNs) are gray-box models that embed the Hamiltonian structure as hard constraints in specialized deep learning architectures. HNNs either directly learn the Hamiltonian function, ensuring conservation laws by construction (Greydanus et al., 2019; Lutter & Peters, 2023), or learn symplectomorphisms that preserve the invariants of interest via symplectic flows (Jin et al., 2020). HNNs were enhanced by including dissipation (Zhong et al., 2020a) and contact (Zhong et al., 2021) models, and utilized for model-based control (Duong & Atanasov, 2021; Zhong et al., 2020b). While most HNNs consider Hamiltonians characterized by a canonical symplectic form — exhibited at least locally for all Hamiltonian systems — few works proposed architectures handling non-canonical forms (Chen et al., 2021) and more general Poisson systems (Jin et al., 2021; Šípka et al., 2023). Although HNNs yielded drastic performance improvements over generic black-box models, their application remains limited to low-dimensional systems with 2-5 dimensions.

Learning the dynamics of high-dimensional physical systems, such as robots, continua, or fluids, is arguably a difficult problem due to the increasing complexity and nonlinearity of their governing equations. Several approaches combine data-driven sparse identification of nonlinear dynamics (SINDy) and dimensionality reduction to discover high-dimensional governing equations (Brunton et al., 2016; Champion et al., 2019). However, they disregard the *apriori*-known structures of physical systems. In contrast, Sharma & Kramer (2024); Friedl et al. (2025) took inspiration from model order reduction (MOR) to learn high-dimensional Lagrangian dynamics. MOR addresses the complexity of nonlinear high-dimensional governing equations, so-called full-order model (FOM), by finding a reduced-order model (ROM), i.e., a computationally-cheaper yet accurate low-dimensional surrogate model (Schilders et al., 2008). While MOR techniques are typically intrusive, i.e., they assume entirely-known FOM dynamics, Sharma et al. (2024) presented a novel non-intrusive MOR-based approach that learns the parameters of a high-dimensional Lagrangian system in a linear structure-preserving subspace. In a similar line, Friedl et al. (2025) adopted a Riemannian perspective on the problem and introduced a physics-inspired neural architecture that jointly learns a non-linear embedded submanifold via a biorthogonal Autoencoder (AE) and its associated low-dimensional conservative dynamics via a geometric Lagrangian neural network (LNN). A different line of works leverage Koopman operator theory to model nonlinear, e.g., Hamiltonian, dynamics via a learned surrogate linear dynamic model embedded in a higher-dimensional latent space (Lusch et al., 2018; Zhang et al., 2024).

**This paper** proposes a novel physics-inspired geometric deep neural network to learn the dynamics of high-dimensional Hamiltonian systems. In contrast to previous works that learn dynamics from high-dimensional observations such as images (Greydanus et al., 2019; Chen et al., 2021; Botev et al., 2021), we consider systems with *intrinsically high-dimensional* phase spaces. Taking inspiration from (Sharma & Kramer, 2024; Friedl et al., 2025), we build on recent advances in Hamiltonian MOR (Peng & Mohseni, 2016; Buchfink et al., 2024) and adopt a differential geometric perspective to embed the high-dimensional Hamiltonian structure as hard constraints in our architecture. **Our first contribution** is a geometrically-constrained symplectic AE that learns a low-dimensional symplectic submanifold from trajectories of a high-dimensional Hamiltonian system. Unlike soft-constrained symplectic networks (Buchfink et al., 2023), our AE guarantees the preservation of the symplectic structure of the FOM, including its conservation laws and stability properties (Lepri et al., 2024), with increased expressivity compared to linear and quadratic symplectic projections (Bendokat & Zimmermann, 2022; Sharma et al., 2023). **Our second contribution** is a geometric HNN that models conservative and dissipative Hamiltonian dynamics while accounting for the Riemannian geometry of its parameters, and resorts to symplectic integration (Tao, 2016) for accurate long-term dynamics simulation. **Our third contribution** is a geometric reduced-order Hamiltonian neural network (RO-HNN) that jointly learns a low-dimensional symplectic submanifold with a geometrically-constrained symplectic AE and the dynamics parameters of the associated Hamiltonian function with a geometric HNN. We validate our approach on three high-dimensional Hamiltonian systems: a pendulum, a thin cloth, and a particle vortex. Our experiments demonstrate that, due to its embedded geometries, RO-HNN predicts accurate, stable, and physically-consistent trajectories, outperforming traditional HNNs and state-of-the-art reduction approaches.

## 2 BACKGROUND

We provide a short background on Hamiltonian dynamics, structure-preserving Hamiltonian MOR, and related neural networks. Preliminaries on Riemannian and symplectic geometry are in App. A.

### 2.1 HAMILTONIAN DYNAMICS ON SYMPLECTIC MANIFOLDS

A symplectic manifold $(\mathcal{M}, \omega)$ is a $2n$-dimensional smooth manifold $\mathcal{M}$ equipped with a symplectic form $\omega$, i.e., a closed ($d\omega = 0$), non-degenerate, differential 2-form represented by a skew-symmetric matrix $\boldsymbol{\omega}$ in coordinates. We slightly abuse notation, equivalently denoting symplectic manifolds as $(\mathcal{M}, \boldsymbol{\omega})$. A Hamiltonian system $(\mathcal{M}, \boldsymbol{\omega}, \mathcal{H})$ is a dynamical system evolving on a symplectic manifold $(\mathcal{M}, \boldsymbol{\omega})$ according to a smooth Hamiltonian function $\mathcal{H} : \mathcal{M} \to \mathbb{R}$. The Hamiltonian vector field $\boldsymbol{X}_{\mathcal{H}} = \boldsymbol{\omega}^{-1}d\mathcal{H}$ is uniquely defined and preserves $\mathcal{H}$. Trajectories $\boldsymbol{\gamma} : \mathcal{I} \to \mathcal{M}$ of the system over a time-interval $\mathcal{I} = [t_0, t_{\mathrm{f}}]$ are solutions of the initial value problem (IVP)

$$\tfrac{d}{dt}\boldsymbol{\gamma}|_t = \boldsymbol{X}_{\mathcal{H}}|_{\boldsymbol{\gamma}(t)} \quad \in T_{\boldsymbol{\gamma}(t)}\mathcal{M}, \qquad \text{with} \qquad \boldsymbol{\gamma}(t_0) = \boldsymbol{\gamma}_0 \quad \in \mathcal{M}. \tag{1}$$

A diffeomorphism $f : (\mathcal{M}, \omega) \to (\mathcal{N}, \boldsymbol{\eta})$ between symplectic manifolds is a symplectomorphism if it preserves the symplectic form, i.e., $f^*\eta = \omega$ with $f^*\eta$ denoting the pullback of $\eta$ by $f$.

Following Darboux theorem, there exists a canonical chart $(U, \phi)$, $\boldsymbol{x} \in U$ for each point $\boldsymbol{x} \in \mathcal{M}$ in which the symplectic form is represented as $\boldsymbol{\omega} = \mathbb{J}_{2n}^{\intercal}$ via the canonical Poisson tensor

$$\mathbb{J}_{2n} = \begin{pmatrix} \mathbf{0} & \boldsymbol{I}_n \\ -\boldsymbol{I}_n & \mathbf{0} \end{pmatrix}, \quad \text{for which} \quad \mathbb{J}_{2n}^{\intercal} = \mathbb{J}_{2n}^{-1} = -\mathbb{J}_{2n}. \tag{2}$$

In other words, every symplectic manifold is locally symplectomorphic to $(\mathbb{R}^{2n}, \mathbb{J}_{2n}^{\intercal})$. A system $(\mathbb{R}^{2n}, \mathbb{J}_{2n}^{\intercal}, \mathcal{H})$ is called a canonical Hamiltonian system.

In this paper, we consider Hamiltonian systems $(\mathcal{M}, \boldsymbol{\omega}, \mathcal{H})$, on $\mathcal{M}$ globally valid canonical symplectic form $\boldsymbol{\omega} = \mathbb{J}_{2n}^{\intercal}$. In this case, the phase space $\mathcal{M}$ can be modeled on the cotangent bundle $\mathcal{T}^* \mathcal{Q}$ of a smooth $n$-dimensional manifold $\mathcal{Q}$ (Weinstein, 1971) with canonical coordinates $(\boldsymbol{q}, \boldsymbol{p})$ with position $\boldsymbol{q} \in \mathcal{Q}$ and conjugate momenta $\boldsymbol{p} \in \mathcal{T}_{\boldsymbol{q}}^* \mathcal{Q}$. The Hamiltonian vector field simplifies to $(\dot{\boldsymbol{q}}^{\intercal}, \dot{\boldsymbol{p}}^{\intercal})^{\intercal} = \boldsymbol{X}_{\mathcal{H}} = \mathbb{J}_{2n} d\mathcal{H}^{\intercal} = \left( \frac{\partial \mathcal{H}}{\partial \boldsymbol{p}}, -\frac{\partial \mathcal{H}}{\partial \boldsymbol{q}} \right)^{\intercal}$. Moreover, the Hamiltonian system $(\mathcal{T}^* \mathcal{Q}, \mathbb{J}_{2n}^{\intercal}, \mathcal{H})$ relates to a Lagrangian function $\mathcal{L} : \mathcal{T}\mathcal{Q} \to \mathbb{R}$ via the Legendre transform, which takes $\mathcal{L}$ to $\mathcal{H} = \dot{\boldsymbol{q}}^{\intercal} \boldsymbol{p} - \mathcal{L}$ with $\boldsymbol{p} = \frac{\partial \mathcal{L}}{\partial \dot{\boldsymbol{q}}}$ and $\dot{\boldsymbol{q}} \in \mathcal{T}_{\boldsymbol{q}} \mathcal{Q}$. Mechanical systems often display a quadratic kinetic energy structure, where the configuration manifold $\mathcal{Q}$ is a Riemannian manifold endowed with the kinetic-energy metric equal to the system's mass–inertia matrix $\boldsymbol{M}(\boldsymbol{q})$. In this case, the Hamiltonian function is given by the sum of the system's kinetic $T(\boldsymbol{q}, \boldsymbol{p})$ and potential $V(\boldsymbol{q})$ energies as $\mathcal{H} = T(\boldsymbol{q}, \boldsymbol{p}) + V(\boldsymbol{q}) = \frac{1}{2} \boldsymbol{p}^{\intercal} \boldsymbol{M}^{-1}(\boldsymbol{q}) \boldsymbol{p} + V(\boldsymbol{q})$ and the momenta is $\boldsymbol{p} = \boldsymbol{M}(\boldsymbol{q}) \dot{\boldsymbol{q}}$.

## 2.2 Structure-Preserving Model-Order Reduction of Hamiltonian Systems

Given the known parametrized dynamic equations of a high-dimensional system, i.e., a FOM, MOR aims to construct a low-dimensional surrogate dynamic model, i.e., a ROM, that accurately and efficiently approximates the FOM trajectories. Structure-preserving MOR preserves the underlying geometric structure of the FOM, ensuring that its properties, e.g. stability and energy conservation, are maintained in the ROM. For Hamiltonian systems $(\mathcal{M}, \boldsymbol{\omega}, \mathcal{H})$, the symplectic structure is preserved by constructing a reduced Hamiltonian $(\check{\mathcal{M}}, \check{\boldsymbol{\omega}}, \check{\mathcal{H}})$ with $\dim(\check{\mathcal{M}}) = d \ll \dim(\mathcal{M}) = n$, whose vector field $\check{\boldsymbol{X}}_{\mathcal{H}}$ approximates the set of solutions $S = \{\boldsymbol{\gamma}(t) \in \mathcal{M} \mid t \in \mathcal{I}\} \subseteq \mathcal{M}$ of (1).

Following the geometric perspective of Buchfink et al. (2024), the reduced Hamiltonian $(\check{\mathcal{M}}, \check{\boldsymbol{\omega}}, \check{\mathcal{H}})$ is derived by identifying the submanifold $\check{\mathcal{M}}$ via a smooth embedding $\varphi : \check{\mathcal{M}} \to \mathcal{M}$ such that

$$\check{\boldsymbol{\omega}} = \varphi^* \boldsymbol{\omega} = d\varphi^{\intercal} \boldsymbol{\omega} d\varphi, \tag{3}$$

is non-degenerate. This implies that $(\check{\mathcal{M}}, \check{\boldsymbol{\omega}})$ is a symplectic manifold and $\varphi$ is a symplectomorphism (Buchfink et al., 2024, Lemma 5.13). Note that structure-preserving Hamiltonian MOR typically considers a canonical FOM $(\mathbb{R}^{2n}, \mathbb{J}_{2n}^{\intercal}, \mathcal{H})$ reduced to a canonical ROM $(\mathbb{R}^{2d}, \mathbb{J}_{2d}^{\intercal}, \check{\mathcal{H}})$ (Peng & Mohseni, 2016; Sharma et al., 2023; Buchfink et al., 2023). The Hamiltonian structure is preserved by constructing $\check{\mathcal{H}}$ via the pullback of the embedding as $\check{\mathcal{H}} = \varphi^* \mathcal{H} = \mathcal{H} \circ \varphi$. Trajectories $\tilde{\boldsymbol{\gamma}}(t)$ of the reduced-order system are then obtained from the ROM $\frac{d}{dt} \tilde{\boldsymbol{\gamma}} \big|_t = \check{\boldsymbol{X}}_{\check{\mathcal{H}}} \big|_{\tilde{\boldsymbol{\gamma}}(t)} \in \mathcal{T}_{\tilde{\boldsymbol{\gamma}}(t)} \check{\mathcal{M}}$, with $\check{\boldsymbol{X}}_{\check{\mathcal{H}}} = \check{\boldsymbol{\omega}}^{-1} d\check{\mathcal{H}}$. The reduced initial value $\tilde{\boldsymbol{\gamma}}_0 = \rho(\boldsymbol{\gamma}_0) \in \check{\mathcal{M}}$ is computed via the point reduction map $\rho : \mathcal{M} \to \check{\mathcal{M}}$ associated with $\varphi$, which must satisfy the projection properties

$$\rho \circ \varphi = \mathrm{id}_{\check{\mathcal{M}}} \quad \text{and} \quad d\rho|_{\varphi(\check{\boldsymbol{x}})} \circ d\varphi|_{\check{\boldsymbol{x}}} = \mathrm{id}_{\mathcal{T}_{\check{\boldsymbol{x}}} \check{\mathcal{M}}}, \quad \forall \check{\boldsymbol{x}} \in \check{\mathcal{M}}. \tag{4}$$

Trajectories of the original system are finally obtained as the approximation $\boldsymbol{\gamma}(t) \approx \varphi(\tilde{\boldsymbol{\gamma}}(t))$.

The embedding $\varphi$ and point reduction $\rho$ are key for MOR as they determine the ROM trajectories. Accurately approximating the FOM requires the minimization of the reconstruction error

$$\ell_{\mathrm{rec}} = \frac{1}{N} \sum_{i=1}^{N} \| \varphi \circ \rho(\boldsymbol{x}_i) - \boldsymbol{x}_i \|^2. \tag{5}$$

Exact reconstruction requires $d\rho$ to be the symplectic inverse of $d\varphi$, i.e., $d\rho = d\varphi^+ = \check{\boldsymbol{\omega}}^{-1} d\varphi^{\intercal} \boldsymbol{\omega}$. In this paper, we introduce a geometrically-constrained AE that fulfills (3) and (4) by design.

## 2.3 Hamiltonian Neural Networks

While MOR reduces the dimensionality of systems with known dynamics, HNNs aim to learn the unknown dynamics of typically low-dimensional systems while ensuring energy conservation. Most HNNs assume canonical Hamiltonian systems or Hamiltonian systems with canonical symplectic

Figure 1: Flowchart of the forward dynamics of the geometric RO-HNN. The geometrically-constrained symplectic AE (in blue) is built as the cotangent lift of a constrained AE (top right). The geometric HNN (in brown) is composed of two SPD networks for the inverse mass-inertia and damping matrices and one MLP for the potential energy.

form (2). In this paper, we build on two HNN variants that *(1)* learn the Hamiltonian function as a single network $\mathcal{H}_{\boldsymbol{\theta}}(\boldsymbol{q}, \boldsymbol{p})$ with parameters $\boldsymbol{\theta}$ (Greydanus et al., 2019), or *(2)* learn the kinetic and potential energy as two distinct networks, i.e., $\mathcal{H}_{\boldsymbol{\theta}}(\boldsymbol{q}, \boldsymbol{p}) = T_{\boldsymbol{\theta}_\mathrm{T}}(\boldsymbol{q}, \boldsymbol{p}) + V_{\boldsymbol{\theta}_\mathrm{V}}(\boldsymbol{q})$ (Zhong et al., 2020b; Lutter & Peters, 2023). Given a set of $N$ observations $\{\boldsymbol{q}_i, \boldsymbol{p}_i, \dot{\boldsymbol{q}}_i, \dot{\boldsymbol{p}}_i\}_{i=1}^N$, the networks are trained to minimize the prediction error of the Hamiltonian vector field via the loss

$$\ell_{\mathrm{HNN}} = \|\frac{\partial \mathcal{H}_{\boldsymbol{\theta}}}{\partial \boldsymbol{p}}(\boldsymbol{q}_i, \boldsymbol{p}_i) - \dot{\boldsymbol{q}}_i\|^2 + \|\frac{\partial \mathcal{H}_{\boldsymbol{\theta}}}{\partial \boldsymbol{q}}(\boldsymbol{q}_i, \boldsymbol{p}_i) - \dot{\boldsymbol{p}}_i\|^2. \tag{6}$$

## 3 GEOMETRIC REDUCED-ORDER HAMILTONIAN NEURAL NETWORKS

We present the geometric reduced-order Hamiltonian neural network (RO-HNN) that learns the unknown dynamics of high-dimensional Hamiltonian systems. We focus on systems $(\mathcal{M}, \mathbb{J}_{2n}^\mathsf{T}, \mathcal{H})$ evolving on a phase space $\mathcal{M}$ with canonical symplectic form $\mathbb{J}_{2n}^\mathsf{T}$ for which the solutions $\boldsymbol{\gamma}(t)$ of the FOM (1) can be accurately approximated by a substantially lower dimensional surrogate model. Our goal is to learn a reduced Hamiltonian system $(\check{\mathcal{M}}, \check{\boldsymbol{\omega}}, \check{\mathcal{H}})$ via non-intrusive structure-preserving MOR, where we set $\check{\mathcal{M}}$ as a phase space with $\check{\boldsymbol{\omega}} = \mathbb{J}_{2d}^\mathsf{T}$. Given a set of high-dimensional observations $\{\boldsymbol{q}_i, \boldsymbol{p}_i\}_{i=1}^N$, we identify low-dimensional dynamics by jointly learning a reduced symplectic manifold $(\check{\mathcal{M}}, \mathbb{J}_{2d}^\mathsf{T})$ via a smooth embedding $\varphi$ and a reduction $\rho$, and a latent Hamiltonian function $\check{\mathcal{H}}$.

The proposed RO-HNN ensures the preservation of the Hamiltonian structure by fulfilling three necessary conditions by design: *(1)* the embedding $\varphi$ is a symplectomorphism, or equivalently

$$\check{\boldsymbol{\omega}} = \mathbb{J}_{2d} = d\varphi^\mathsf{T} \mathbb{J}_{2n} d\varphi; \tag{7}$$

*(2)* the embedding $\varphi$ and reduction map $\rho$ satisfy the projection properties (4); and *(3)* $\check{\mathcal{H}}$ is a valid Hamiltonian function, thus preserving the reduced energy $\check{\mathcal{E}} = \mathcal{E} \circ \varphi$. The RO-HNN fulfill *(1)*-*(2)* via a novel geometrically-constrained symplectic AE (Sec. 3.1), while *(3)* is guaranteed by a reduced-order geometric HNN (Sec. 3.2), whose trajectories are obtained via symplectic integration (Sec. 3.3). Accurate modeling of the high-dimensional dynamics is achieved by jointly training the AE and the HNN (Sec. 3.4). The proposed RO-HNN is illustrated in Fig. 1.

### 3.1 GEOMETRICALLY-CONSTRAINED SYMPLECTIC AUTOENCODER

Preserving the geometric structure of the original Hamiltonian FOM is crucial for the learned ROM to display similar dynamics. We introduce a geometrically-constrained symplectic AE that projects a high-dimensional Hamiltonian system $(\mathcal{M}, \mathbb{J}_{2n}^\mathsf{T}, \mathcal{H})$ onto a low-dimensional nonlinear symplectic manifold $(\check{\mathcal{M}}, \mathbb{J}_{2d}^\mathsf{T})$ such that the reduced system strictly retains the Hamiltonian structure of the FOM. We parametrize the point reduction $\rho : \mathcal{M} \to \check{\mathcal{M}}$ and embedding $\varphi : \check{\mathcal{M}} \to \mathcal{M}$ as the encoder and decoder of an AE designed to satisfy symplecticity (7) and projection properties (4) by construction. To do so, we leverage the cotangent bundle structure of the phase space $\mathcal{M} = \mathcal{T}^* \mathcal{Q}$.

Given a smooth embedding $\varphi_\mathcal{Q} : \check{\mathcal{Q}} \to \mathcal{Q}$ and associated point reduction $\rho_\mathcal{Q} : \mathcal{Q} \to \check{\mathcal{Q}}$ satisfying (4), we define the cotangent-lifted embedding $\varphi$ and point reduction $\rho$ in canonical coordinates as

$$\varphi(\check{\boldsymbol{q}}, \check{\boldsymbol{p}}) = \begin{pmatrix} \varphi_\mathcal{Q}(\check{\boldsymbol{q}}) \\ d\rho_\mathcal{Q}|_{\varphi_\mathcal{Q}(\check{\boldsymbol{q}})}^\mathsf{T} \check{\boldsymbol{p}} \end{pmatrix} \qquad \text{and} \qquad \rho(\boldsymbol{q}, \boldsymbol{p}) = \begin{pmatrix} \rho_\mathcal{Q}(\boldsymbol{q}) \\ d\varphi_\mathcal{Q}|_{\rho_\mathcal{Q}(\boldsymbol{q})}^\mathsf{T} \boldsymbol{p} \end{pmatrix}, \qquad (8)$$

where the pullbacks $d\rho_\mathcal{Q}|_{\varphi_\mathcal{Q}(\check{\boldsymbol{q}})}^\mathsf{T}\check{\boldsymbol{p}}$ and $d\varphi_\mathcal{Q}|_{\rho_\mathcal{Q}(\boldsymbol{q})}^\mathsf{T}\boldsymbol{p}$ are computed using the Jacobian of $\rho_\mathcal{Q}$ and $\varphi_\mathcal{Q}$.

**Proposition 1.** *The reduction map $\rho(\boldsymbol{q}, \boldsymbol{p})$ (8) satisfies the projection properties* (4).

*Proof.* It is clear that the cotangent-lifted map $\rho$ fulfills (4) as $\rho_\mathcal{Q}$ satisfies (4) by assumption. $\qquad\square$

**Proposition 2.** *The embedding $\varphi(\check{\boldsymbol{q}}, \check{\boldsymbol{p}})$ (8) satisfies the symplecticity property* (7).

*Proof.* Proving the statement is equivalent to show that the differential $d\varphi = \begin{pmatrix} d\varphi_\mathcal{Q} & \boldsymbol{0} \\ \frac{\partial(d\rho_\mathcal{Q}^\mathsf{T}\check{\boldsymbol{p}})}{\partial \check{\boldsymbol{q}}} & d\rho_\mathcal{Q}^\mathsf{T} \end{pmatrix}$ belongs to the symplectic Stiefel manifold $\mathrm{Sp}(2n, 2d) = \{\boldsymbol{U} \in \mathbb{R}^{2n \times 2d} \mid \boldsymbol{U}^\mathsf{T}\mathbb{J}_{2n}\boldsymbol{U} = \mathbb{J}_{2d}\}$. A block matrix $\boldsymbol{U} = \begin{pmatrix} A & B \\ C & D \end{pmatrix}$ belongs to $\mathrm{Sp}(2n, 2d)$ if its block elements satisfy the condition

$$\boldsymbol{U}^\mathsf{T}\mathbb{J}_{2n}^\mathsf{T}\boldsymbol{U} = \begin{pmatrix} A^\mathsf{T} & C^\mathsf{T} \\ B^\mathsf{T} & D^\mathsf{T} \end{pmatrix} \begin{pmatrix} \boldsymbol{0} & -\boldsymbol{I}_n \\ \boldsymbol{I}_n & \boldsymbol{0} \end{pmatrix} \begin{pmatrix} A & B \\ C & D \end{pmatrix} = \begin{pmatrix} C^\mathsf{T}A - A^\mathsf{T}C & C^\mathsf{T}B - A^\mathsf{T}D \\ D^\mathsf{T}A - B^\mathsf{T}C & D^\mathsf{T}B - B^\mathsf{T}D \end{pmatrix} = \begin{pmatrix} \boldsymbol{0} & -\boldsymbol{I}_d \\ \boldsymbol{I}_d & \boldsymbol{0} \end{pmatrix},$$

i.e., the differential $d\varphi$ must satisfy

$$d\varphi^\mathsf{T}\mathbb{J}^\mathsf{T}d\varphi = \begin{pmatrix} C^\mathsf{T}d\varphi_\mathcal{Q} - d\varphi_\mathcal{Q}^\mathsf{T}C & -d\varphi_\mathcal{Q}^\mathsf{T}d\rho_\mathcal{Q}^\mathsf{T} \\ d\rho_\mathcal{Q}d\varphi_\mathcal{Q} & \boldsymbol{0} \end{pmatrix} = \begin{pmatrix} \boldsymbol{0} & -\boldsymbol{I}_d \\ \boldsymbol{I}_d & \boldsymbol{0} \end{pmatrix} \quad \text{with} \quad C = \frac{\partial(d\rho_\mathcal{Q}|_{\varphi_\mathcal{Q}(\check{\boldsymbol{q}})}^\mathsf{T}\check{\boldsymbol{p}})}{\partial\check{\boldsymbol{q}}}. \tag{9}$$

By assumption, $\rho_\mathcal{Q}$ fulfills the projection properties (4), so that $d\rho_\mathcal{Q}d\varphi_\mathcal{Q} = d\varphi_\mathcal{Q}^\mathsf{T}d\rho_\mathcal{Q}^\mathsf{T} = \boldsymbol{I}_d$ holds by construction. It remains to prove $C^\mathsf{T}d\varphi_\mathcal{Q} - d\varphi_\mathcal{Q}^\mathsf{T}C = \boldsymbol{0}$. We denote the elements of the canonical and reduced canonical coordinates as $q^i$, $p_i$ and $\check{q}^\alpha$, $\check{p}_\alpha$, respectively. By definition, we have $(d\varphi_\mathcal{Q})_\alpha^i = \frac{\partial q^i}{\partial \check{q}^\alpha}$ and $(d\rho_\mathcal{Q})_i^\alpha = \frac{\partial \check{q}^\alpha}{\partial q^i}$ and the projection properties hold by assumption, i.e., $(d\rho_\mathcal{Q})_i^\alpha(d\varphi_\mathcal{Q})_\beta^i = \delta_\beta^\alpha \ \forall\check{q} \in \check{\mathcal{Q}}$, with $\delta_\beta^\alpha = 1$ if $\alpha = \beta$ and $\delta_\beta^\alpha = 0$ otherwise. Therefore, we have $p_i = (d\rho_\mathcal{Q})_i^\alpha\check{p}_\alpha$, and $C_{i\gamma} = \frac{\partial p_i}{\partial\check{q}^\gamma} = \frac{\partial}{\partial\check{q}^\gamma}((d\rho_\mathcal{Q})_i^\alpha\check{p}_\alpha) = \frac{\partial(d\rho_\mathcal{Q})_i^\alpha}{\partial q^j}\frac{\partial q^j}{\partial\check{q}^\gamma}\check{p}_\alpha = \frac{\partial(d\rho_\mathcal{Q})_i^\alpha}{\partial q^j}(d\varphi_\mathcal{Q})_\gamma^j\check{p}_\alpha$. We aim to show that $C^\mathsf{T}d\varphi_\mathcal{Q}$ is symmetric, i.e., $(d\varphi_\mathcal{Q})_\beta^i C_{i\gamma} = (d\varphi_\mathcal{Q})_\gamma^i C_{i\beta}$. Using the projection properties, we can write $\check{p}_\beta = (d\varphi_\mathcal{Q})_\beta^i(d\rho_\mathcal{Q})_i^\alpha\check{p}_\alpha$. Differentiating with respect to $\check{q}^\gamma$ yields

$$0 = \frac{\partial}{\partial\check{q}^\gamma}\left((d\varphi_\mathcal{Q})_\beta^i(d\rho_\mathcal{Q})_i^\alpha\check{p}_\alpha\right) = \frac{(d\varphi_\mathcal{Q})_\beta^i}{\partial\check{q}^\gamma}(d\rho_\mathcal{Q})_i^\alpha\check{p}_\alpha + (d\varphi_\mathcal{Q})_\beta^i\frac{(d\rho_\mathcal{Q})_i^\alpha}{\partial q^j}\frac{\partial q^j}{\partial\check{q}^\gamma}\check{p}_\alpha$$

$$= \frac{(d\varphi_\mathcal{Q})_\beta^i}{\partial\check{q}^\gamma}p_i + (d\varphi_\mathcal{Q})_\beta^i C_{i\gamma} = \frac{\partial q^i}{\partial\check{q}^\beta\partial\check{q}^\gamma}p_i + (d\varphi_\mathcal{Q})_\beta^i C_{i\gamma}.$$

As the Hessian in the first term is symmetric, the equality implies the symmetricity of the second term, i.e., $(d\varphi_\mathcal{Q})_\beta^i C_{i\gamma} = (d\varphi_\mathcal{Q})_\gamma^i C_{i\beta}$, and thus (9) holds. $\qquad\square$

Note that $d\rho\mathbb{J}_{2n}d\rho^\mathsf{T} = \mathbb{J}_{2d}$ is shown to hold on $\varphi(\check{\mathcal{M}})$ with similar arguments. Moreover, a similar proof is presented by Sharma et al. (2023) in the context of quadratic symplectic projections.

In practice, we learn the embedding $\varphi_\mathcal{Q}$ and point reduction $\rho_\mathcal{Q}$ via the constrained AE architecture from Otto et al. (2023), and compute their differentials analytically to construct the cotangent-lifted maps (8) (see App. D for details). The encoder and decoder are given as a composition of feedforward layers $\rho_\mathcal{Q} = \rho_\mathcal{Q}^{(1)} \circ \ldots \circ \rho_\mathcal{Q}^{(L)}$ and $\varphi_\mathcal{Q} = \varphi_\mathcal{Q}^{(L)} \circ \ldots \circ \varphi_\mathcal{Q}^{(1)}$ with $\rho_\mathcal{Q}^{(l)} : \mathbb{R}^{n_l} \to \mathbb{R}^{n_{l-1}}$, $\varphi_\mathcal{Q}^{(l)} : \mathbb{R}^{n_{l-1}} \to \mathbb{R}^{n_l}$, and $n_{l-1} \leq n_l$. The key to fulfill the projection properties (4) is the construction of the layer pairs as

$$\rho_\mathcal{Q}^{(l)}(\boldsymbol{q}^{(l)}) = \sigma_-\left(\boldsymbol{\Psi}_l^\mathsf{T}(\boldsymbol{q}^{(l)} - \boldsymbol{b}_l)\right) \qquad \text{and} \qquad \varphi_\mathcal{Q}^{(l)}(\check{\boldsymbol{q}}^{(l-1)}) = \boldsymbol{\Phi}_l\sigma_+(\check{\boldsymbol{q}}^{(l-1)}) + \boldsymbol{b}_l, \qquad (10)$$

where $(\boldsymbol{\Phi}_l, \boldsymbol{\Psi}_l)$ and $(\sigma_+, \sigma_-)$ are pairs of weight matrices and smooth activation functions such that $\boldsymbol{\Psi}_l^\mathsf{T}\boldsymbol{\Phi}_l = \boldsymbol{I}_{n_{l-1}}$ and $\sigma_- \circ \sigma_+ = \mathrm{id}$, respectively, and $\boldsymbol{b}_l$ are bias vectors. Therefore, each layer pair (10) satisfies $\rho_\mathcal{Q}^{(l)} \circ \varphi_\mathcal{Q}^{(l)} = \mathrm{id}_{\mathbb{R}^{n_{l-1}}}$ and the constrained AE fulfills (4). Following (Friedl et al., 2025), we ensure that the pairs of weight matrices adhere to the biorthogonality constraint $\boldsymbol{\Psi}_l^\mathsf{T}\boldsymbol{\Phi}_l = \boldsymbol{I}_d$ by accounting for the Riemannian geometry of biorthogonal matrices (see App. B.2 for a background). Specifically, we consider each pair $(\boldsymbol{\Phi}_l, \boldsymbol{\Psi}_l)$ as an element of the biorthogonal manifold $\mathcal{B}_{n_l, n_{l-1}} = \{(\boldsymbol{\Phi}, \boldsymbol{\Psi}) \in \mathbb{R}^{n_l \times n_{l-1}} \times \mathbb{R}^{n_l \times n_{l-1}} : \boldsymbol{\Psi}^\mathsf{T}\boldsymbol{\Phi} = \boldsymbol{I}_{n_{l-1}}\}$ and optimize them to minimize the reconstruction error (5) via Riemannian optimization (Absil et al., 2007; Boumal, 2023)

(see. App. F). Note that this Riemannian approach was shown to consistently outperform the over-parametrization proposed by Otto et al. (2023), achieving lower reconstruction errors (Friedl et al., 2025). The constraint $\sigma_- \circ \sigma_+ = \text{id}$ is met by utilizing the smooth, invertible activation functions defined in (Otto et al., 2023, Eq. 12), see also App. D.

As will be shown in Sec. 4, the resulting geometrically-constrained symplectic AE provides increased expressivity compared to linear and quadratic symplectic projection approaches (Peng & Mohseni, 2016; Sharma et al., 2023), while guaranteeing the symplectic structure of the latent space in contrast to weakly-symplectic AEs based on soft constraints (Buchfink et al., 2023). In the intrusive case, i.e., if the FOM is known, we construct the reduced Hamiltonian function via the pullback of the cotangent-lifted embedding as $\check{\mathcal{H}} = \varphi^* \mathcal{H}$, which yields the Hamiltonian ROM. Instead, in this paper, we consider the case where the high-dimensional dynamics are unknown, and learn the reduced-order Hamiltonian function $\check{\mathcal{H}}$ with a geometric HNN, as explained next.

### 3.2 CONSERVATIVE AND DISSIPATIVE HAMILTONIAN REDUCED-ORDER MODELS

**Conservative dynamics.** We propose to learn the reduced Hamiltonian dynamics in the embedded symplectic submanifold $(\check{\mathcal{M}}, \mathbb{J}_{2d}^{\mathsf{T}})$ via a HNN. For general systems, we encode the reduced-order Hamiltonian function as a single neural network $\check{\mathcal{H}}_{\boldsymbol{\theta}}(\check{\boldsymbol{q}}, \check{\boldsymbol{p}})$ with parameters $\theta$, akin to (Greydanus et al., 2019). However, additional prior knowledge on the structure of the Hamiltonian is often available. For instance, the Hamiltonian function of mechanical systems sums a quadratic kinetic energy and a potential term. Leveraging that the learned symplectic submanifold preserves the original system structure, we propose to model the reduced Hamiltonian function as $\check{\mathcal{H}}_{\boldsymbol{\theta}}(\check{\boldsymbol{q}}, \check{\boldsymbol{p}}) = \frac{1}{2}\check{\boldsymbol{p}}^{\mathsf{T}} \check{\boldsymbol{M}}_{\boldsymbol{\theta}_{\check{T}}}^{-1}(\check{\boldsymbol{q}})\check{\boldsymbol{p}} + \check{V}_{\boldsymbol{\theta}_{\check{V}}}(\check{\boldsymbol{q}})$ via two neural networks $\check{\boldsymbol{M}}_{\boldsymbol{\theta}_{\check{T}}}^{-1}$ and $\check{V}_{\boldsymbol{\theta}_{\check{V}}}$ with parameters $\boldsymbol{\theta} = \{\boldsymbol{\theta}_{\check{T}}, \boldsymbol{\theta}_{\check{V}}\}$. Existing HNNs (Lutter & Peters, 2023; Zhong et al., 2020a) enforce the symmetric positive-definiteness of the inverse mass-inertia matrix via a Euclidean network encoding its Cholesky decomposition $\boldsymbol{L}$, i.e., $\boldsymbol{M}^{-1} = \boldsymbol{L}\boldsymbol{L}^{\mathsf{T}}$. However, as for LNNs (Friedl et al., 2025), this parametrization leads to flawed measures of distances in the space of symmetric positive-definite (SPD) matrices and ultimately results in inaccurate dynamics predictions. To overcome this issue, we parametrize $\check{\boldsymbol{M}}_{\boldsymbol{\theta}_{\check{T}}}^{-1}$ via the SPD network from Friedl et al. (2025) that accounts for the Riemannian geometry of the SPD manifold $\mathcal{S}_{++}^d$ (see Apps. A, C). The network $\check{\boldsymbol{M}}_{\boldsymbol{\theta}_{\check{T}}}^{-1}(\boldsymbol{q}) = (g_{\text{Exp}} \circ g_{\mathbb{R}})(\boldsymbol{q})$ is composed of *(1)* a standard Euclidean multilayer perceptron (MLP) $g_{\mathbb{R}} : \mathbb{R}^d \to \mathbb{R}^{d(d+1)/2}$ that maps the input configuration to the elements of a symmetric matrix $\boldsymbol{U} \in \text{Sym}^d$, and *(2)* an exponential map layer $g_{\text{Exp}}$ that interprets $\boldsymbol{U}$ as an element of the tangent space $\mathcal{T}_{\boldsymbol{P}}\mathcal{S}_{++}^d$, and maps it onto $\mathcal{S}_{++}^d$.

**Dissipative dynamics.** While classical Hamiltonian dynamics conserve energy, dissipation and external inputs often appear in real-world systems. Both can be modeled in HNNs by complementing the Hamiltonian vector field with a force field $\boldsymbol{X}_{\mathcal{F}}$, so that the total vector field is $\boldsymbol{X} = \boldsymbol{X}_{\mathcal{H}} + \boldsymbol{X}_{\mathcal{F}}$ (Sosanya & Greydanus, 2022; Zhong et al., 2020a). We propose to leverage the structure-preserving symplectic submanifold and model dissipation and external inputs as a reduced-order force field $\check{\boldsymbol{X}}_{\mathcal{F}}$ on $(\check{\mathcal{M}}, \mathbb{J}_{2d}^{\mathsf{T}})$. Specifically, we consider high-dimensional systems with observed external inputs $\boldsymbol{\tau}_{\text{ext}}$ and viscous damping following a Rayleigh dissipative function $\mathcal{D}(\boldsymbol{q}, \dot{\boldsymbol{q}}) = \frac{1}{2}\dot{\boldsymbol{q}}^{\mathsf{T}} \boldsymbol{D}(\boldsymbol{q})\dot{\boldsymbol{q}}$ with unknown positive semi-definite dissipation matrix $\boldsymbol{D}(\boldsymbol{q})$. The resulting force field is $\boldsymbol{X}_{\mathcal{F}} = \begin{pmatrix} \boldsymbol{0} \\ \boldsymbol{\tau}_{\text{ext}} + \boldsymbol{\tau}_d \end{pmatrix}$ with damping force $\boldsymbol{\tau}_d = \frac{\partial \mathcal{D}(\boldsymbol{q}, \dot{\boldsymbol{q}})}{\partial \dot{\boldsymbol{q}}} = \boldsymbol{D}(\boldsymbol{q})\dot{\boldsymbol{q}}$.

**Proposition 3.** *The reduced vector field $\check{\boldsymbol{X}} = \varphi^* \boldsymbol{X}$ obtained via the pullback of the cotangent-lifted embedding $\varphi$ (8) preserves the structure of the total vector field $\boldsymbol{X} = \boldsymbol{X}_{\mathcal{H}} + \boldsymbol{X}_{\mathcal{F}}$ with $\boldsymbol{X}_{\mathcal{F}} = \begin{pmatrix} \boldsymbol{0} \\ \boldsymbol{\tau}_{ext} + \boldsymbol{\tau}_d \end{pmatrix}$.*

*Proof.* The reduced vector field decomposes as $\check{\boldsymbol{X}} = \varphi^* \boldsymbol{X} = \varphi^* \boldsymbol{X}_{\mathcal{H}} + \varphi^* \boldsymbol{X}_{\mathcal{F}} = \check{\boldsymbol{X}}_{\check{\mathcal{H}}} + \check{\boldsymbol{X}}_{\check{\mathcal{F}}}$ with $\check{\boldsymbol{X}}_{\check{\mathcal{H}}} = \check{\omega}^{-1} d\check{\mathcal{H}}$ (see Sec. 2.2) and $\check{\boldsymbol{X}}_{\check{\mathcal{F}}}$ is obtained by pulling back the external and damping terms. Since generalized forces belong to the cotangent bundle $\mathcal{T}^* \mathcal{Q}$, they are embedded and reduced via the cotangent-lifted maps (8) as $\varphi(\check{\boldsymbol{q}}, \check{\boldsymbol{\tau}})$ and $\rho(\boldsymbol{q}, \boldsymbol{\tau})$, leading to the reduced external inputs $\check{\boldsymbol{\tau}}_{\text{ext}} = d\varphi_{\mathcal{Q}}|_{\check{\boldsymbol{q}}}^{\mathsf{T}} \boldsymbol{\tau}_{\text{ext}}$. The reduced Rayleigh dissipative function is obtained via the pullback of the tangent-lifted embedding $\varphi_{\mathcal{T}\mathcal{Q}}(\check{\boldsymbol{q}}, \dot{\check{\boldsymbol{q}}}) = (\varphi_{\mathcal{Q}}(\check{\boldsymbol{q}})^{\mathsf{T}}, (d\varphi_{\mathcal{Q}}|_{\check{\boldsymbol{q}}}\dot{\check{\boldsymbol{q}}})^{\mathsf{T}})^{\mathsf{T}}$ as $\check{\mathcal{D}} = \varphi_{\mathcal{T}\mathcal{Q}}^* \mathcal{D} = \frac{1}{2}\dot{\check{\boldsymbol{q}}}^{\mathsf{T}} \check{\boldsymbol{D}}(\check{\boldsymbol{q}})\dot{\check{\boldsymbol{q}}}$ with positive semi-definite reduced damping matrix $\check{\boldsymbol{D}}(\check{\boldsymbol{q}}) = d\varphi_{\mathcal{Q}}^{\mathsf{T}} \boldsymbol{D}(\boldsymbol{q})d\varphi_{\mathcal{Q}}$. The reduced damping force is then $\check{\boldsymbol{\tau}}_d = \frac{\partial \check{\mathcal{D}}(\check{\boldsymbol{q}}, \dot{\check{\boldsymbol{q}}})}{\partial \dot{\check{\boldsymbol{q}}}} = \check{\boldsymbol{D}}(\check{\boldsymbol{q}})\dot{\check{\boldsymbol{q}}}$. Therefore, the reduced force field is $\check{\boldsymbol{X}}_{\check{\mathcal{F}}} = \begin{pmatrix} \boldsymbol{0} \\ \check{\boldsymbol{\tau}}_{\text{ext}} + \check{\boldsymbol{\tau}}_d \end{pmatrix}$. $\square$

We propose to model the reduced Rayleigh dissipative function $\check{\mathcal{D}}_{\boldsymbol{\theta}_{\check{D}}}(\check{\boldsymbol{q}}, \dot{\check{\boldsymbol{q}}}) = \frac{1}{2}\dot{\check{\boldsymbol{q}}}^{\mathsf{T}} \check{\boldsymbol{D}}_{\boldsymbol{\theta}_{\check{D}}}(\check{\boldsymbol{q}})\dot{\check{\boldsymbol{q}}}$ via a neural network $\check{\boldsymbol{D}}_{\boldsymbol{\theta}_{\check{D}}}$. Dissipative HNNs (Zhong et al., 2020a) constrain the dissipation matrix to

be positive semi-definite through its Cholesky decomposition, i.e., $\boldsymbol{D} = \boldsymbol{L}\boldsymbol{L}^{\mathsf{T}}$, thus overlooking its intrinsic geometric structure. Instead, we utilize a second SPD network $\check{\boldsymbol{D}}_{\boldsymbol{\theta}_{\check{D}}}(\check{\boldsymbol{q}}) = (g_{\mathrm{Exp}} \circ g_{\mathbb{R}})(\boldsymbol{q})$.

Note that the dissipative dynamics no longer preserve a symplectic structure. However, Proposition 3 shows that the RO-HNN latent space still preserves the structure of high-dimensional dissipative dynamical systems characterized by a Rayleigh dissipative function. It is worth noting that these systems are equivalent to Port-Hamiltonian systems with energy dissipation matrix $\boldsymbol{R}(\boldsymbol{q}, \boldsymbol{p}) = \begin{pmatrix} \mathbf{0} & \mathbf{0} \\ \mathbf{0} & D(\boldsymbol{q}) \end{pmatrix}$ and input $\boldsymbol{G}(\boldsymbol{q}, \boldsymbol{p})\boldsymbol{u} = \boldsymbol{\tau}_{\mathrm{ext}}$, and to contact Hamiltonians $\mathcal{H}_c(\boldsymbol{q}, \boldsymbol{p}, s)$ with $\dot{s} = \mathcal{D}$.

**Predicting dynamics.** The geometric HNN predicts conservative and dissipative dynamics as

$$\dot{\check{\boldsymbol{q}}}_{\mathrm{p}} = \frac{\partial \check{\mathcal{H}}_{\boldsymbol{\theta}}}{\partial \check{\boldsymbol{p}}} \qquad \text{and} \qquad \dot{\check{\boldsymbol{p}}}_{\mathrm{p}} = -\frac{\partial \check{\mathcal{H}}_{\boldsymbol{\theta}}}{\partial \check{\boldsymbol{q}}} - \check{\boldsymbol{D}}_{\boldsymbol{\theta}_D}(\check{\boldsymbol{q}}) \frac{\partial \check{\mathcal{H}}}{\partial \check{\boldsymbol{p}}} + \check{\boldsymbol{\tau}}_{\mathrm{ext}}, \tag{11}$$

where the predictions of the dynamic model are denoted via the subscript p. Note that $\check{\mathcal{D}}_{\boldsymbol{\theta}_D} = 0$ and $\check{\boldsymbol{\tau}}_{\mathrm{ext}} = \boldsymbol{0}$ in the conservative case. The architecture is illustrated in Fig. 1-*middle*.

Predicting system trajectories according to the learned reduced-order Hamiltonian dynamics involves *(1)* integrating the latent predictions $(\dot{\check{\boldsymbol{q}}}_{\mathrm{p}}, \dot{\check{\boldsymbol{p}}}_{\mathrm{p}})$ (11), and *(2)* decoding the obtained reduced-order position and momentum $(\check{\boldsymbol{q}}_{\mathrm{p}}, \check{\boldsymbol{p}}_{\mathrm{p}})$ into the high-dimensional coordinates of the original system with the lifted point embedding $\varphi$ (8), i.e., $(\tilde{\boldsymbol{q}}_{\mathrm{p}}, \tilde{\boldsymbol{p}}_{\mathrm{p}}) = \varphi(\check{\boldsymbol{q}}_{\mathrm{p}}, \check{\boldsymbol{p}}_{\mathrm{p}})$. In this paper, we propose to integrate the learned reduced-order Hamiltonian flow via symplectic integration, as explained next.

### 3.3 Trajectory Prediction via Symplectic Integration

Symplectic integrators are particularly well suited to integrate Hamiltonian dynamics as they preserve the geometric structure and invariants of the Hamiltonian flow (Leimkuhler & Reich, 2005). Symplectic integrators were shown to be key to accurately integrate learned HNNs dynamics, thus preventing long-term drifting of numerical solutions (Chen et al., 2020; Xiong et al., 2021).

The Hamiltonian dynamics learned in Sec. 3.2 are nonseparable, thus prohibiting the usage of standard explicit integration schemes, e.g., leapfrog (Leimkuhler & Reich, 2005). Instead, we integrate the reduced-order Hamiltonian flow (11) using the second-order symplectic integrator of (Tao, 2016) based on Strang splitting, akin to (Xiong et al., 2021). In a nutshell, the integrator considers an augmented Hamiltonian $\bar{\mathcal{H}}(\boldsymbol{q}, \boldsymbol{p}, \boldsymbol{x}, \boldsymbol{y}) = \mathcal{H}(\boldsymbol{q}, \boldsymbol{y}) + \mathcal{H}(\boldsymbol{p}, \boldsymbol{x}) + \frac{1}{2}w(\|\boldsymbol{q}, \boldsymbol{x}\|^2 + \|\boldsymbol{p}, \boldsymbol{y}\|^2)$ with extended phase space, for which high-order separable symplectic integrators with explicit updates can be constructed. A numerical integrator approximating $\bar{\mathcal{H}}$ is obtained by composing the obtained explicit flows, which we refer to as Strang-symplectic integrator. Additional details are provided in App. E.

### 3.4 Model Training

Finally, we propose to jointly learn the parameters $\{\boldsymbol{\Phi}_l, \boldsymbol{\Psi}_l, \boldsymbol{b}_l\}_{l=1}^{L}$ of the AE and $\{\boldsymbol{\theta}_{\check{T}}, \boldsymbol{\theta}_{\check{V}}, \boldsymbol{\theta}_{\check{D}}\}$ of the latent geometric HNN. As the learned dynamics are expected to predict multiple steps, we consider a loss that numerically integrates the latent predictions $(\dot{\check{\boldsymbol{q}}}_{\mathrm{p}}, \dot{\check{\boldsymbol{p}}}_{\mathrm{p}})$ (11) via $H$ forward Strang-symplectic integration steps before decoding. Given sets of observations $\{\boldsymbol{q}_i(\mathcal{I}_i), \boldsymbol{p}_i(\mathcal{I}_i), \boldsymbol{\tau}_i(\mathcal{I}_i)\}_{i=1}^{N}$ over intervals $\mathcal{I}_i = [t_i, t_i + H\Delta t]$ with constant integration time $\Delta t$, the resulting multi-step loss is

$$\ell_{\text{RO-HNN}} = \frac{1}{HN} \sum_{i=1}^{N} \sum_{j=1}^{H} \underbrace{\|\tilde{\boldsymbol{q}}_i(t_{i,j}) - \boldsymbol{q}_i(t_{i,j})\|^2 + \|\tilde{\boldsymbol{p}}_i(t_{i,j}) - \boldsymbol{p}_i(t_{i,j})\|^2}_{\ell_{\text{AE}}} + \underbrace{\lambda \|\check{\boldsymbol{q}}_{\mathrm{p},i}(t_{i,j}) - \check{\boldsymbol{q}}_i(t_{i,j})\|^2}_{\ell_{\text{HNN},d}} \tag{12}$$

$$+ \underbrace{\lambda \|\check{\boldsymbol{p}}_{\mathrm{p},i}(t_{i,j}) - \check{\boldsymbol{p}}_i(t_{i,j})\|^2}_{\ell_{\text{HNN},d}} + \underbrace{\|\tilde{\boldsymbol{q}}_{\mathrm{p},i}(t_{i,j}) - \boldsymbol{q}_i(t_{i,j})\|^2 + \|\tilde{\boldsymbol{p}}_{\mathrm{p},i}(t_{i,j}) - \boldsymbol{p}_i(t_{i,j})\|^2}_{\ell_{\text{HNN},n}} + \gamma \|\boldsymbol{\theta}\|_2^2,$$

where $\check{\boldsymbol{q}}_{\mathrm{p},i}(t_{i,j}) = \int_{t_i}^{t_{i,j}} \dot{\check{\boldsymbol{q}}}_{\mathrm{p},i} \mathrm{dt}$ and $\check{\boldsymbol{p}}_{\mathrm{p},i}(t_{i,j}) = \int_{t_i}^{t_{i,j}} \dot{\check{\boldsymbol{p}}}_{\mathrm{p},i} \mathrm{dt}$ with $t_{i,j} = t_i + j\Delta t$, and loss scaling $\lambda \in \mathbb{R}_{>0}$. Note that initial conditions to the latent IVPs are given by the encoded observations at the initial timestep $\check{\boldsymbol{q}}_{\mathrm{p},i}(t_i) := \check{\boldsymbol{q}}_i$ and $\check{\boldsymbol{p}}_{\mathrm{p},i}(t_i) := \check{\boldsymbol{p}}_i$. We optimize the network parameters via Riemannian Adam (Becigneul & Ganea, 2019).

## 4 Experiments

We evaluate the proposed RO-HNN to learn the dynamics of three simulated high-dimensional Hamiltonian systems: a 15-degrees-of-freedom (DoF) pendulum, a 600-DoF thin cloth, and a 90-DoF particle vortex. Our experiments showcase that RO-HNNs accurately predict long-term tra-

Table 1: Mean and standard deviation of prediction errors (↓) over 10 test pendulum trajectories.

| | $H\Delta t$ (s) | RO-HNN | 15-DoF HNN | 3-DoF HNN | HNKO |
|---|---|---|---|---|---|
| $\frac{\|\tilde{\boldsymbol{q}}_{\mathrm{p}}-\boldsymbol{q}\|}{\|\boldsymbol{q}\|}$ | 0.25 | $(1.66 \pm 1.38) \times 10^{-1}$ | $(5.33 \pm 6.02) \times 10^{-1}$ | $(\mathbf{1.22 \pm 0.92}) \times \mathbf{10^{-1}}$ | $(5.64 \pm 4.41) \times 10^{-1}$ |
| | 5 | $(7.08 \pm 7.56) \times 10^{-1}$ | — | $(\mathbf{5.44 \pm 6.93}) \times \mathbf{10^{-1}}$ | $(1.32 \pm 0.94) \times 10^{0}$ |
| $\frac{\|\tilde{\boldsymbol{p}}_{\mathrm{p}}-\boldsymbol{p}\|}{\|\boldsymbol{p}\|}$ | 0.25 | $(5.33 \pm 5.23) \times 10^{-2}$ | $(1.76 \pm 2.40) \times 10^{-1}$ | $(\mathbf{2.50 \pm 2.96}) \times \mathbf{10^{-2}}$ | $(5.93 \pm 10.73) \times 10^{-1}$ |
| | 5 | $(1.98 \pm 2.67) \times 10^{-1}$ | — | $(\mathbf{1.85 \pm 3.94}) \times \mathbf{10^{-1}}$ | $(1.23 \pm 2.08) \times 10^{0}$ |

Table 2: Mean and standard deviation of reconstruction, prediction, and symplecticity errors (↓) of intrusive symplectic dimensionality reduction approaches over 10 test pendulum trajectories.

| | Linear SMG | Quadr. SMG | Weakly-sympl. AE | Geom. Sympl. AE (ours) |
|---|---|---|---|---|
| $\|\tilde{\boldsymbol{q}} - \boldsymbol{q}\|/\|\boldsymbol{q}\|$ | $(2.21 \pm 1.17) \times 10^{-1}$ | $(2.84 \pm 4.23) \times 10^{0}$ | $(1.43 \pm 0.68) \times 10^{-1}$ | $(\mathbf{8.84 \pm 6.22}) \times \mathbf{10^{-2}}$ |
| $\|\tilde{\boldsymbol{p}} - \boldsymbol{p}\|/\|\boldsymbol{p}\|$ | $(4.43 \pm 3.99) \times 10^{-1}$ | $(2.75 \pm 1.60) \times 10^{-1}$ | $(1.57 \pm 1.55) \times 10^{-1}$ | $(\mathbf{4.09 \pm 3.99}) \times \mathbf{10^{-2}}$ |
| $\|\tilde{\boldsymbol{q}}_{\mathrm{p}} - \boldsymbol{q}\|/\|\boldsymbol{q}\|$ | $(2.58 \pm 2.33) \times 10^{-1}$ | $(3.53 \pm 5.18) \times 10^{0}$ | $(7.10 \pm 7.02) \times 10^{-1}$ | $(\mathbf{1.13 \pm 0.92}) \times \mathbf{10^{-1}}$ |
| $\|\tilde{\boldsymbol{p}}_{\mathrm{p}} - \boldsymbol{p}\|/\|\boldsymbol{p}\|$ | $(2.16 \pm 1.89) \times 10^{-1}$ | $(8.68 \pm 1.04) \times 10^{-1}$ | $(2.16 \pm 1.89) \times 10^{-1}$ | $(\mathbf{4.68 \pm 4.32}) \times \mathbf{10^{-2}}$ |
| $\|\mathbb{J}_{2d} - d\varphi^{\intercal}\mathbb{J}_{2n}d\varphi\|$ | $\mathbf{0.0 \pm 0.0}$ | $\mathbf{0.0 \pm 0.0}$ | $(1.67 \pm 0.35) \times 10^{-2}$ | $\mathbf{0.0 \pm 0.0}$ |
| $\|d\rho - d\varphi^{+}\|$ | $\mathbf{0.0 \pm 0.0}$ | $\mathbf{0.0 \pm 0.0}$ | $(5.32 \pm 1.45) \times 10^{0}$ | $(9.53 \pm 5.35) \times 10^{-1}$ |

jectories of high-dimensional Hamiltonian systems, highlighting the importance of embedding geometric inductive biases as hard constraints in the AE and HNN. Details about datasets, network architectures, and model training are provided in App. G. Additional results are provided in App. H.

### 4.1 COUPLED PENDULUM (15 DoF)

We consider a 15-DoF augmented pendulum whose nonlinear dynamics are specified from the symplectomorphism of a latent 3-DoF pendulum augmented with a 12-DoF mass-spring mesh. As the mesh oscillations are small, the system dynamics are approximately reducible to 3 dimensions.

**Learning high-dimensional dynamics.** We train a RO-HNN with latent dimension $d = 3$ and a conservative geometric HNN with Strang-symplectic integration on 3000 observations $\{\boldsymbol{q}_i, \boldsymbol{p}_i\}$ (see App. G.1 for details). We compare our RO-HNN with a 15-DoF geometric HNN that directly learns high-dimensional dynamic parameters, and with a Hamiltonian neural Koopman operator (HNKO) that learns a discrete linear predictor embedded in a 100-dimensional lifted space. For completeness, we also consider a 3-DoF geometric HNN trained on observations of the latent system. Notice that this model would not be deployable in practice as it requires ground truth information, i.e., latent observations, that would not be available.

Short- and long-term relative prediction errors over $H\Delta t = \{0.25, 5\}$s are reported in Table 1. The RO-HNN outperforms the 15-DoF HNN and the HNKO, leading to significantly lower prediction errors. Due to the high dimensionality, the 15-DoF HNN was difficult to train and did not lead to stable long-term predictions. As also shown in Fig. 2, the HNKO learns stable, but inaccurate long-term predictions. In contrast, the RO-HNN achieves similar long-term predictions as the 3-DoF HNN, which is expected to perform best as trained directly on the low-dimensional system (see also Figs. 7-8 in App. H.1). This validates the RO-HNN ability to jointly learn a latent symplectic submanifold and associated dynamics. Table 5 in App. H.1 shows that the RO-HNN is robust to observation noise, consistently outperforming the HNKO.

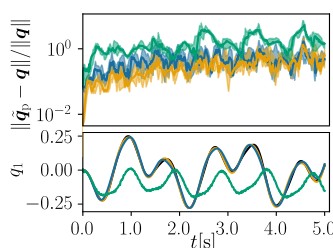

Figure 2: Median and quartiles of relative error and reconstructed trajectories of the RO-HNN (——), 3-DoF HNN (——), and HNKO (——) vs. ground truth (——) for a horizon $H\Delta t = 5$s. The 15-DoF HNN diverges and is not shown.

**AE architecture.** The quality of the learned symplectic submanifold is crucial for learning accurate dynamics, as they may systematically deviate from the ground truth if the submanifold does not accurately capture the solution space of the high-dimensional system. We analyze the influence of the reduction method in the RO-HNN and compare the proposed geometrically-constrained symplectic AE with linear and quadratic symplectic manifold Galerkin (SMG) projections (Peng & Mohseni, 2016; Sharma et al., 2023) which preserve the symplectic structure by construction, and a weakly-symplectic AE (Buchfink et al., 2023) which encourages structure preservation via a penalty term in the loss (see App. G.1 for details). We train each approach on 3000 observations of the 15-DoF pendulum. Here, we consider an intrusive MOR setup and project the known FOM dynamics onto the learned submanifold to predict new trajectories ($H\Delta t = 0.25$s). Table 2 shows that, due to their increased expressivity, the AEs outperform the linear and quadratic projections, with the geometrically-constrained symplectic AEs achieving

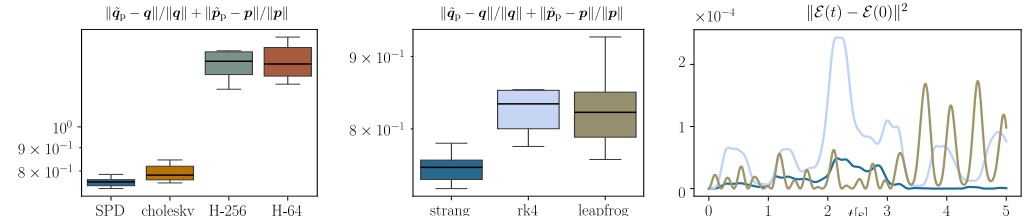

Figure 3: Ablation of the latent HNN architecture (*left*) and latent integrator (*middle*, *right*) of the RO-HNN to learn the dynamics of a 15-DoF pendulum.

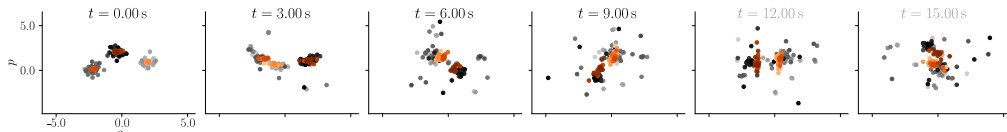

Figure 4: Predicted (•, •, •) vs ground truth (•, •, •) positions of the particle vortex. Times beyond 10s are out of the training data distribution.

the lowest reconstruction and prediction errors (see also Fig. 9 in App. H.1). Note that only the geometrically-constrained AE yielded stable longer-term predictions (see Fig. 2). Moreover, only the weakly-symplectic AE results in an error on the symplecticity condition (7), which is expected as both SMG projections and geometrically-constrained symplectic AE fulfill it by construction. Both SMG projections also ensure by design that the differential $d\rho$ is the symplectic inverse of $d\varphi$, while the geometrically-constrained AE leads to a lower error than the weakly-constrained one. Note that jointly training the AE with the geometric HNN in the RO-HNN further reduces this error to $(7.42 \pm 1.21) \times 10^{-1}$, showcasing the benefit of joint training.

**Latent HNN architecture.** We compare the performance of our geometric HNN to learn the low-dimensional dynamics of the latent 3-DoF pendulum against *(1)* a non-geometric variant that parametrizes the inverse mass-inertia matrix via a Cholesky network, and *(2)* two HNNs encoded as a single black-box network $\mathcal{H}_{\boldsymbol{\theta}}$, where we consider two MLPs of 64- and 256-neurons width. As shown in Fig. 3-*left*, the geometric HNN achieves the lowest reconstruction error, followed by the Cholesky HNN (see also App. H.1). This showcases the importance of considering both the quadratic energy structure of mechanical systems, and the geometry of their mass-inertia matrices.

**Latent integrator.** We compare the Strang symplectic integrator against a symplectic leapfrog that disregards that the Hamiltonian is non-separable, and a Runge-Kutta of order 4 that overlooks its symplectic structure. Fig. 3-*middle*, *right* show that the Strang symplectic integrator achieves the lowest reconstruction error and conserves energy best during integration (see also App. H.1).

### 4.2 PARTICLE VORTEX (90-DoF)

Next, we learn the dynamics of a particle vortex composed of $n = 90$ particles with uniform interaction strengths. As the particle vortex dynamics are purely determined via the logarithmic interaction, its Hamiltonian function does not separate into kinetic and potential energies. We train RO-HNNs with $d = \{6, 10\}$ and *(1)* a geometric HNN and *(2)* a black-box HNN $\mathcal{H}_{\boldsymbol{\theta}}$, both with Strang-symplectic integration (see App. G.2 for details). Fig. 4 depicts the predicted particle positions and momenta for a prediction horizon of $H = 100$, showing that the RO-HNN accurately predicts the particle vortex dynamics and generalizes beyond the data support ($t > 10$s). As shown in Table 3 and Fig. 13 in App. H.2 , the geometric HNNs outperform the black-box HNNs despite the lack of structure of the ground truth Hamiltonian. This suggests that the AE learns a symplectomorphism to a latent space where the Hamiltonian can be decomposed into two energy terms, thereby taking advantage of the additional structure of the geometric HNN. Moreover, the 6-dimensional models slightly outperform the 10-dimensional ones, showing that the choice of latent dimension trades off between the latent space expressivity and the limitations of HNNs in higher dimensions (see App. H.2 for more results).

Table 3: RO-HNN prediction errors ($\downarrow$) for different latent HNNs over 10 particle vortex trajectories.

| HNN | $d = 6$ | | $d = 10$ | |
| | $\|\tilde{q}_{\mathrm{p}} - q\|/\|q\|$ | $\|\tilde{p}_{\mathrm{p}} - p\|/\|p\|$ | $\|\tilde{q}_{\mathrm{p}} - q\|/\|q\|$ | $\|\tilde{p}_{\mathrm{p}} - p\|/\|p\|$ |
|---|---|---|---|---|
| Black-box | $(6.73 \pm 2.83) \times 10^{-1}$ | $(6.28 \pm 2.18) \times 10^{-1}$ | $(7.34 \pm 3.03) \times 10^{-1}$ | $(7.04 \pm 2.27) \times 10^{-1}$ |
| Geometric | $\mathbf{(4.00 \pm 2.01) \times 10^{-1}}$ | $\mathbf{(4.33 \pm 0.14) \times 10^{-1}}$ | $(4.44 \pm 0.60) \times 10^{-1}$ | $(4.60 \pm 0.32) \times 10^{-1}$ |

Figure 5: Predicted positions of the damped cloth with a RO-HNN with $d = 10$ for a $625\times$ longer horizon than during training. Times beyond $0.3$s are out of the training data distribution.

Table 4: Mean and standard deviation of RO-HNN reconstruction and prediction errors ($\downarrow$) for different parametrization of the latent dissipation matrix $\check{D}$ over 10 test cloth trajectories.

| DoF | $\check{D}$ | $\lVert \tilde{q}_{\mathrm{p}} - q \rVert / \lVert q \rVert$ | $\lVert \tilde{p}_{\mathrm{p}} - p \rVert / \lVert p \rVert$ | $\lVert \check{q}_{\mathrm{p}} - \check{q} \rVert / \lVert \check{q} \rVert$ | $\lVert \check{p}_{\mathrm{p}} - \check{p} \rVert / \lVert \check{p} \rVert$ |
|---|---|---|---|---|---|
| 6 | Cholesky | $(4.21 \pm 1.07) \times 10^{-2}$ | $(4.11 \pm 2.93) \times 10^{-1}$ | $(3.45 \pm 1.13) \times 10^{-2}$ | $(6.90 \pm 10.90) \times 10^{-2}$ |
| | SPD | $(4.15 \pm 2.10) \times 10^{-2}$ | $(3.58 \pm 3.03) \times 10^{-1}$ | $(3.11 \pm 1.86) \times 10^{-2}$ | $(8.42 \pm 10.40) \times 10^{-2}$ |
| | Ground truth | $(3.18 \pm 0.74) \times 10^{-2}$ | $(3.45 \pm 3.99) \times 10^{-1}$ | $(2.58 \pm 0.94) \times 10^{-2}$ | $(4.77 \pm 5.36) \times 10^{-2}$ |
| 10 | Cholesky | $(2.62 \pm 0.74) \times 10^{-2}$ | $(3.39 \pm 3.01) \times 10^{-1}$ | $(1.86 \pm 0.69) \times 10^{-2}$ | $(3.90 \pm 4.36) \times 10^{-2}$ |
| | SPD | $(3.21 \pm 1.25) \times 10^{-2}$ | $\mathbf{(3.33 \pm 2.88) \times 10^{-1}}$ | $(3.45 \pm 1.13) \times 10^{-2}$ | $\mathbf{(1.96 \pm 1.10) \times 10^{-2}}$ |
| | Ground truth | $\mathbf{(2.31 \pm 0.70) \times 10^{-2}}$ | $(3.44 \pm 3.05) \times 10^{-1}$ | $\mathbf{(1.37 \pm 0.63) \times 10^{-2}}$ | $(4.26 \pm 4.58) \times 10^{-2}$ |

## 4.3 CLOTH (600-DoF)

Next, we learn the dynamics of a high-deformable damped system, namely a simulated 600-DoF thin cloth falling onto spheres of different radius, akin to (Friedl et al., 2025). The system is intrinsically damped due to external dissipation forces $\tau_{\mathrm{d}}$. We train two RO-HNNs with $d = \{6, 10\}$ and a dissipative geometric HNN with Strang-symplectic integration on 20 trajectories of 3000 observations $\{q_i, p_i, \tau_i\}$ each, where $\tau = \tau_{\mathrm{c}}$ are measured external constraint forces (see App. G.3 for details). Fig. 5 depicts the predicted cloth configurations for a horizon $H\Delta t = 0.5\,\mathrm{s}$, showing that the RO-HNN accurately predicts the high-dimensional dissipative dynamics of the cloth, generalizing beyond the data support ($t > 0.3$s) (see App. H.3 for additional results and ablations).

**Latent damping.** We compare the performance of the dissipative RO-HNN against *(1)* a conservative RO-HNN, where the dissipation forces $\tau_{\mathrm{d}}$ are not learned but provided as ground truth in the external input $\tau = \tau_{\mathrm{c}} + \tau_{\mathrm{d}}$, and *(2)* a dissipative RO-HNN where the dissipation matrix is parametrized via Cholesky decomposition. Note that the mass-inertia matrix is parametrized via SPD networks in all cases. Table 4 shows that both dissipative RO-HNNs successfully learn the dissipation forces, achieving similar prediction errors as the conservative RO-HNN (see also Fig. 14 in App. H.3). The geometric HNN slightly outperforms its Cholesky counterpart, showing the importance of considering geometry. However, the effect is less pronounced as when learning the inverse mass-inertia matrix, which we attribute to the reduced influence of damping compared to inertia in the overall dynamics. As the dissipative dynamics do not preserve the symplectic structure, we compare the Strang symplectic integrator, which assumes a symplectic structure, against a non-symplectic Runge-Kutta integrator of order 4. Fig. 20 in App. H.3 shows that, despite the dissipative structure, the Strang symplectic integrator outperforms the Runge-Kutta one. We hypothesize that this is due to the fact that the evolution of this dissipative system is mostly governed by its Hamiltonian function, especially over the short timesteps taken by the integrators.

## 5 CONCLUSIONS

This paper proposed a novel physics-inspired neural network, RO-HNN, for learning the dynamics of high-dimensional Hamiltonian systems from data. Our model provides physically-consistent, accurate, and stable predictions that generalize beyond the data support. To achieve this, our model systematically integrates geometric inductive bias by defining structure-preserving symplectic embeddings, considering the geometry of the dynamics parameters within the model and for optimization, and leveraging structure-preserving symplectic integrators. We showed that the structural incorporation of these priors in the architecture is essential to learn high-dimensional dynamics, whereas Euclidean and soft-constrained approaches consistently underperformed. Future work will extend RO-HNN to Hamiltonian systems with non-canonical symplectic forms. To do so, we plan to leverage Darboux theorem and explore the development of local RO-HNNs. We will also generalize the RO-HNN to learn more general dynamics of Port-Hamiltonian and contact Hamiltonian systems. Finally, we will investigate model-based control strategies within the RO-HNN latent space.

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

# A RIEMANNIAN AND SYMPLECTIC GEOMETRY

In this section, we provide a short background on Riemannian and symplectic geometry, which compose the theoretical backbone of the RO-HNN. We refer the interested reader to (Abraham & Marsden, 1987; Lee, 2013) for more details.

As Riemannian and symplectic manifolds are smooth manifolds with special structures. A smooth manifold $\mathcal{M}$ of dimension $n$ can be intuitively conceptualized as a manifold that is locally, but not globally, similar to the Euclidean space $\mathbb{R}^n$. The smooth structure of $\mathcal{M}$ allows the definition of derivative of curves on the manifold, which are tangent vectors. The set of all tangent vectors at a point $\boldsymbol{x} \in \mathcal{M}$ defines the tangent space $\mathcal{T}_{\boldsymbol{x}}\mathcal{Q}$ which is a $n$-dimensional vector space. Tangent vectors can be represented on an ordered basis of $\mathcal{T}_{\boldsymbol{x}}\mathcal{Q}$ as $\boldsymbol{v} = v^i \frac{\partial}{\partial x^i}|_{\boldsymbol{x}}$. The tangent bundle $\mathcal{T}\mathcal{M}$ is the disjoint union of all tangent spaces on $\mathcal{M}$ and is $2n$-dimensional smooth manifold.

The cotangent space $\mathcal{T}_{\boldsymbol{x}}^*\mathcal{M}$ at $\boldsymbol{x} \in \mathcal{M}$ is the dual of the tangent space $\mathcal{T}_{\boldsymbol{x}}\mathcal{Q}$, i.e., $\mathcal{T}_{\boldsymbol{x}}^*\mathcal{M} = \{\lambda | \lambda : \mathcal{T}_{\boldsymbol{x}}\mathcal{Q} \to \mathbb{R} \text{ linear}\}$. Cotangent vectors can be represented on an ordered basis of $\mathcal{T}_{\boldsymbol{x}}^*\mathcal{M}$ as $\boldsymbol{\lambda} = \lambda_i \mathrm{d}x^i|_{\boldsymbol{x}}$. The cotangent bundle $\mathcal{T}^*\mathcal{M}$ is the disjoint union of all cotangent spaces on $\mathcal{M}$ and is $2n$-dimensional smooth manifold, similarly as the tangent bundle.

A smooth mapping $f$ between two smooth manifolds $\check{\mathcal{M}}$ and $\mathcal{M}$ with $\dim(\check{\mathcal{M}}) = d \ll \dim(\mathcal{M}) = n$ is an immersion if the differential $\mathrm{d}f|_{\check{\boldsymbol{x}}} : \mathcal{T}_{\check{\boldsymbol{x}}}\mathcal{M} \to \mathcal{T}_{f(\check{\boldsymbol{x}})}\mathcal{Q}$. An embedding is an immersion that is also a homeomorphism onto its image, i.e., it is an injective and structure-preserving map. In this case, $\check{\mathcal{M}}$ is an embedded submanifold of $\mathcal{M}$. The pullback of a function $h : \mathcal{M} \to \mathbb{R}$ by a smooth mapping $f : \mathcal{N} \to \mathcal{M}$ between two smooth manifolds $\mathcal{N}$ and $\mathcal{M}$ is a smooth function $f^*h$ with

$$f^*h(\boldsymbol{x}) = h(f(\boldsymbol{x})) = (h \circ f)(\boldsymbol{x}). \tag{13}$$

## A.1 RIEMANNIAN GEOMETRY

A Riemannian manifold $(\mathcal{M}, g)$ is a smooth manifold $\mathcal{M}$ endowed with a Riemannian metric $g$, i.e., a smoothly-varying inner product $g_{\boldsymbol{x}} : \mathcal{T}_{\boldsymbol{x}}\mathcal{Q} \times \mathcal{T}_{\boldsymbol{x}}\mathcal{Q} \to \mathbb{R}$. In coordinates, a Riemannian metric is represented by a SPD matrix. The Riemannian metric defines the notion of distance on the manifold, as well as the so-called geodesics, which are length-minimizing curves on the manifold.

Learning and optimization methods involving Riemannian data typically take advantage of their Euclidean tangent spaces to operate. Specifically, the exponential map $\mathrm{Exp}.\mathcal{T}_{\boldsymbol{x}}\mathcal{Q} \to \mathcal{M}$ and logarithmic map $\mathrm{Exp}.\mathcal{M} \to \mathcal{T}_{\boldsymbol{x}}\mathcal{Q}$, derived from the Riemannian metric, allows us to map back and forth between the Euclidean tangent space and the manifold. Moreover, the parallel transport $\mathrm{PT}_{\boldsymbol{x}\to\boldsymbol{y}} : \mathcal{T}_{\boldsymbol{x}}\mathcal{Q} \to \mathcal{T}_{\boldsymbol{y}}\mathcal{Q}$ move tangent vectors across tangent spaces such that their inner product is conserved.

A Lagrangian system $(\mathcal{M}, g, \mathcal{L})$ is a dynamical system evolving on a Riemannian manifold $(\mathcal{M}, g)$ according to a smooth Lagrangian function $\mathcal{L} : \mathcal{T}\mathcal{M} \to \mathbb{R}$.

## A.2 SYMPLECTIC GEOMETRY

A symplectic manifold $(\mathcal{M}, \omega)$ is a $2n$-dimensional smooth manifold $\mathcal{M}$ equipped with a symplectic form $\omega$, i.e., a closed, non-degenerate, differential 2-form $g_{\boldsymbol{x}} : \mathcal{T}_{\boldsymbol{x}}\mathcal{Q} \times \mathcal{T}_{\boldsymbol{x}}\mathcal{Q} \to \mathbb{R}$, which satisfies

$$\omega(\boldsymbol{u}, \boldsymbol{v}) = -\omega(\boldsymbol{v}, \boldsymbol{u}), \quad \omega(\boldsymbol{u}, \boldsymbol{v}) \,\forall \boldsymbol{v} \Rightarrow \boldsymbol{u} = \boldsymbol{0}, \quad \text{and} \quad d\omega = 0 \tag{14}$$

for all $\boldsymbol{u}, \boldsymbol{v} \in \mathcal{T}_{\boldsymbol{x}}\mathcal{Q}$. In coordinates, a symplectic form is represented by a skew-symmetric matrix $\boldsymbol{\omega}$. We slightly abuse notation, equivalently denoting symplectic manifolds as $(\mathcal{M}, \boldsymbol{\omega})$. Notice that the non-degeneracy of $\omega$ implies that all symplectic manifolds are of even dimension.

A diffeomorphism $f : (\mathcal{M}, \boldsymbol{\omega}) \to (\mathcal{N}, \boldsymbol{\eta})$ between symplectic manifolds is a symplectomorphism if it preserves the symplectic form, i.e., $f^*\eta = \omega$ with $f^*\eta$ denoting the pullback of $\eta$ by $f$. The Hamiltonian flow $\phi_t : (\mathcal{M}, \boldsymbol{\omega}) \to (\mathcal{M}, \boldsymbol{\omega})$ induced by $\boldsymbol{X}_{\mathcal{H}}$ is a symplectomorphism, as it maps points $\boldsymbol{x} \in \mathcal{M}$ along the integral curves of the manifold thus preserving the symplectic form.

Following Darboux' theorem, there exists a canonical chart $(U, \phi)$, $\boldsymbol{x} \in U$ for each point $\boldsymbol{x} \in \mathcal{M}$ in which the symplectic form is represented as $\boldsymbol{\omega} = \mathbb{J}_{2n}^{\mathsf{T}}$ via the canonical Poisson tensor

$$\mathbb{J}_{2n} = \begin{pmatrix} \boldsymbol{0} & \boldsymbol{I}_n \\ -\boldsymbol{I}_n & \boldsymbol{0} \end{pmatrix}, \quad \text{for which} \quad \mathbb{J}_{2n}^{\mathsf{T}} = \mathbb{J}_{2n}^{-1} = -\mathbb{J}_{2n}.$$

In other words, every symplectic manifold is locally symplectomorphic to $(\mathbb{R}^{2n}, \mathbb{J}_{2n}^{\intercal})$. A system $(\mathbb{R}^{2n}, \mathbb{J}_{2n}^{\intercal}, \mathcal{H})$ is called a canonical Hamiltonian system. Moreover, the cotangent bundle $\mathcal{T}^*\mathcal{Q}$ any $n$-dimensional smooth manifold $\mathcal{Q}$ carries a canonical symplectic structure, making it a symplectic manifold $(\mathcal{T}^*\mathcal{Q}, \mathbb{J}_{2n})$.

A Hamiltonian system $(\mathcal{M}, \boldsymbol{\omega}, \mathcal{H})$ is a dynamical system evolving on a symplectic manifold $(\mathcal{M}, \boldsymbol{\omega})$ according to a smooth Hamiltonian function $\mathcal{H} : \mathcal{M} \to \mathbb{R}$.

# B  RIEMANNIAN MANIFOLDS OF INTEREST

This section provides a brief overview of the Riemannian manifolds of interest for this paper, namely the manifold of SPD matrices $\mathcal{S}_{++}^n$ (App. B.1), and the biorthogonal manifold $\mathcal{B}_{n,d}$ (App. B.2).

## B.1  THE MANIFOLD OF SPD MATRICES

We denote the set of $n \times n$ symmetric matrices as $\mathrm{Sym}^n = \{\boldsymbol{S} \in \mathbb{R}^{n \times n} | \boldsymbol{S} = \boldsymbol{S}^{\intercal}\}$. The set of SPD matrices $\mathcal{S}_{++}^n = \{\boldsymbol{\Sigma} \in \mathrm{Sym}^n \mid \boldsymbol{\Sigma} \succ \boldsymbol{0}\}$ forms a smooth manifold of dimension $\dim(\mathcal{S}_{++}^n) = \frac{n(n+1)}{2}$, which can be represented as the interior of a convex cone embedded in $\mathrm{Sym}^n$. The tangent space $\mathcal{T}_{\boldsymbol{\Sigma}}\mathcal{S}_{++}^n$ at a point $\boldsymbol{\Sigma} \in \mathcal{S}_{++}^n$ is identified with $\mathrm{Sym}^n$.

The SPD manifold can be endowed with various Riemannian metrics, resulting in different theoretical properties and closed-form operations. We utilize the widely-used affine-invariant metric (Pennec et al., 2006), which places symmetric matrices with non-positive eigenvalues at infinite distance from any SPD matrix and prevents the well-known swelling effect (Feragen & Fuster, 2017). The affine-invariant metric defines the inner product $g : \mathcal{T}_{\boldsymbol{\Sigma}}\mathcal{S}_{++}^n \times \mathcal{T}_{\boldsymbol{\Sigma}}\mathcal{S}_{++}^n \to \mathbb{R}$ given two matrices $\boldsymbol{T}_1$, $\boldsymbol{T}_2 \in \mathcal{T}_{\boldsymbol{\Sigma}}\mathcal{S}_{++}^n$, as

$$\langle \boldsymbol{T}_1, \boldsymbol{T}_2 \rangle_{\boldsymbol{\Sigma}} = \mathrm{tr}(\boldsymbol{\Sigma}^{-\frac{1}{2}} \boldsymbol{T}_1 \boldsymbol{\Sigma}^{-1} \boldsymbol{T}_2 \boldsymbol{\Sigma}^{-\frac{1}{2}}). \tag{15}$$

The corresponding geodesic distance, exponential map, logarithmic maps, and parallel transport are computed in closed form as

$$d_{\mathcal{M}}(\boldsymbol{\Lambda}, \boldsymbol{\Sigma}) = \| \log(\boldsymbol{\Sigma}^{-\frac{1}{2}} \boldsymbol{\Lambda} \boldsymbol{\Sigma}^{-\frac{1}{2}}) \|_{\mathrm{F}}, \tag{16}$$

$$\mathrm{Exp}_{\boldsymbol{\Sigma}}(\boldsymbol{S}) = \boldsymbol{\Sigma}^{\frac{1}{2}} \exp(\boldsymbol{\Sigma}^{-\frac{1}{2}} \boldsymbol{S} \boldsymbol{\Sigma}^{-\frac{1}{2}}) \boldsymbol{\Sigma}^{\frac{1}{2}}, \tag{17}$$

$$\mathrm{Log}_{\boldsymbol{\Sigma}}(\boldsymbol{\Lambda}) = \boldsymbol{\Sigma}^{\frac{1}{2}} \log(\boldsymbol{\Sigma}^{-\frac{1}{2}} \boldsymbol{\Lambda} \boldsymbol{\Sigma}^{-\frac{1}{2}}) \boldsymbol{\Sigma}^{\frac{1}{2}}, \tag{18}$$

$$\mathrm{PT}_{\boldsymbol{\Sigma} \to \boldsymbol{\Lambda}}(\boldsymbol{T}) = \boldsymbol{A}_{\boldsymbol{\Sigma} \to \boldsymbol{\Lambda}} \, \boldsymbol{T} \, \boldsymbol{A}_{\boldsymbol{\Sigma} \to \boldsymbol{\Lambda}}^{\intercal}, \tag{19}$$

where $\exp(\cdot)$ and $\log(\cdot)$ denote the matrix exponential and logarithm functions, and $\boldsymbol{A}_{\boldsymbol{\Sigma} \to \boldsymbol{\Lambda}} = \boldsymbol{\Lambda}^{\frac{1}{2}} \boldsymbol{\Sigma}^{-\frac{1}{2}}$. These operations are key for the SPD networks encoding the mass-inertia and damping matrices in geometric HNNs (see Sec. 3.2), and for the on-manifold parameter optimization of SPD parameters of the network when training the model (see Sec. 3.4).

## B.2  THE BIORTHOGONAL MANIFOLD

The biorthogonal manifold is the smooth manifold $\mathcal{B}_{n,d} = \{(\boldsymbol{\Phi}, \boldsymbol{\Psi}) \in \mathbb{R}^{n \times d} \times \mathbb{R}^{n \times d} \mid \boldsymbol{\Psi}^{\intercal} \boldsymbol{\Phi} = \boldsymbol{I}_d\}$ formed by pairs of full-row-rank matrices $\boldsymbol{\Phi}, \boldsymbol{\Psi} \in \mathbb{R}^{n \times d}$, with $n \geq d \geq 1$ satisfying the biorthogonality condition $\boldsymbol{\Psi}^{\intercal} \boldsymbol{\Phi} = \boldsymbol{I}$ (Otto et al., 2023). The biorthogonal matrix manifold $\mathcal{B}_{n,d}$ is an embedded submanifold of the Euclidean product space $\mathbb{R}^{n \times d} \times \mathbb{R}^{n \times d}$ with dimension $\dim(\mathcal{B}_{n,d}) = 2nd - d^2$. The tangent space at a point $(\boldsymbol{\Phi}, \boldsymbol{\Psi}) \in \mathcal{B}_{n,d}$ is given by

$$\mathcal{T}_{(\boldsymbol{\Phi}, \boldsymbol{\Psi})}\mathcal{B}_{n,d} = \{(\boldsymbol{V}, \boldsymbol{W}) \in \mathbb{R}^{n \times d} \times \mathbb{R}^{n \times d} \; : \; \boldsymbol{W}^{\intercal} \boldsymbol{\Phi} + \boldsymbol{\Psi}^{\intercal} \boldsymbol{V} = \boldsymbol{0}\}. \tag{20}$$

A pair of matrices $(\boldsymbol{X}, \boldsymbol{Y}) \in \mathbb{R}^{n \times d} \times \mathbb{R}^{n \times d}$ can be projected onto the tangent space $\mathcal{T}_{(\boldsymbol{\Phi}, \boldsymbol{\Psi})}\mathcal{B}_{n,d}$ via the projection operation $\mathrm{Proj}_{(\boldsymbol{\Phi}, \boldsymbol{\Psi})} : \mathbb{R}^{n \times d} \times \mathbb{R}^{n \times d} \to \mathcal{T}_{(\boldsymbol{\Phi}, \boldsymbol{\Psi})}\mathcal{B}_{n,d}$ defined as

$$\mathrm{Proj}_{(\boldsymbol{\Phi}, \boldsymbol{\Psi})}(\boldsymbol{X}, \boldsymbol{Y}) = (\boldsymbol{X} - \boldsymbol{\Psi}\boldsymbol{A}, \boldsymbol{Y} - \boldsymbol{\Phi}\boldsymbol{A}^{\intercal}), \tag{21}$$

where $\boldsymbol{A}$ is a solution to the Sylvester equation $\boldsymbol{A}(\boldsymbol{\Phi}^{\intercal} \boldsymbol{\Phi}) + (\boldsymbol{\Psi}^{\intercal} \boldsymbol{\Psi})\boldsymbol{A} = \boldsymbol{Y}^{\intercal} \boldsymbol{\Phi} + \boldsymbol{\Psi}^{\intercal} \boldsymbol{X}$.

When optimizing the parameters of the geometrically-constrained symplectic AE presented in Sec. 3.1, it is crucial to account for the biorthogonal geometry of the pairs of weight matrices (Friedl et al., 2025). Therefore, we train the model by optimizing pairs of weight matrices via Riemannian optimization on the biorthogonal manifold (see Sec. 3.4). Riemannian optimization algorithms utilize the exponential map and the parallel transport operations, which are difficult to obtain in closed form for the biorthogonal manifold. Therefore, we leverage a first-order approximation of the exponential map, i.e., a retraction map $R_{(\boldsymbol{\Phi}, \boldsymbol{\Psi})} : \mathcal{T}_{(\boldsymbol{\Phi}, \boldsymbol{\Psi})} \mathcal{B}_{n,d} \to \mathcal{B}_{n,d}$, defined as

$$R_{(\boldsymbol{\Phi}, \boldsymbol{\Psi})}(\boldsymbol{V}, \boldsymbol{W}) = \left( (\boldsymbol{\Phi} + \boldsymbol{V}) \left( (\boldsymbol{\Psi} + \boldsymbol{W})^{\mathsf{T}} (\boldsymbol{\Phi} + \boldsymbol{V}) \right)^{-1}, (\boldsymbol{\Psi} + \boldsymbol{W}) \right). \tag{22}$$

Moreover, we use a first-order approximation of the parallel transport operation defined via the successive application of retraction and projection as

$$\mathrm{PT}_{(\boldsymbol{\Phi}_1, \boldsymbol{\Psi}_1) \to (\boldsymbol{\Phi}_2, \boldsymbol{\Psi}_2)} = \mathrm{Proj}_{(\boldsymbol{\Phi}_2, \boldsymbol{\Psi}_2)} \circ R_{(\boldsymbol{\Phi}_1, \boldsymbol{\Psi}_1)}. \tag{23}$$

## C  SPD Network

As explained in Sec. 3.2, we learn reduced Hamiltonian dynamics in the embedded symplectic submanifold via a latent geometric HNN that parametrizes the inverse mass-inertia and damping matrices via SPD networks that account for their intrinsic geometry. We use a SPD network introduced in (Friedl et al., 2025) composed of *(1)* Euclidean layers $g_{\mathbb{R}}$), and *(2)* an exponential map layer $g_{\mathrm{Exp}}$, which we detail next.

**Euclidean Layers** $g_{\mathbb{R}}$. The SPD network leverages classical fully-connected layers to model functions that return elements on the tangent space of a manifold. The output of the $l$-th Euclidean layer $\boldsymbol{x}^{(l)}$ is given by

$$\boldsymbol{x}^{(l)} = \sigma(\boldsymbol{A}_l \boldsymbol{x}^{(l-1)} + \boldsymbol{b}_l), \tag{24}$$

with $\boldsymbol{A}_l \in \mathbb{R}^{n_l \times n_{l-1}}$ and $\boldsymbol{b}_l \in \mathbb{R}^{n^{(l)}}$ the weight matrix and bias of the layer $l$, and $\sigma$ a nonlinear activation function of choice.

**Exponential Map Layer** $g_{\mathrm{Exp}}$. The exponential map layer is used to map layer inputs $\boldsymbol{X}^{(l-1)} \in \mathrm{Sym}^n$ from the tangent space onto the manifold $\mathcal{S}_{++}^n$. The layer output is given by

$$\boldsymbol{X}^{(l)} = \mathrm{Exp}_{\boldsymbol{P}}(\boldsymbol{X}^{(l-1)}), \tag{25}$$

with $\boldsymbol{P} \in \mathcal{S}_{++}^n$ denoting the basepoint of the considered tangent space. Following the results of the ablation conducted in (Friedl et al., 2025), we define $\boldsymbol{P}$ as equal to the identity matrix $\mathbf{I}$, so that the layer input is assumed to lie in the tangent space at the origin of the cone.

Note that Friedl et al. (2025) additionally consider SPD layers mapping SPD matrices to SPD matrices, analogous to fully-connected Euclidean layers. However, the SPD networks with additional SPD layers were shown to achieve similar performances as those employing solely Euclidean and exponential-map layers. Therefore, we do not integrate such layers in the SPD networks of the RO-HNN.

## D  Additional Details on the Geometrically-constrained Symplectic Autoencoder

### D.1  Constrained Autoencoder

The geometrically-constrained symplectic AE presented in Sec. 3.1 builds on the constrained AE architecture introduced in (Otto et al., 2023). Specifically, we learn the embedding $\varphi_{\mathcal{Q}}$ and associated point reduction $\rho_{\mathcal{Q}}$ via a constrained AE with layer pairs (10), and compute their differential to construct the tangent-lifted maps (8), as explained in Sec. 3.1. To guarantee the projection properties, the constrained AE architecture from (Otto et al., 2023) leverages pairs of biorthogonal weight matrices, which are described in Sec. 3.1, and pairs of invertible activation functions, which we introduce next.

The nonlinear activation functions $\sigma_-$ and $\sigma_+$ employed in the encoder and decoder network must satisfy $\sigma_- \circ \sigma_+ = \mathrm{id}$. To do so, they are defined as

$$\sigma_\pm(x_i) = \frac{bx_i}{a} \mp \frac{\sqrt{2}}{a\sin(\alpha)} \pm \frac{1}{a}\sqrt{\left(\frac{2x_i}{\sin(\alpha)\cos(\alpha)} \mp \frac{\sqrt{2}}{\cos(\alpha)}\right)^2 + 2a}, \tag{26}$$

with

$$\begin{cases} a &= \csc^2(\alpha) - \sec^2(\alpha), \\ b &= \csc^2(\alpha) + \sec^2(\alpha). \end{cases} \tag{27}$$

The activations then resemble smooth, rotation-symmetric versions of the common leaky ReLu activations. The parameter $0 < \alpha < \frac{\pi}{4}$ sets the slope of the activation functions. Throughout our experiments, we set $\alpha = \frac{\pi}{8}$.

Otto et al. (2023) proposed to incorporate the biorthogonality of the weight matrices by considering an overparametrization of the biorthogonal weights along with a soft constraint in the form of additional penalty losses. However, this approach does not guarantee the biorthogonality condition, in contrast to the Riemannian approach we use in this paper. Moreover, as shown in (Friedl et al., 2025), the overparametrized model leads to higher reconstruction errors compared to constrained AE trained on the biorthogonal manifold.

### D.2 COMPUTATION OF THE COTANGENT-LIFTED MAPS

We construct the cotangent-lifted maps (8) by differentiating the outputs of the encoder $\rho_{\mathcal{Q}}$ and decoder $\varphi_{\mathcal{Q}}$ networks with respect to their inputs. To avoid the computational cost related to the automatically-differentiated transposed Jacobian-vector product, our implementation computes layer-wise analytical derivatives and obtains the full differentials via the chain rule. The derivatives of the nonlinear activations $\sigma_\pm$ are given by

$$\sigma'_\pm(x_i) = \frac{d}{dx_i}\sigma_\pm(x_i) = \frac{b}{a} \pm \frac{2}{a\sin(\alpha)\cos(\alpha)} \frac{\frac{2x_i}{\sin(\alpha)\cos(\alpha)} \mp \frac{\sqrt{2}}{\cos(\alpha)}}{\sqrt{\left(\frac{2x_i}{\sin(\alpha)\cos(\alpha)} \mp \frac{\sqrt{2}}{\cos(\alpha)}\right)^2 + 2a}}, \tag{28}$$

thus fulfilling the inverse-derivative property $\sigma'_-(\sigma_+(x_i))\,\sigma'_+(x_i) = 1$ by construction.

The pullbacks $d\rho_{\mathcal{Q}}|^{\mathsf{T}}_{\varphi_{\mathcal{Q}}(\check{q})}\check{p}$ and $d\varphi_{\mathcal{Q}}|^{\mathsf{T}}_{\rho_{\mathcal{Q}}(q)}p$ are computed analytically as a composition of transposed layer derivatives $d\rho_{\mathcal{Q}} = d\rho_{\mathcal{Q}}^{(L)\mathsf{T}} \circ \ldots \circ d\rho_{\mathcal{Q}}^{(1)\mathsf{T}}$ and $d\varphi_{\mathcal{Q}} = d\varphi_{\mathcal{Q}}^{(1)\mathsf{T}} \circ \ldots \circ d\varphi_{\mathcal{Q}}^{(L)\mathsf{T}}$, with $d\rho_{\mathcal{Q}}^{(l)} \in \mathbb{R}^{n_{l-1} \times n_l}$ and $d\varphi_{\mathcal{Q}}^{(l)} \in \mathbb{R}^{n_l \times n_{l-1}}$. From the definition of the layer pairs (10), the transpose of the layer derivatives are given as

$$d\rho_{\mathcal{Q}}^{(l)}|^{\mathsf{T}}_{q^{(l-1)}} = \mathbf{\Psi}_l \mathrm{diag}(\sigma'_-(q^{(l-1)})) \qquad \text{and} \qquad d\varphi_{\mathcal{Q}}^{(l)}|^{\mathsf{T}}_{\check{q}^{(l-1)}} = \mathrm{diag}(\sigma'_+(\check{q}^{(l-1)}))\mathbf{\Phi}_l^{\mathsf{T}}, \tag{29}$$

with $\mathrm{diag}(v) = \begin{pmatrix} v_1 & \ldots & 0 \\ \vdots & \ddots & \vdots \\ 0 & \ldots & v_d \end{pmatrix}$.

The computation of the layer derivatives requires storing the intermediate reduced and reconstructed positions $\check{q}^{(l-1)}$ and $q^{(l-1)}$ for each layer, which are obtained during the forward pass through the position encoder $\rho_{\mathcal{Q}}$ and decoder $\varphi_{\mathcal{Q}}$. During the momentum forward pass, we store each intermediate $p^{(l-1)} = d\rho_{\mathcal{Q}}^{(l)\mathsf{T}}p^{(l)}$ and $\check{p}^{(l)} = d\varphi_{\mathcal{Q}}^{(l)\mathsf{T}}p^{(l-1)}$. This allows the computational cost of one momentum forward pass to roughly be equal to that of one forward pass of the position projection, scaling constantly through the matrix-multiplication of weights with system dimensionality $\dim(\mathcal{Q})$. We provide wall-clock evaluation times of our geometrically-constrained symplectic AE on the 600-DoF cloth dataset in App. G.3 .

## E  STRANG-SYMPLECTIC INTEGRATOR

As explained in Sec. 3.3, we integrate the learned reduced-order Hamiltonian flow (11) using the second-order symplectic integrator of (Tao, 2016), which we refer to as Strang-symplectic integrator.

The Strang-symplectic integrator approximates the flow of a non-separable Hamiltonian function $\mathcal{H}(\boldsymbol{q}, \boldsymbol{p})$ by considering an augmented Hamiltonian function

$$\bar{\mathcal{H}}(\boldsymbol{q}, \boldsymbol{p}, \boldsymbol{x}, \boldsymbol{y}) = \mathcal{H}_A(\boldsymbol{q}, \boldsymbol{y}) + \mathcal{H}_B(\boldsymbol{p}, \boldsymbol{x}) + w\mathcal{H}_C(\boldsymbol{q}, \boldsymbol{p}, \boldsymbol{x}, \boldsymbol{y}), \tag{30}$$

in an extended phase space, where $\mathcal{H}_A(\boldsymbol{q}, \boldsymbol{y})$ and $\mathcal{H}_B(\boldsymbol{p}, \boldsymbol{x})$ are two copies of the original system with mixed-up positions and momenta, and $\mathcal{H}_C = \frac{1}{2}(\|\boldsymbol{q}, \boldsymbol{x}\|^2 + \|\boldsymbol{p}, \boldsymbol{y}\|^2)$ is an artificial restraint with parameter $w$ controlling the binding of $\mathcal{H}_A(\boldsymbol{q}, \boldsymbol{y})$ and $\mathcal{H}_B(\boldsymbol{p}, \boldsymbol{x})$. The dynamics of the augmented Hamiltonian $\bar{\mathcal{H}}$ are

$$\dot{\boldsymbol{q}} = \frac{\partial}{\partial \boldsymbol{p}} \bar{\mathcal{H}}(\boldsymbol{q}, \boldsymbol{p}, \boldsymbol{x}, \boldsymbol{y}) = \frac{\partial}{\partial \boldsymbol{p}} H(\boldsymbol{x}, \boldsymbol{p}) + w(\boldsymbol{p} - \boldsymbol{y}) \tag{31}$$

$$\dot{\boldsymbol{p}} = \frac{\partial}{\partial \boldsymbol{q}} \bar{\mathcal{H}}(\boldsymbol{q}, \boldsymbol{p}, \boldsymbol{x}, \boldsymbol{y}) = \frac{\partial}{\partial \boldsymbol{q}} H(\boldsymbol{q}, \boldsymbol{y}) - w(\boldsymbol{q} - \boldsymbol{x}) \tag{32}$$

$$\dot{\boldsymbol{x}} = \frac{\partial}{\partial \boldsymbol{y}} \bar{\mathcal{H}}(\boldsymbol{q}, \boldsymbol{p}, \boldsymbol{x}, \boldsymbol{y}) = \frac{\partial}{\partial \boldsymbol{y}} H(\boldsymbol{q}, \boldsymbol{y}) + w(\boldsymbol{y} - \boldsymbol{p}) \tag{33}$$

$$\dot{\boldsymbol{y}} = \frac{\partial}{\partial \boldsymbol{p}} \bar{\mathcal{H}}(\boldsymbol{q}, \boldsymbol{p}, \boldsymbol{x}, \boldsymbol{y}) = \frac{\partial}{\partial \boldsymbol{p}} H(\boldsymbol{x}, \boldsymbol{p}) - w(\boldsymbol{x} - \boldsymbol{q}) \tag{34}$$

and leads to the same exact IVP solutions as the original function $\mathcal{H}(\boldsymbol{q}, \boldsymbol{p})$. High-order symplectic integrators can be construct for each of the component of the augmented Hamiltonian $\bar{\mathcal{H}}$ as

$$\phi_{\mathcal{H}_A}^{\delta} = \begin{pmatrix} \boldsymbol{q} \\ \boldsymbol{p} - \delta \frac{\partial}{\partial \boldsymbol{q}} H(\boldsymbol{q}, \boldsymbol{y}) \\ \boldsymbol{x} + \delta \frac{\partial}{\partial \boldsymbol{y}} H(\boldsymbol{q}, \boldsymbol{y}) \\ \boldsymbol{y} \end{pmatrix}, \quad \phi_{\mathcal{H}_B}^{\delta} = \begin{pmatrix} \boldsymbol{q} + \delta \frac{\partial}{\partial \boldsymbol{p}} H(\boldsymbol{x}, \boldsymbol{p}) \\ \boldsymbol{p} \\ \boldsymbol{x} \\ \boldsymbol{y} - \delta \frac{\partial}{\partial \boldsymbol{x}} H(\boldsymbol{x}, \boldsymbol{p}) \end{pmatrix}, \tag{35}$$

$$\phi_{w\mathcal{H}_C}^{\delta} = \frac{1}{2} \begin{pmatrix} \begin{pmatrix} \boldsymbol{q} + \boldsymbol{x} \\ \boldsymbol{p} + \boldsymbol{y} \end{pmatrix} + \boldsymbol{R}(\delta) \begin{pmatrix} \boldsymbol{q} - \boldsymbol{x} \\ \boldsymbol{p} - \boldsymbol{y} \end{pmatrix} \\ \begin{pmatrix} \boldsymbol{q} + \boldsymbol{x} \\ \boldsymbol{p} + \boldsymbol{y} \end{pmatrix} - \boldsymbol{R}(\delta) \begin{pmatrix} \boldsymbol{q} - \boldsymbol{x} \\ \boldsymbol{p} - \boldsymbol{y} \end{pmatrix} \end{pmatrix}, \quad \text{with} \quad \boldsymbol{R}(\delta) = \begin{pmatrix} \cos(2w\delta)\boldsymbol{I} & \sin(2w\delta)\boldsymbol{I} \\ -\sin(2w\delta)\boldsymbol{I} & \cos(2w\delta)\boldsymbol{I} \end{pmatrix}. \tag{36}$$

Tao (2016) proposed to construct a numerical symplectic integrator that approximates the flow of $\bar{\mathcal{H}}$ by composing these maps according to Strang splitting as

$$\phi_{\bar{\mathcal{H}}} = \phi_{\mathcal{H}_A}^{\delta/2} \circ \phi_{\mathcal{H}_B}^{\delta/2} \circ \phi_{w\mathcal{H}_C}^{\delta/2} \circ \phi_{\mathcal{H}_B}^{\delta/2} \circ \phi_{\mathcal{H}_A}^{\delta/2}. \tag{37}$$

The obtained Strang-symplectic integrator preserves the symplectic volume like the exact Hamiltonian flow.

The scalar parameter $w \in \mathbb{R}$, binding the two augmented Hamiltonians during the integration process, is obtained as optimization parameter during training. To enforce $w \leq 0$, we do not learn $w$ directly. Instead, we learn it using the SoftPlus function with a small numerical offset for stability as $\log(1 + e^{\boldsymbol{\theta}_w}) + 10^{-4}$, as part of the HNN network parameters $\boldsymbol{\theta}_w \in \boldsymbol{\theta}$.

## F NETWORK TRAINING VIA RIEMANNIAN OPTIMIZATION

Training a neural network corresponds to finding a solution to an optimization problem

$$\min_{\boldsymbol{x} \in \mathcal{M}} \ell(\boldsymbol{x}), \tag{38}$$

where $\ell$ is the loss we aim to minimize, and $\boldsymbol{x} \in \mathcal{M}$ is the optimization variable, a.k.a the network parameters. For the RO-HNN, we train the network by minimizing the loss $\ell_{\text{RO-HNN}}$ (12). In this case, $\mathcal{M}$ is defined as a product of Euclidean, SPD, and biorthogonal manifolds to jointly optimize the parameters $\{\boldsymbol{\Phi}_l, \boldsymbol{\Psi}_l, \boldsymbol{b}_l\}_{l=1}^L$ of the AE and $\{\boldsymbol{\theta}_{\tilde{T}}, \boldsymbol{\theta}_{\tilde{V}}, \boldsymbol{\theta}_{\tilde{D}}\}$ of the latent geometric HNN. To account for the curvature of the non-Euclidean parameter spaces, we leverage Riemannian optimization (Absil et al., 2007; Boumal, 2023) to optimize the RO-HNN loss $\ell_{\text{RO-HNN}}$ (12).

Conceptually, each iteration step in a first-order (stochastic) Riemannian optimization method consists of the three following successive operations:

$$\boldsymbol{\eta}_t \leftarrow h\big(\text{grad}\, \ell(\boldsymbol{x}_t), \boldsymbol{\tau}_{t-1}\big), \quad \boldsymbol{x}_{t+1} \leftarrow \text{Exp}_{\boldsymbol{x}_t}(-\alpha_t \boldsymbol{\eta}_t), \quad \boldsymbol{\tau}_t \leftarrow \text{PT}_{\boldsymbol{x}_t \to \boldsymbol{x}_{t+1}}(\boldsymbol{\eta}_t). \tag{39}$$

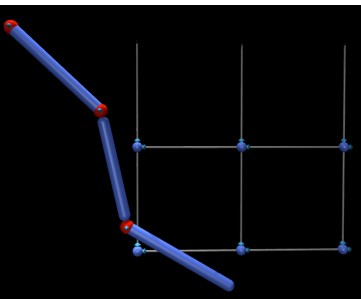

Figure 6: Illustration of the latent system used to obtain the dynamics of a 15-DoF augmented pendulum via a symplectomorphism. The latent system consists of an independent 3-DoF planar pendulum and a 12-DoF planar mass-spring mesh.

where *(1)* given the current parameter estimate $x_t$, a search direction $\eta_t \in \mathcal{T}_{x_t}\mathcal{M}$ is computed based on a function $h$ (determined by the choice of the optimization method) of the Riemannian gradient grad $\ell$, and of $\tau_{t-1}$, which corresponds to the parallel-transport of the previous search direction on to the new estimate's tangent space $\mathcal{T}_{x_t}\mathcal{M}$; *(2)* the estimate $x_t$ is updated by projecting the search direction $\eta_t$ scaled by a learning rate learning rate $\alpha_t$ onto the manifold via the exponential map, *(3)* the current search direction is parallel-transported to the tangent space of the updated estimate to prepare for the next iteration. In this paper, we use the Riemannian Adam (Becigneul & Ganea, 2019) implemented in Geoopt (Kochurov et al., 2020) to optimize the RO-HNN parameters. The relevant manifold operations for the optimization procedure are given in closed-form in App. B.

# G    ADDITIONAL DETAILS ON EXPERIMENTS

This section presents additional details on the experimental setup of Sec. 4.

## G.1    COUPLED PENDULUM OF SECTION 4.1

### G.1.1    DATASET

**System.** Our first set of experiments is conducted on the augmented pendulum, a nonlinear conservative system with $n = 15$-DoF. The pendulum dynamics are specified from the symplectomorphism of a latent Hamiltonian system composed of two independent subsystems: a 3-DoF planar pendulum, and a 12-DoF planar oscillating mass-spring mesh, see Fig. 6. The pendulum dynamics evolve on a slower timescale and with larger amplitude than the mesh oscillations. Consequently, a surrogate model based solely on the pendulum would capture the dominant behavior of the full system, i.e. the system is well-reducible with a Hamiltonian ROM. As we have access to the ground truth dynamics of the 15-DoF pendulum, this scenario allows for various ablations on the network architecture.

We simulate both subsystems in MUJOCO (Todorov et al., 2012). The pendulum's links $i = \{1, 2, 3\}$ are modeled as capsules of radius $r_i = 0.025$ m, length $l_i = 0.5$ m, and mass $m_i = 0.5$ kg, connected via hinge joints. The initial configurations and velocities for each DoF are randomly sampled from the intervals $q_{\text{pend},i}(t = 0) \in [-30, 30]\,^\circ$ and $\dot{q}_{\text{pend},i}(t = 0) \in [-23, 23]\,^\circ\,\text{s}^{-1}$. The mass-spring mesh consists of 6 masses $m_j = 0.005$ kg, equally spaced in a $3 \times 2$ grid along the $x-$ and $z-$axes of the simulation environment. Each mass is connected to its immediate neighbors, and the top three masses are each additionally connected to a fixed anchor point above the grid, via springs of resting length $s_j = 0.5$ m and linear stiffness constants $k_j = 0.01\,\text{N}\,\text{m}^{-1}$. Initial displacements and velocities for each DoF are randomly sampled from the intervals $q_{\text{ms},j}(t = 0) \in [-1, 1] \times 10^{-2}$m and $\dot{q}_{\text{ms},j}(t = 0) \in [-2, 2] \times 10^{-3}\text{m}\,\text{s}^{-1}$.

**Data generation.** Each simulation is recorded for $T = 5$ s at a timestep of $\Delta t = 10^{-2}$s, yielding $N = 30$ training trajectories $\mathcal{D}_{\text{pend}} = \{\{q_{\text{pend},n,k}, p_{\text{pend},n,k}\}_{k=1}^K\}_{n=1}^N$ and $\mathcal{D}_{\text{ms}} = \{\{q_{\text{ms},n,k}, p_{\text{ms},n,k}\}_{k=1}^K\}_{n=1}^N$ with $K = 500$ observations each. To form the full 15-dimensional dataset, the position and momentum vectors of the pendulum and mass-spring mesh are con-

catenated as $\boldsymbol{q}_{\mathrm{aug}} = (\boldsymbol{q}_{\mathrm{pend}}^{\intercal} \quad \boldsymbol{q}_{\mathrm{ms}}^{\intercal})^{\intercal}$ and $\boldsymbol{p}_{\mathrm{aug}} = (\boldsymbol{p}_{\mathrm{pend}}^{\intercal}, \quad \boldsymbol{p}_{\mathrm{ms}}^{\intercal})^{\intercal}$ to obtain a dataset $\mathcal{D}_{\mathrm{aug}} = \{\{\boldsymbol{q}_{\mathrm{aug},n,k}, \boldsymbol{p}_{\mathrm{aug},n,k}\}_{k=1}^{K}\}_{n=1}^{N}$ with $(\boldsymbol{q}_{\mathrm{aug},n,k}, \boldsymbol{p}_{\mathrm{aug},n,k}) \in \mathcal{T}^*(\mathcal{Q}_{\mathrm{pend}} \times \mathcal{Q}_{\mathrm{ms}})$.

To ensure that the reducibility of the augmented dataset is not purely of numerical nature, we transform the observed dynamics of the latent system onto more complex ones via a symplectomorphism $h : (\mathcal{T}^*(\mathcal{Q}_{\mathrm{pend}} \times \mathcal{Q}_{\mathrm{ms}}), \mathbb{J}_{2n}) \rightarrow (\mathcal{T}^*\mathcal{Q}, \mathbb{J}_{2n})$ and obtain the final dataset $\mathcal{D} = \{\{h(\boldsymbol{q}_{\mathrm{aug},n,k}, \boldsymbol{p}_{\mathrm{aug},n,k})\}_{k=1}^{K}\}_{n=1}^{N}$. Practically, the symplectomorphism $h$ is defined via the cotangent-lifted embedding $\varphi$ of a map $\rho_{\mathcal{Q}} : \mathcal{Q}_{\mathrm{pend}} \times \mathcal{Q}_{\mathrm{ms}} \rightarrow \mathcal{Q}$, that we parametrize as a 3-layer encoder of the constrained AE from Sec. 3.1 and D. With $l = \{1, 2, 3\}$ layers of constant layer and latent dimension $n_l = n_0 = 15$, weights of each layer $\boldsymbol{\Psi}_l$ initialized as random orthogonal matrices $\boldsymbol{O} \in \mathbb{R}^{n_l \times n_l}$ sampled from the Haar distribution, and zero biases $\boldsymbol{b}_l = \boldsymbol{0}$. Notice that due to the constant dimension through the AE-layers, with decoder weights set to $\boldsymbol{\Phi}_l = \boldsymbol{\Psi}_l = \boldsymbol{O}$, the position decoder of the constrained AE returns an analytic inverse, and its cotangent lift $h^{-1}$.

The testing dataset is constructed in the same manner for $N = 10$ trajectories.

### G.1.2 Model Training

For the experiments of Sec. 4.1, we train a geometric RO-HNN composed of a geometrically-constrained symplectic AE and a latent geometric HNN. As described in Sec. 3.1, the geometrically-constrained symplectic AE is built from the cotangent lift of a constrained AE composed of layer pairs $\rho_{\mathcal{Q}}^{(l)} : \mathbb{R}^{n_l} \rightarrow \mathbb{R}^{n_{l-1}}$ and $\varphi_{\mathcal{Q}}^{(l)} : \mathbb{R}^{n_{l-1}} \rightarrow \mathbb{R}^{n_l}$ as defined in (10) (see Sec. 3.1 and App. D). We use $l = \{1, 2, 3\}$ pairwise biorthogonal encoder and decoder layers of sizes $n_l = \{6, 12, 15\}$ with latent space dimension $n_0 = 3$. The biorthogonal weight matrices are initialized by sampling a random orthogonal matrix $\boldsymbol{O} \in \mathbb{R}^{n_l \times n_l}$ from the Haar distribution and setting $\boldsymbol{\Phi} = \boldsymbol{\Psi} = \boldsymbol{O}_{[:,:n_{l-1}]}$, where $\boldsymbol{O}_{[:,:n_{l-1}]}$ are the first $n_{l-1}$ column entries of $\boldsymbol{O}$. Bias vectors are initialized as $\boldsymbol{b}_l = \boldsymbol{0}$. For the latent geometric HNN, we parametrize the potential energy network $\check{V}_{\boldsymbol{\theta}_{\check{V}}}$ and the Euclidean part $g_{\mathbb{R}}$ of the inverse mass-inertia network $\check{\boldsymbol{M}}_{\boldsymbol{\theta}_{\check{T}}}^{-1}$ each with $L_{\check{V}} = L_{\check{T},\mathbb{R}} = 2$ hidden Euclidean layers of 32 neurons and SoftPlus activation functions. We fix the basepoint of the exponential map layer $g_{\mathrm{Exp}}$ to the origin $\boldsymbol{P} = \mathbf{I}$. Weights are initialized by sampling from a Xavier normal distribution with gain $\sqrt{2}$ and bias vector entries set to 1. We train the model on the joint loss (12) with scaling factor $\lambda = 1$ for the latent loss on 3000 uniformly sampled random points from the dataset $\mathcal{D}$ with Strang-symplectic integration (see Sec. 3.3) over a training horizon of $H_{\mathcal{D}} = 12$ timesteps. We use a learning rate of $1.5 \times 10^{-2}$ for the AE parameters and $7 \times 10^{-4}$ for the HNN parameters. We train the model with Riemannian Adam (Becigneul & Ganea, 2019) until convergence at 3000 epochs.

**AE baselines.** In Sec. 4.1, we compare the geometrically-constrained symplectic AE with linear and quadratic symplectic manifold Galerkin (SMG) projections (Peng & Mohseni, 2016; Sharma et al., 2023), and a weakly-symplectic AE (Buchfink et al., 2023). We implement the linear and quadratic SMG projections onto a 3-dimensional symplectic submanifold following (Sharma et al., 2023). We compute the reduction parameters based on a singular value decomposition computed from 3000 randomly sampled training datapoints in $\mathcal{D}$.

For the weakly-symplectic AE (Buchfink et al., 2023), we train two independent constrained AEs for position and momentum reduction and embedding, i.e., $\rho_{\mathcal{Q}}^{(l)} : \mathbb{R}^{n_l} \rightarrow \mathbb{R}^{n_{l-1}}$, $\varphi_{\mathcal{Q}}^{(l)} : \mathbb{R}^{n_{l-1}} \rightarrow \mathbb{R}^{n_l}$, and $\rho_{\mathcal{P}}^{(l)} : \mathbb{R}^{n_l} \rightarrow \mathbb{R}^{n_{l-1}}$, $\varphi_{\mathcal{P}}^{(l)} : \mathbb{R}^{n_{l-1}} \rightarrow \mathbb{R}^{n_l}$, and compute the embedding and reduction for the symplectic manifold as

$$\varphi(\check{\boldsymbol{q}}, \check{\boldsymbol{p}}) = \begin{pmatrix} \varphi_{\mathcal{Q}} \\ \varphi_{\mathcal{P}} \end{pmatrix} \qquad \text{and} \qquad \rho(\boldsymbol{q}, \boldsymbol{p}) = \begin{pmatrix} \rho_{\mathcal{Q}} \\ \rho_{\mathcal{P}} \end{pmatrix}. \tag{40}$$

Note that this architecture also fulfills the projection properties (4) by construction, as the other reduction approaches. However, it does not satisfy the symplecticity property (7). To enforce this property, Buchfink et al. (2023) introduces a symplecticity loss

$$\ell_{\mathrm{sympl}} = \frac{1}{N} \sum_{i=1}^{N} \|\mathbb{J}_{2d} - d\varphi^{\intercal} \mathbb{J}_{2n} d\varphi\|_{\mathrm{F}}^2. \tag{41}$$

The weakly-symplectic AE is trained by minimizing the sum of the reconstruction loss $\ell_{\mathrm{AE}}$ from (12) and the symplecticity loss (41).

For the geometrically-constrained symplectic AE, we consider the same architecture as in the RO-HNN described above.

All AE architectures consist of $l = \{1, 2, 3\}$ biorthogonal encoder and decoder layers with $n_l = \{6, 12, 15\}$ with latent space dimension $n_0 = 3$. We train both AE on 3000 samples from the dataset $\mathcal{D}$ with Riemannian Adam with a learning rate of $1.5 \times 10^{-2}$ until convergence at 3000 epochs.

**HNN baselines.** In Sec. 4.1, we also ablate the choice of latent HNN and integrator. To isolate the HNN performance, we consider the low-dimensional dataset $\mathcal{D}_{\text{pend}}$ of the 3-DoF pendulum and no reduction. For the Cholesky HNN where the inverse mass-inertia matrix is parametrized via a Cholesky network, we implement shared parameters for the inverse mass-matrix and potential energy networks, i.e., $\boldsymbol{\theta}_{\tilde{T}} \cap \boldsymbol{\theta}_{\tilde{V}}$, following (Lutter & Peters, 2023). The MLP consists of 2 hidden Euclidean SoftPlus layers of 64 neurons, while separate output layers return the potential energy and the Cholesky decomposition. For the black-box HNNs, we use a single fully-connected MLP to model a Hamiltonian function $\mathcal{H}_{\boldsymbol{\theta}}$. We conduct experiments with two black-box HNNs of 2 hidden layers with a width of 64, and 256 neurons, respectively. In all cases, the weights are initialized by sampling from a Xavier normal distribution with gain $\sqrt{2}$, and the bias vector entries are initialized to 1.

We train all architectures on 3000 datapoints of the dataset $\mathcal{D}_{\text{pend}}$ with Riemannian Adam optimizer on the HNN term $\ell_{\text{HNN},d}$ of the loss (12) over a training horizon of $H_{\mathcal{D}} = 12$ timesteps. For the ablation of the HNN architecture, we use the Strang-symplectic integrator. The geometric HNN and Cholesky networks are trained until convergence at 2500 epochs with learning rate set to $7 \times 10^{-4}$. The black-box HNNs are trained at a learning rate of $2 \times 10^{-3}$ for 3000 epochs.

For the ablation of the integrator, we use the geometric HNN and compare the Strang-symplectic integrator with an explicit Euler integrator, a Runge-Kutta integrator of order 4, and a symplectic leapfrog integrator.

**HNKO baseline.** In Sec. 4.1, we compare the RO-HNN against the HNKO proposed by Zhang et al. (2024). Moreover, in App. H.1, we evaluate the performance of the RO-HNN against the HNKO under noisy observations. Hyperparameters are selected and refined empirically following the supplementary material and code provided by Zhang et al. (2024).

The HNKO first maps the $2n = 30$-dimensional observations into a $d = 100$-dimensional lifted latent space via a fully-connected neural network of 6 hidden layers with Tanh activations. The latent dynamics are then propagated on a 50-dimensional sphere via the special orthogonal Hamiltonian Koopman operator, implemented by a constrained linear, bias-free layer with 100-dimensional input and output. The predicted states are mapped back onto the original space with a fully-connected neural network with 3 hidden layers and Tanh activations. The overall model is trained on 3000 randomly sampled datapoints of the dataset with $\mathcal{D}_{\text{pend}}$, using the Adam optimizer until convergence at 15000 epochs. For a fair comparison and for stable predictions over longer horizons, we adjusted the loss on the latent Koopman predictions, referred to as $\mathcal{L}_{\text{koop}}$ in (Zhang et al., 2024), to sum over a training horizon of $H_{\mathcal{D}} = 12$ timesteps, similar to our latent loss term $\ell_{\text{HNN},d}$ in (12).

### G.2 PARTICLE VORTEX OF SECTION 4.2

#### G.2.1 DATASET

**System.** In Sec. 4.2, we learn the dynamics of an $n = 90$-dimensional particle vortex, consisting of $j = \{1, ..., N\}$ particles with phase-space coordinates $\boldsymbol{x}_j = (q_j, p_j)^{\mathsf{T}}$ and uniform interaction strengths $\Gamma_j = 1$. The particle vortex dynamics are governed by the Hamiltonian

$$\mathcal{H}(\boldsymbol{q}, \boldsymbol{p}) = -\sum_{j<k} \log |\boldsymbol{x}_j - \boldsymbol{x}_k|, \tag{42}$$

that models the interaction between each $j \neq k$ pair of particles (Xiong et al., 2021). Note that, as the particle vortex dynamics are purely determined via the logarithmic interaction, its Hamiltonian function does not separate into kinetic and potential energy, in contrast to mechanical systems such as the pendulum and the cloth.

**Data generation.** We generate a training dataset $\mathcal{D}_{\text{pv}} = \{\{\boldsymbol{q}_{n,k}, \boldsymbol{p}_{n,k}\}_{k=1}^K\}_{n=1}^N$ by simulating $N = 20$ trajectories of the conservative system over the time interval $\mathcal{I} = [0, 10.0]$s with timestep $\Delta t =$

$10^{-3}$s and Strang-symplectic solver with weight parameter $w = 0.1$, resulting in $K = 10000$ steps per trajectory. For each trajectory, initial conditions are randomly sampled to mimic clustered vortex distributions. The particles are evenly split among $j = \{1, 2, 3\}$ clusters. For each cluster, we randomly sample a center $\boldsymbol{c}_j$ within a radius of $R = 6$ m form the origin. Then, a cluster radius is sampled uniformly from $r_j \in [0.1, 2]$m, and particles within a cluster are positioned following a Gaussian distribution $\mathcal{N} \sim (\boldsymbol{c}_j, r_j^2 \mathbf{I})$ around the center $\boldsymbol{c}_j$. For the testing dataset, we generate $N = 10$ trajectories via the same distribution of initial conditions, but simulating the system over a time interval of $\mathcal{I} = [0, 15.0]$s.

### G.2.2 Model Training

The results presented in Sec. 4.2 are obtained via RO-HNNs composed of a geometrically-constrained symplectic AE and a latent geometric HNN. We conduct experiments with two RO-HNN with latent space dimensions $d = 3$ and $d = 6$. The constrained AE is composed of $l = \{1, 2, 3, 4\}$ pairwise biorthogonal encoder and decoder layers of sizes $n_l = \{32, 64, 128, 600\}$. The biorthogonal weight matrices are initialized by sampling a random orthogonal matrix $\boldsymbol{O} \in \mathbb{R}^{n_l \times n_l}$ from the Haar distribution and setting $\boldsymbol{\Phi} = \boldsymbol{\Psi} = \boldsymbol{O}_{[:,:n_{l-1}]}$, where $\boldsymbol{O}_{[:,:n_{l-1}]}$ are the first $n_{l-1}$ column entries of $\boldsymbol{O}$. Bias vectors are initialized as $\boldsymbol{b}_l = \mathbf{0}$. The latent Hamiltonian network $\check{V}_{\boldsymbol{\theta}_{\check{V}}}$ is parametrized by 2 hidden Euclidean layers of 32 neurons with SoftPlus activation functions. All weights are initialized by sampling from a Xavier normal distribution with gain $\sqrt{2}$, and all bias vector entries are initialized to 1.

We train the model on the joint loss (12) on 3000 random samples from the dataset $\mathcal{D}$ with Strang-symplectic integration over a training horizon $H_{\mathcal{D}} = 8$ timesteps. For better convergence, we scale the loss term $\ell_{\text{HNN,d}}$ via a scalar factor $\lambda = 10^3$. The parameters are optimized via Riemannian Adam (Becigneul & Ganea, 2019) until convergence at 3000 epochs with a learning rate of $1.5 \times 10^{-2}$ for the AE parameters and $7 \times 10^{-4}$ for the HNN parameters.

In Sec. 4.2, we consider a comparison with a RO-HNN with a latent black-box HNN $\check{\mathcal{H}}_{\boldsymbol{\theta}}$ composed of 2 layers of 64 neurons. We set the learning rate to $2 \times 10^{-3}$. The remaining of the RO-HNN architecture and training pipeline are unchanged.

### G.3 Cloth of Section 4.3

### G.3.1 Dataset

**System.** Our second set of experiments is conducted on a deformable thin cloth modeled in MUJOCO as a flexible composite object with $i = \{1, ..., 200\}$ masses $m_i = 0.1$ kg, equally spaced over a width of $0.1$ m and length of $0.2$ m. Generalized coordinates are given by the Cartesian positions $\boldsymbol{q}_i = (x_i, y_i, z_i)^\mathsf{T}$ of each mass' center of mass in the world frame. The viscous damping coefficient is uniformly set to $d_i = 0.01$N s m$^{-1}$.

**Data generation.** Each trajectory captures the cloth falling on a sphere from a height of $0.12$ m in the center above the origin of the sphere. To vary scenarios, the radius of the sphere is randomly-sampled from $r \in [0.02, 0.12]$ m. The state evolution is simulated with timestep $\Delta t = 10^{-4}$ s over a time interval $\mathcal{I} = [0, 0.3]$s, resulting in $K = 3000$ samples per trajectory. We generate $N = 20$ trajectories for a training dataset $\mathcal{D}_{\text{cloth}} = \{\{\boldsymbol{q}_{n,k}, \boldsymbol{p}_{n,k}, \boldsymbol{\tau}_{n,k}\}_{k=1}^K\}_{n=1}^N$, and $N = 10$ testing trajectories over a longer time interval $\mathcal{I} = [0, 0.5]$s. When learning the damping force via a dissipative HNN, the generalized force vector consists of external constraint forces, i.e., $\boldsymbol{\tau} = \boldsymbol{\tau}_c$. The ablation of Sec. 4.3 compares the dissipative geometric HNN against a conservative HNN for which all external forces are provided. In this case, the training dataset is composed of generalized force vector $\boldsymbol{\tau} = \boldsymbol{\tau}_d + \boldsymbol{\tau}_c$ that contains both the damping forces $\boldsymbol{\tau}_d$ and the constraint forces $\boldsymbol{\tau}_c$.

### G.3.2 Model Training

For the RO-HNN experiments in Sec. 4.3, we train a RO-HNN composed of a geometrically-constrained symplectic AE and a latent dissipative geometric HNN. The underlying constrained AE $l = \{1, 2, 3, 4\}$ pairwise biorthogonal encoder and decoder layers of sizes $n_l = \{32, 64, 128, 600\}$ with latent space dimension $n_0 = 6$ or $n_0 = 10$. The biorthogonal weight matrices are initialized by sampling a random orthogonal matrix $\boldsymbol{O} \in \mathbb{R}^{n_l \times n_l}$ from the Haar distribution and setting

$\boldsymbol{\Phi}_l = \boldsymbol{\Psi}_l = \boldsymbol{O}_{[:,:n_{l-1}]}$, where $\boldsymbol{O}_{[:,:n_{l-1}]}$ are the first $n_{l-1}$ column entries of $\boldsymbol{O}$. Bias vectors are initialized as $\boldsymbol{b}_l = \boldsymbol{0}$.

The latent potential energy network $\check{V}_{\boldsymbol{\theta}_{\check{V}}}$ is parametrized with $L_{\check{V}} = L_{\check{T},\mathbb{R}} = 2$ hidden Euclidean layers of 32 neurons. The Euclidean part $g_{\mathbb{R}}$ of the inverse mass-inertia network $\check{\boldsymbol{M}}_{\boldsymbol{\theta}_{\check{T}}}^{-1}$ and of the damping-matrix network $\check{\boldsymbol{D}}_{\boldsymbol{\theta}_{\check{D}}}$ are composed of 2 hidden layers with 32 neurons. For both networks, we fix the basepoint of the exponential map layer $g_{\mathrm{Exp}}$ to the origin $\boldsymbol{P} = \mathbf{I}$. All activation functions are SoftPlus, all weights are initialized by sampling from a Xavier normal distribution with gain $\sqrt{2}$, and all bias vector entries are initialized to 1.

We train the model on the joint loss (12) on 3000 samples from the dataset $\mathcal{D}$ with Strang-symplectic integration over a training horizon $H_{\mathcal{D}} = 8$ timesteps. The scaling constant on the latent loss term $\ell_{\mathrm{HNN,d}}$ is set to $\lambda = 10^4$. We train the RO-HNN with Riemannian Adam (Becigneul & Ganea, 2019) until convergence at 3000 epochs with a learning rate of $1.5 \times 10^{-2}$ for the AE parameters and $7 \times 10^{-4}$ for the HNN parameters.

In Fig. 21, we compare the geometric RO-HNN with a black-box variant where the latent HNN is encoded as a single black-box network $\mathcal{H}_{\boldsymbol{\theta}}$ corresponding to a fully-connected MLP of 2 hidden layers with a width of 256 neurons. The HNN weights are initialized by sampling from a Xavier normal distribution with gain $\sqrt{2}$, and the bias vector entries are initialized to 1. This black-box RO-HNN is training with the same parameters as the geometric RO-HNN, except for the learning rate of the HNN parameters, which we set as $2 \times 10^{-3}$.

**Sequentially-trained baseline.** To assess the effectiveness of the proposed joint training procedure, we compare the jointly-trained RO-HNN with a variant that sequentially trains first the geometrically-constrained AE, and second the latent HNN. As convergence is difficult to achieve when training only the latent HNN on a fully-trained representation of the AE, we first train only the AE by optimizing $\ell_{\mathrm{AE}}$ for 3000 epochs with a learning rate of $1.5 \times 10^{-2}$. Subsequently, we jointly optimize the AE and latent loss (12). We train the networks jointly within the RO-HNN with Riemannian Adam (Becigneul & Ganea, 2019) until convergence at learning rates $1.5 \times 10^{-2}$ for the AE parameters and $7 \times 10^{-4}$ for the HNN parameters.

**Projection and AE baselines.** In App. H.3 (see Fig. 18), we compare the ability of a latent HNN to learn accurate dynamics using different reduction methods to obtain the symplectic embedding $\varphi$ and corresponding reduction $\rho$. We compare the RO-HNN with geometrically-constrained symplectic AE with linear and quadratic symplectic manifold Galerkin (SMG) projections (Peng & Mohseni, 2016; Sharma et al., 2023), and a weakly-symplectic AE (Buchfink et al., 2023). We compute the linear and quadratic SMG projections onto latent spaces of symplectic submanifolds of three different dimensionalities $d = \{2, 6, 10\}$, following (Sharma et al., 2023), via 3000 training datapoints. In both cases, we then train a latent HNN on the terms $\ell_{\mathrm{HNN,n}}$ and $\ell_{\mathrm{HNN,d}}$ of the joint loss equation (12) on 3000 samples from the dataset $\mathcal{D}$ with Strang-symplectic integration over a training horizon $H_{\mathcal{D}} = 8$ timesteps. The model is trained with Riemannian Adam (Becigneul & Ganea, 2019) until convergence at 3000 epochs with a learning rate of $7 \times 10^{-4}$. Note that this essentially corresponds to a scenario with pre-trained symplectic submanifolds, as the parameter optimization for the linear and quadratic embedding maps happens once in the beginning.

The weakly-symplectic AE consists of two independent constrained AEs for position and momentum. We use 4 layers of size $n_l = \{32, 64, 128, 600\}$ with varying latent space dimension. We train the network jointly on the sum of the losses (12) and (41) via Riemannian Adam (Becigneul & Ganea, 2019) until convergence at 3000 epochs with a learning rate of $1.5 \times 10^{-2}$ for the AE parameters and $7 \times 10^{-4}$ for the HNN parameters.

**Non-symplectic AE baseline.** In App. H.3 (see Fig. 20), we ablate geometrically-constrained symplectic AE of the RO-HNN in a dissipative scenario. To do so, we train a RO-HNN that utilizes a non-symplectic projection-constrained AE instead of geometrically-constrained symplectic AE. Specifically, we use a single vanilla constrained AE with latent space dimension $n_0 = 20$ with 4 pairwise biorthogonal encoder-decoder layers of sizes $n_l = \{64, 128, 256, 1200\}$. Note that we assume the first 10 output dimensions of the latent space to correspond to reduced position $\check{\boldsymbol{q}}$ and the last 10 to correspond to the reduced momentum $\check{\boldsymbol{p}}$, which are used as inputs for the latent dissipative HNN.

**Comparison against reduced-order LNN (RO-LNN).** In App. H.3, we compare the RO-HNNs against RO-LNNs (Friedl et al., 2025). The RO-LNNs are trained on the dataset $\mathcal{D}_{\text{cloth, vel}} = \{\{q_{n,k}, \dot{q}_{n,k}, \tau_{n,k}\}_{k=1}^{K}\}_{n=1}^{N}$ obtained from the dataset described in App. G.3.1 by transforming the momentum data into velocities via $\dot{q} = M(q)^{-1} p$.

We construct the RO-LNN following the procedure described in (Friedl et al., 2025) and use the same architecture for the constrained AE as for the RO-HNN. Specifically, we consider a latent space of dimension $n_0 = 10$ and use $l = \{1, 2, 3, 4\}$ pairwise biorthogonal encoder and decoder layers of sizes $n_l = \{32, 64, 128, 600\}$. The kinetic and potential energy networks of the latent geometric LNN consist of 2 hidden Euclidean layers with 64 neurons and SoftPlus activation functions, initialized as for the RO-HNN. Notice that, for the RO-LNN, the dissipation forces $\tau_d$ are not learned but provided as ground truth in the external input $\tau = \tau_c + \tau_d$.

The RO-LNN is trained on 3000 samples from $\mathcal{D}_{\text{cloth, vel}}$ with a Runge-Kutta integrator of order 4 over a training horizon $H_{\mathcal{D}} = 8$ timesteps. We train the RO-HNN with Riemannian Adam (Becigneul & Ganea, 2019) until convergence at 3000 epochs with a learning rate of $5 \times 10^{-2}$ for the AE parameters, $2 \times 10^{-4}$ for the HNN parameters, and a regularization $\gamma = 2 \times 10^{-5}$ for 3000 epochs.

In our comparison, we also consider a black-box version of the RO-HNN, hereinafter referred to as black-box RO-LNN, where the latent LNN is encoded as a single black-box network $\mathcal{L}_{\theta}$ representing the Lagrangian function, which we model via a single fully-connected MLP of 2 hidden layers with a width of 256 neurons. The network weights are initialized by sampling from a Xavier normal distribution with gain $\sqrt{2}$, and the bias vector entries are initialized to 1. The black-box RO-LNN is trained with the same parameters as the original RO-LNN, except for the learning rate for the LNN parameters, which is set as of $2 \times 10^{-3}$.

# H  ADDITIONAL EXPERIMENTAL RESULTS

This section presents additional results, complementing those presented in Sec. 4.

## H.1  COUPLED PENDULUM OF SECTION 4.1

This section presents additional results on learning the Hamiltonian dynamics of a 15-DoF coupled pendulum.

**Learning high-dimensional dynamics.** Fig. 7 complements Fig. 2 by depicting the predicted long-term (5s) positions and momenta. For the ease of visualization, we change the prediction coordinates and plot the first 3-DoF corresponding to the latent pendulum. We observe that RO-HNN leads to accurate long-term predictions similar to those of the 3-DoF HNN. Unlike the 15-dimensional full-order HNN, the HNKO yields stable, but inaccurate long-term predictions, exhibiting significantly higher deviation from the ground truth trajectory than the predictions of the RO-HNN.

Fig. 8 depicts the original 15-DoF trajectories projected into the 3-DoF latent space learned by the RO-HNN, along with its latent dynamic predictions over the full prediction horizon of 5s. We observe that the latent predictions are accurate and match the projected original trajectories. As expected, the learned latent space does not coincide with the phase-space of the original pendulum due to the nonlinear dimensionality reduction conducted via the AE, but displays comparable frequencies and amplitudes.

**AE architecture.** Figure 9 accompanies and validates the results of Table 2 by displaying the median and quartiles of the prediction errors obtained by different symplectic dimensionality reduction methods in the intrusive MOR scenario of Sec. 4.1.

**Latent HNN architecture.** Here we further evaluate the impact of HNN architecture. We compare the performance of our geometric HNN to learn the low-dimensional dynamics of the latent 3-DoF pendulum against *(1)* a non-geometric variant that parametrizes the inverse mass-inertia matrix via a Cholesky network, and *(2)* two HNNs encoded as a single black-box network $\mathcal{H}_{\theta}$, where we consider two MLPs of 64- and 256-neurons width. Compared to Sec. 4.1, we consider a doubled amount of training datapoints with 6000 random samples. As shown in Fig. 10-*left*, the geometric HNN still achieves the lowest reconstruction error, with differences compared to the black-box HNN increased

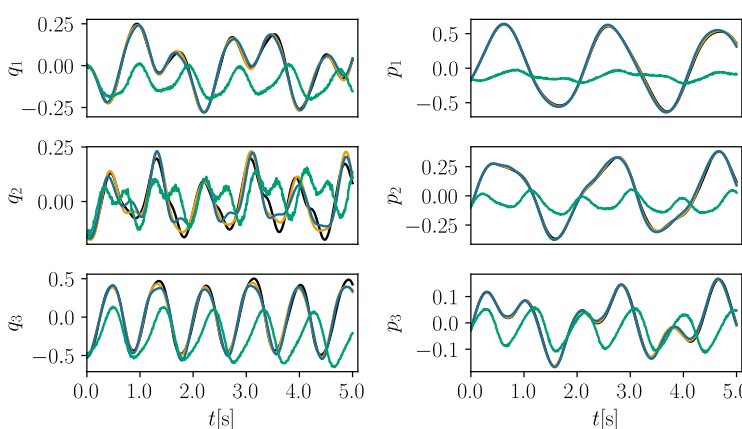

Figure 7: Reconstructed trajectories of the RO-HNN (—), 3-DoF HNN (—), and HNKO (—) compared to ground truth (—). The 15-DoF HNN leads to unstable long-term predictions and is not depicted.

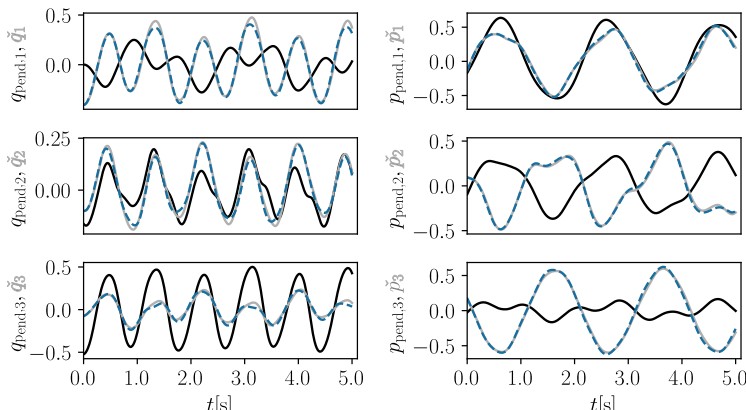

Figure 8: Trajectories of the original 15-DoF pendulum projected into the latent space of the RO-HNN (—), and corresponding dynamic predictions obtained via the latent HNN (⋯⋯). As expected, they does not coincide directly with the trajectories of the underlying 3-DoF pendulum representation (—).

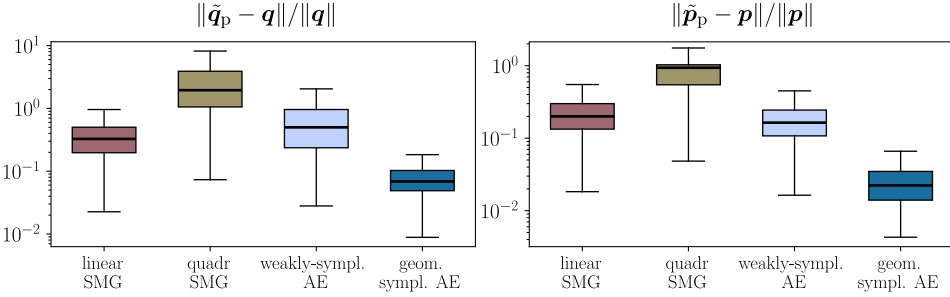

Figure 9: Prediction errors (↓) of intrusive symplectic dimensionality reduction approaches over 10 test pendulum trajectories.

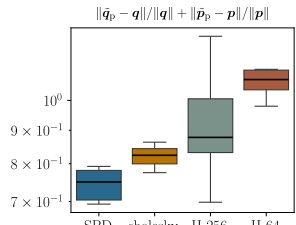 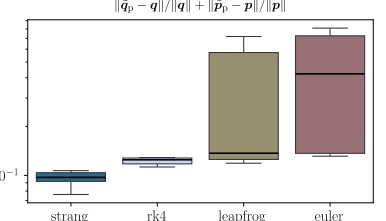 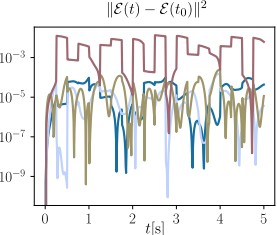

Figure 10: *Left*: Ablation of the latent HNN architecture on a doubled training set size $|\mathcal{D}| = 6000$ compared to Fig. 10-*left*. *Middle, right*: Ablation of the latent integrator of the geometric RO-HNN at $|\mathcal{D}| = 3000$ for learning the dynamics of a 15-DoF pendulum. Errors are obtained via short-term prediction horizons $H\Delta t = 0.25$ s.

Table 5: Mean and standard deviation of prediction errors ($\downarrow$) over $N = 10$ noise-free test pendulum trajectories, comparing the performances of the RO-HNNs and HNKOs trained on noisy observations.

| | $H\Delta t$ (s) | $c_{\text{noise}} = 0$ | | $c_{\text{noise}} = 5\%$ | | $c_{\text{noise}} = 10\%$ | |
| --- | --- | --- | --- | --- | --- | --- | --- |
| | | RO-HNN | HNKO | RO-HNN | HNKO | RO-HNN | HNKO |
| $\frac{\|\tilde{q}_p - q\|}{\|q\|}$ | 0.25 | $(1.66 \pm 1.38) \times 10^{-1}$ | $(5.64 \pm 4.41) \times 10^{-1}$ | $(2.44 \pm 1.94) \times 10^{-1}$ | $(9.76 \pm 10.48) \times 10^{-1}$ | $(3.23 \pm 2.86) \times 10^{-1}$ | $(1.02 \pm 0.78) \times 10^{0}$ |
| | 5 | $(7.08 \pm 7.56) \times 10^{-1}$ | $(1.32 \pm 0.94) \times 10^{0}$ | $(8.40 \pm 8.17) \times 10^{-1}$ | $(1.98 \pm 1.84) \times 10^{0}$ | $(9.14 \pm 10.02) \times 10^{-1}$ | $(4.06 \pm 3.47) \times 10^{0}$ |
| $\frac{\|\tilde{p}_p - p\|}{\|p\|}$ | 0.25 | $(5.33 \pm 5.23) \times 10^{-2}$ | $(5.93 \pm 10.73) \times 10^{-1}$ | $(7.02 \pm 7.17) \times 10^{-2}$ | $(4.56 \pm 8.34) \times 10^{-1}$ | $(9.64 \pm 10.30) \times 10^{-2}$ | $(9.60 \pm 17.23) \times 10^{-1}$ |
| | 5 | $(1.98 \pm 2.67) \times 10^{-1}$ | $(1.23 \pm 2.08) \times 10^{0}$ | $(3.34 \pm 4.97) \times 10^{-1}$ | $(1.62 \pm 2.97) \times 10^{0}$ | $(3.61 \pm 4.94) \times 10^{-1}$ | $(3.13 \pm 5.58) \times 10^{0}$ |

compared to the smaller dataset of Fig. 3-*left*. This showcases the importance of considering both the quadratic energy structure of mechanical systems, and the geometry of their mass-inertia matrices, for both enhanced performance and data efficiency.

**Latent integrator.** We compare the Strang symplectic integrator against *(1)* a symplectic leapfrog integrator that disregards that the Hamiltonian is non-separable, *(2)* a Runge-Kutta integrator of order 4 that overlooks its symplectic structure, and *(3)* an explicit Euler integrator that also overlooks the symplectic structure. Compared to Sec. 4.1 (see Fig. 3-middle,right), we consider shorter prediction horizons, feeding the model with ground truth initial conditions every $H\Delta t = 0.25$ s, since the explicit Euler integrator did not lead to stable long-term predictions for $H\Delta t = 5$ s. Figs. 10-*middle*, *right* show that the networks trained via the Strang-symplectic integrator achieve the lowest reconstruction error and conserves energy best during integration, showcasing the importance of considering the symplectic structure of the system during numerical integration for stable predictions on short- and long-term time horizons.

**Training under noisy observations.** To assess the robustness of the RO-HNNs, we evaluate its performance under noisy observations and compare it against the HNKOs baseline, which is reported to be robust to noise in high-dimensional systems.

We generate noisy training data $\{q_i + \epsilon_{q,i}, p_i + \epsilon_{p,i}\}$ corrupted with zero-mean Gaussian noise $\epsilon_{q,i} \sim \mathcal{N}(0, \sigma_q^2 I)$ and $\epsilon_{p,i} \sim \mathcal{N}(0, \sigma_p^2 I)$. The noise level $c_{\text{noise}}$ is determines the standard deviations, which is also proportional to the maximum entry of the position and momentum, i.e., $\sigma_q = c_{\text{noise}} \max_{j,k} |q_{j,k}|$ and $\sigma_p = c_{\text{noise}} \max_{j,k} |p_{j,k}|$.

Table 5 reports the prediction errors on a testing dataset of 10 noise-free trajectories over time horizons $H\Delta t = \{0.25, 5\}$s for three noise level $c_{\text{noise}} = \{0, 0.05, 0.1\}$. The noiseless results are repeated from Table 1 for completeness. As expected, the performance of both models decreases with increasing noise magnitude. In each scenario, the RO-HNN outperforms the HNKO baseline. Note that the RO-HNN trained at $c_{\text{noise}} = 0.1$ outperforms the HNKO trained without noise, demonstrating the enhanced accuracy and robustness of the RO-HNN to noisy observations.

## H.2 PARTICLE VORTEX (90-DOF) OF SECTION 4.2

This section presents additional results on learning the Hamiltonian dynamics of a 90-DoF particle vortex.

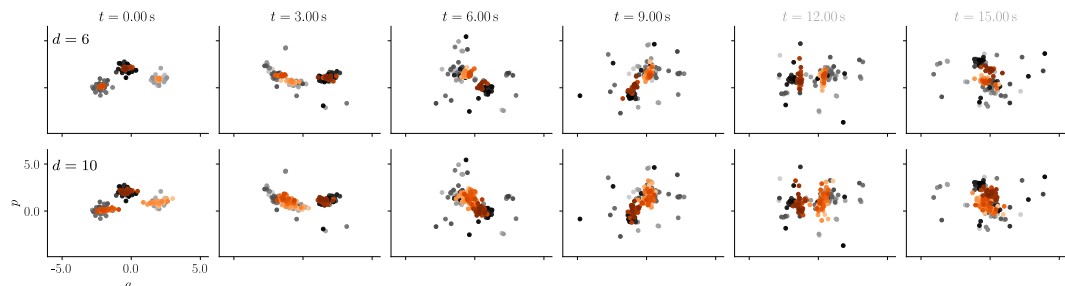

Figure 11: Predicted (•, •, •) vs ground truth (•, •, •) positions of the particle vortex. The dynamics are learned with RO-HNN with $d = 6$ and $d = 10$. Times beyond 10s are out of the training data distribution.

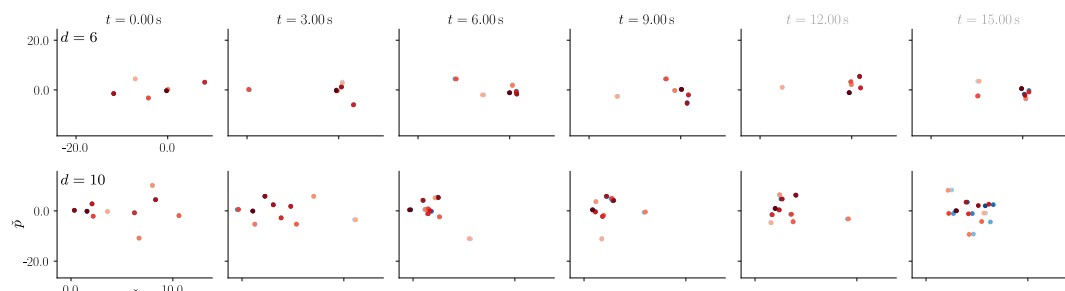

Figure 12: Predicted (•, •, •) vs ground truth (•, •, •) reduced positions of the particle vortex in the latent space of the RO-HNN with $d = 6$ and $d = 10$. Times beyond 10s are out of the training data distribution.

Fig. 11 depicts the predicted positions and momenta of the particles along with the ground truth in the high-dimensional state space for RO-HNNs with latent dimension $d = \{6, 10\}$. Fig. 12 depicts the predicted positions and momenta of the particles in the reduced phase space of the AE along with the projected ground truth. We observe that both models accurately predict the particle vortex dynamics, with the $d = 6$-dimensional model slightly outperforming the 10-dimensional one (see also Table 3 and Fig. 13). This shows that the choice of latent dimension is a trade off between the latent space expressivity and the limitations of HNNs in higher dimensions. In general, we observed that errors initially decrease as the latent dimension increases, suggesting that higher-dimensional latent spaces better capture the original high-dimensional dynamics. The errors then increase beyond a certain latent dimension, indicating that the latent HNN becomes harder to train.

Fig. 13 accompanies and validates the results of Table 3 by displaying the median and quartiles of the prediction errors obtained by different latent HNNs.

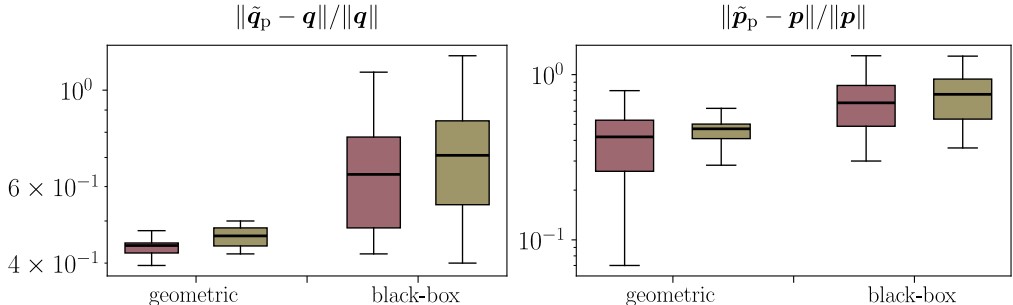

Figure 13: RO-HNN prediction errors (↓) for black-box and geometric latent HNNs with latent dimensions $d = 6$ (■) and $d = 10$ (■) over 10 particle vortex trajectories.

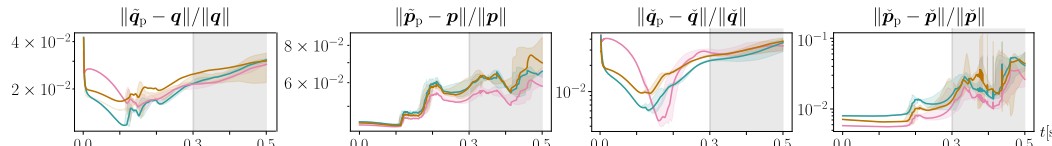

Figure 14: Median and quartiles of the latent and reconstructed prediction errors of 10-dimensional RO-HNNs with latent dissipation matrix parametrized with a SPD network (—), a Cholesky network (—), and ground truth values (—). The gray-shaded area indicates the time horizon beyond the training data support.

## H.3 CLOTH (600-DoF) OF SECTION 4.3

This section presents additional results on learning the Hamiltonian dynamics of a 600-DoF thin cloth falling on a sphere.

**Learning high-dimensional dynamics with dissipation.** Fig. 15 complements Fig. 5 by depicting the predicted cloth configurations for the RO-HNNs with latent dimensions $d = \{6, 10\}$ for a horizon $H\Delta t = 0.5$ s. We observe that both RO-HNNs accurately predict the high-dimensional dissipative dynamics of the cloth, generalizing beyond the data support ($t > 0.3$s). As also shown in Table 4, the $10-$dimensional model slightly outperforms the 6-dimensional one, modeling more details of the cloth, as shown in Fig. 15.

Fig. 14 accompanies Table 4 by visualizing the median and quartiles of the RO-HNN ($d = 10$) reconstructed and latent prediction errors over time for two different parametrization of the reduced dissipation matrix $\check{D}$. Both dissipative RO-HNN perform similarly to the conservative RO-HNN, showing that the RO-HNN can successfully predict dissipative dynamics in a stable manner, including beyond the training time horizon.

Fig. 16 shows the predictions of the RO-HNNs with different parametrizations of the dissipation matrix $\check{D}$ for selected dimensions of a test trajectory. This shows that the dissipative RO-HNNs successfully learn the dissipation forces, achieving similar prediction errors as the conservative models Fig. 17 displays the predicted latent energy to be compared with the ground-truth energy projected in the symplectic latent space. Overall, our results demonstrate the ability of the RO-HNN to infer long-term predictions of dissipative systems.

**Latent dimension and training ablation.** We compare the performance of our dissipative RO-HNN across several latent dimensions $d = \{2, 6, 10\}$ with jointly-trained geometrically-constrained symplectic AE and latent geometric HNN against sequentially-trained architectures. Specifically, we consider *(1)* linear and *(2)* quadratic symplectic manifold Galerkin (SMG) projections (Peng & Mohseni, 2016; Sharma et al., 2023), *(3)* a weakly-symplectic AE trained jointly with a latent geometric HNN, and *(4)* a RO-HNN with pretrained geometrically-constrained AE. Fig. 18 shows that our jointly-trained RO-HNN significantly outperforms all baselines for all dimensions, leading to reduced relative reconstruction, latent prediction, and reconstructed prediction errors. This showcases *(1)* the higher expressivity of the AEs compared to linear and quadratic projection methods, *(2)* the importance of structurally-embedding the symplecticity condition, unlike the weakly-symplectic AE, and *(3)* the importance of joint training, allowing the RO-HNN to jointly learn a symplectic submanifold and the associated dynamics.

Finally, we compare the performance of the dissipative RO-HNN against *(1)* a conservative RO-HNN, where the dissipation forces $\tau_d$ are not learned but provided as ground truth in the external input $\tau = \tau_c + \tau_d$, and *(2)* a dissipative RO-HNN where the dissipation matrix is parametrized via Cholesky decomposition for latent dimensions $d = \{2, 6, 10\}$. The mass-inertia matrix is parametrized via SPD networks in all cases. Fig. 19 shows the obtained latent prediction and reconstructed prediction errors. Both dissipative HNNs achieve errors close to the conservative HNN where the ground truth dissipative forces are provided, with the geometric HNN slightly outperforming its Cholesky counterpart. However, the effect is less pronounced as when learning the inverse mass-inertia matrix, which we attribute to the reduced influence of damping compared to inertia in the overall dynamics.

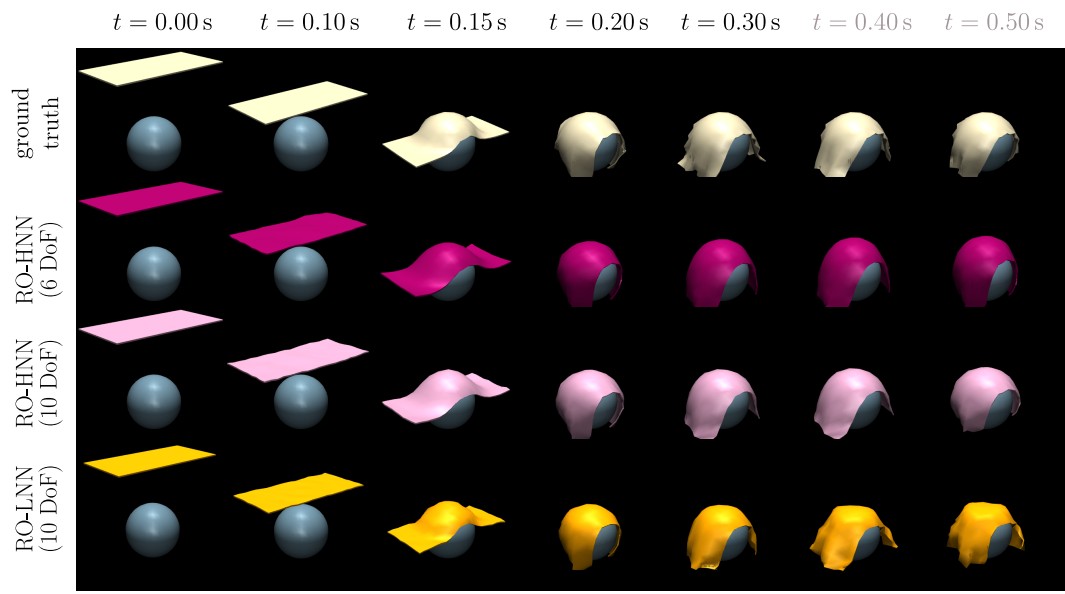

Figure 15: Predicted positions of the damped cloth with RO-HNNs with $d = \{6, 10\}$ and RO-LNN with $d = 10$ for a $625\times$ longer horizon than during training. Times beyond 0.3s are out of the training data distribution.

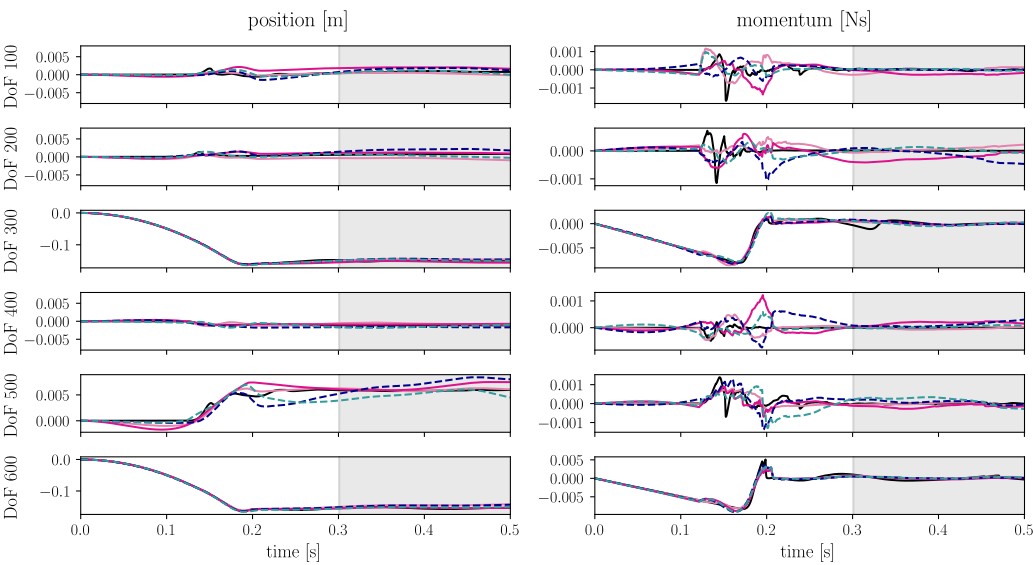

Figure 16: Predicted cloth positions and momenta for 6-dimensional RO-HNNs with latent dissipation matrix parametrized with a SPD network (——), a Cholesky network (——), and ground truth values (····), and 10-dimensional RO-HNNs with latent dissipation matrix parametrized with a SPD network (——), a Cholesky network (——), and ground truth values (····). The grey-shaded areas indicates interval beyond the data support.

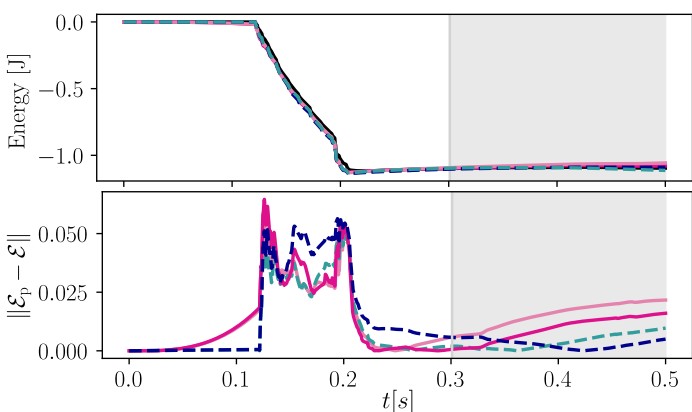

Figure 17: *Top*: Ground truth (—) and predicted latent energies for 6-dimensional RO-HNNs with latent dissipation matrix parametrized with a SPD network (—), a Cholesky network (—), and ground truth values (⋯⋯), and 10-dimensional RO-HNNs with latent dissipation matrix parametrized with a SPD network (—), a Cholesky network (—), and ground truth values (⋯⋯). *Bottom*: Energy errors for the same models. The grey-shaded areas indicate intervals beyond the data support, for which the ground truth is extrapolated from the last observation.

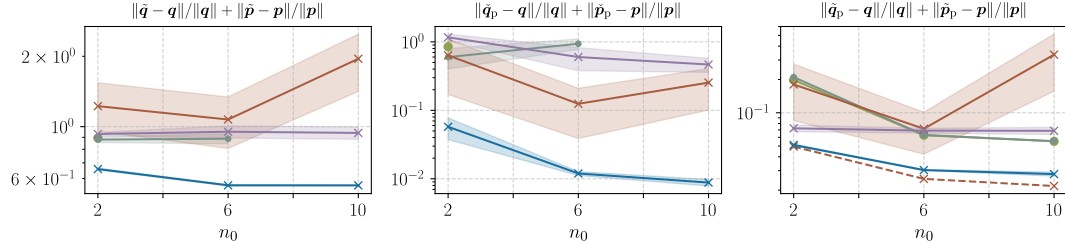

Figure 18: Mean and standard deviation of the relative reconstruction (*left*), latent prediction (*middle*), and reconstructed prediction (*right*) errors over 10 cloth trajectories with $H\Delta t = 0.0025\,\text{s}$. Our RO-HNN with geometrically-constrained symplectic AE ( ✶ ) is compared against linear SMG reduction (–●–), quadratic SMG reduction (–●–), a weakly symplectic AE ( ✶ ), and a sequentially-trained RO-HNN with pretrained geometrically-constrained symplectic AE ( ✶ ). The pretrained AE ( ·✶· ) is depicted for completeness. Notice that the linear SMG and quadratic SMG projections led to diverging dynamics for $d > 2$ and $d > 6$, respectively, for which results are not depicted.

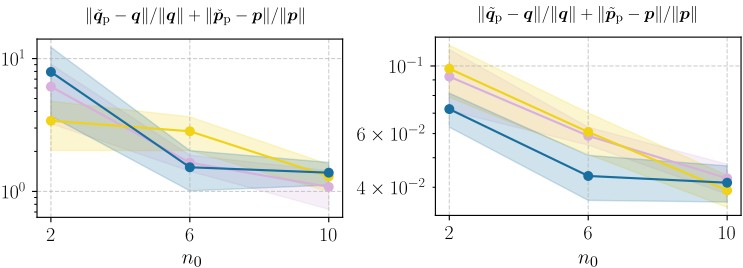

Figure 19: Mean and standard deviation of the latent prediction (*left*) and reconstructed prediction (*right*) errors for different parametrization of the latent dissipation matrix $\check{D}$ over 10 test cloth trajectories. We compare our SPD network (—) against a Cholesky network (—), and the ground truth parametrization (—).

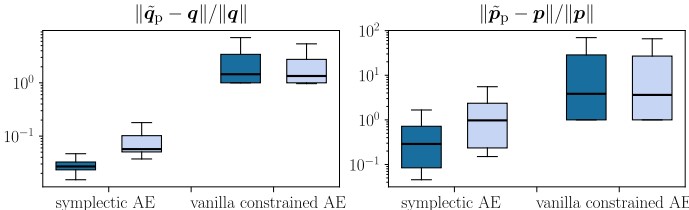

Figure 20: Prediction errors (↓) of RO-HNNs with geometrically-constrained symplectic or vanilla constrained AE with Strang symplectic integrator (■) or Runge-Kutta integrator of order 4 (■) over 10 testing cloth trajectories.

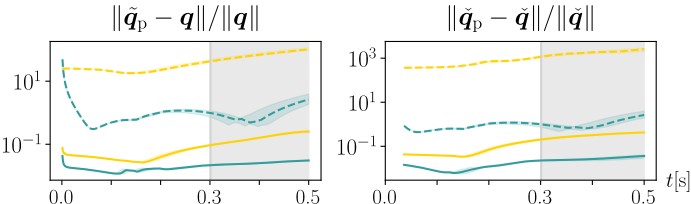

Figure 21: Median and quartiles of the latent and reconstructed prediction errors of 10-dimensional geometric RO-HNN (——), geometric RO-LNN (——), a RO-HNN with latent black-box HNN (····), and a RO-LNN with latent black-box LNN (····).

**Ablation of the symplectic architecture to learn dissipative dynamics.** As discussed in Sec. 3.2, the dissipative dynamics do not preserve a symplectic structure. Our proposed RO-HNN features two main symplecticity-preserving components, namely the geometrically-constrained symplectic AE and the Strang-symplectic integrator, which we ablate here. We consider two variations of each components, i.e., (1) our geometrically-constrained symplectic AE and a vanilla projection-constrained AE, and (2) the Strang-symplectic integrator and a Runge-Kutta integrator of order 4. Throughout all experiments, we set the latent dimension $d = 10$. Fig. 20 shows the obtained reconstructed prediction errors. We observe that our geometrically-constrained symplectic AE leads to significantly lower median errors than vanilla projection-constrained AE independently of the choice of integrator, showcasing the benefit of preserving the structure of FOM vector field in the ROM (see Proposition 3). Moreover, despite the dissipative structure, the RO-HNN obtained with the Strang-symplectic integrator outperforms the non-symplectic Runge-Kutta integrator. We hypothesize that this is due to the fact that the evolution of this dissipative system is mostly governed by its Hamiltonian function, especially over the short timesteps taken by the integrators.

**Comparison against RO-LNN.** The last row of Fig. 15 depicts the predicted cloth configuration for the RO-LNN with latent dimension $d = 10$. Moreover, Fig. 21 compares the latent prediction and reconstructed prediction errors of the geometric and black-box RO-HNN and RO-LNN over time.

Our results show that the geometric RO-HNN outperforms the RO-LNN, leading to more accurate predictions. It is worth noting that the RO-HNN leads to increased performances despite that it also learns the dissipation forces via the latent damping matrix, which are instead provided as ground truth to the RO-LNN. Moreover, Fig. 21 shows that the geometric RO-HNN and RO-LNN featuring geometric latent HNN and LNN outperform their black-box counterparts, showcasing the importance of considering the quadratic energy structure of mechanical systems in both network types.

We hypothesize that the improved accuracy of the RO-HNN compared to the RO-LNN can be attributed to *(1)* the first-order dynamic formulation stemming from Hamiltonian mechanics, which is easier to learn and optimize than the second-order Lagrangian formulation, *(2)* the Strang-symplectic integrator which is specifically designed for Hamiltonian systems, in contrast to the Runge Kutta integrators typically used in the case of continuous-time Lagrangians. This aligns with the discussions in (Liu et al., 2024a), which showed that, by using position and momentum observations, HNNs learn mass-inertia matrices that are close to the physical solutions, while LNNs only learn one of the solutions satisfying the Euler-Lagrange equations.

Table 6: Evaluation wall clock times for different ODE-solvers on analytic FOM compared to RO-HNN. Runtimes are averaged over 10 forward passes and given in s.

| | pendulum ($n = 15, d = 3$) | | | cloth ($n = 600, d = 10$) | | |
| | $\Delta t = 10^{-2}$s | | $\Delta t = 10^{-1}$s | $\Delta t = 10^{-4}$s | | $\Delta t = 10^{-3}$s |
| | Strang | Euler | Strang | Strang | Euler | Strang |
|---|---|---|---|---|---|---|
| FOM, $n-$DoF | 1.29 | 0.75 | 0.18 | 255.24 | 77.62 | 109.74 |
| RO-HNN, $d-$DoF | 0.79 | — | — | 16.01 | — | — |

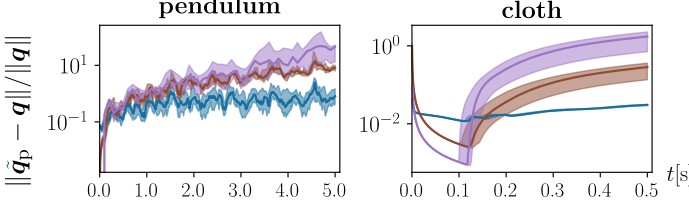

Figure 22: Position prediction errors introduced by various simulation speedup methods: Euler-forward integration (—), Strang integration (—) at larger stepsize, and RO-HNN (—).

## H.4 RUNTIMES

This section compares the runtimes of different models. All experiments were performed locally on a MacBook Pro with M3 CPU.

**Speedup of simulation via the RO-HNN.** We aim at providing an idea of the computational effort of the RO-HNN compared to the evaluation of the FOMs. To do so, we symbolically derive the Hamiltonian equations of motion with known physical quantities of the 15-DoF coupled pendulum of Sec. 4.1 and obtain the equations of motion of the 600-DoF cloth from Sec. 4.3 from Mujoco. We compare the wall-clock time of the evaluation of these two FOMs against the respective RO-HNN averaged over 10 single trajectory roll-outs from the same initial conditions as in the testing dataset. We consider two different integrators, namely an Euler forward and the Strang symplectic integrator, and different step size $\Delta t$ on a time-horizon of $H\Delta t = 5$s for the pendulum and $H\Delta t = 0.5$s for the cloth. The evaluation times for the coupled pendulum and the cloth are given in Table 6, with the corresponding relative position prediction errors depicted in Fig. 22. We observe a prominent reduction of the evaluation time for the RO-HNNs compared to the FOMs. This reduction is exacerbated for higher-dimensional systems, e.g., the cloth, where the evaluation time of the RO-HNN remains significantly lower than that of the FOM evaluated with the computationally-cheaper forward Euler integrator or increased step size. Moreover, as shown in Fig. 22, the RO-HNNs exhibit lower prediction errors than their FOM counterparts in addition to a reduced computational complexity. This showcases that the RO-HNN not only enable accurate learning of unknown high-dimensional Hamiltonian dynamics, but also the computationally-efficient and accurate evaluation of known systems via surrogate dynamics.

**Comparison of runtimes of HNNs and RO-HNNs.** Table 7 reports the averaged runtimes for the forward pass of the differently-sized network architectures considered in Sec. 4.1. The reported times correspond to the wall clock time of one forward pass of a batch of 10 initial conditions, predicted over $H = 10$ timesteps with the Strang-symplectic integrator. We observe that the RO-HNN speeds up the forward dynamics computation compared to the HNN, highlighting the computational advantages of ROMs compared to FOMs. Moreover, the black-box HNN is computationally more efficient than the geometric HNN at the expense of prediction accuracy.

Table 7: Evaluation wall clock times for different network architectures on the 15-DoF pendulum. Runtimes are averaged over 10 forward passes and given in ms.

| 15-DoF | | 3-DoF | |
| Geometric HNN | Geometric RO-HNN | Geometric HNN | Black-box HNN |
|---|---|---|---|
| 100.25 | 26.34 | 18.10 | 8.04 |

Table 8: Evaluation wall clock times of different variants for the AE on the 600-DoF cloth dataset. Runtimes are averaged over 10 forward passes and given in ms.

| Position-level AE $\varphi_{\mathcal{Q}} \circ \rho_{\mathcal{Q}}(\boldsymbol{q})$ | Geometric AE with analytic lift $\varphi \circ \rho(\boldsymbol{q}, \boldsymbol{p})$ | Geometric AE with `autodiff.vjp` lift $\varphi \circ \rho(\boldsymbol{q}, \boldsymbol{p})$ | AE with naive lift $\varphi_{\mathcal{Q}} \circ \rho_{\mathcal{Q}}(\boldsymbol{q}, \boldsymbol{p})$ |
|---|---|---|---|
| 16.92 | 20.93 | 54.24 | 19.03 |

**Runtimes of the lifted AE.** Table 8 reports the average wall-clock times for one forward pass of a batch of 100 states for several projection scenarios on the 600-DoF cloth dataset. We consider reduction to a latent space of dimension $d = 10$ via 4 layers of size $n_l = \{32, 64, 128, 600\}$. We consider (1) a position-level constrained AE, where only position projections $\tilde{\boldsymbol{q}} = \varphi_{\mathcal{Q}} \circ \rho_{\mathcal{Q}}(\boldsymbol{q})$ are computed via the encoder and decoder layers, (2) a geometric symplectic AE, whose lifted mappings (8) are computed analytically as a composition of layer derivatives (see App. D) to project both positions and momenta via $(\tilde{\boldsymbol{q}}, \tilde{\boldsymbol{p}}) = \varphi \circ \rho(\boldsymbol{q}, \boldsymbol{p})$, (3) a geometric symplectic AE whose lifted maps are computed via automatic differentiation using Pytorch's `autograd.vjp` function, and (4) a naive constrained AE that jointly projects the positions and momenta in $2d = 20$-dimensional latent space via doubled layer dimensions, i.e., $n_l = \{64, 128, 256, 1200\}$. The first two variants were evaluated under `torch.no_grad()`, reflecting a realistic scenario for evaluation of forward dynamics in the RO-HNN.

The runtimes reported in Table 8 show that the analytic computation of the lifted mappings is significantly faster than the automatic-differentiation-based implementation. This is expected, as our analytic implementation avoids the construction of a backward graph. It is worth emphasizing that the geometric AE with analytic lifts requires significantly less than twice the runtime of the position-level constrained AE. Therefore, using a single cotangent-lifted AE that jointly projects positions and momenta is computationally more advantageous than training two separate AEs for separate projections.

