# OpenReview forum: "Learning Hamiltonian Dynamics at Scale:  A Differential-Geometric Approach"
_ICLR.cc/2026/Conference — Submitted to ICLR 2026_

### Official Review · Reviewer_fRyk · 2025-10-26

**Soundness:** 3
**Presentation:** 2
**Contribution:** 2
**Rating:** 4
**Confidence:** 4

**Summary:**

This paper proposes a dimension reduction that is faithful to symplectic geometry for Hamiltonian Neural Networks. Many methods have been proposed to efficiently learn high-dimensional dynamical systems by projecting them into a low-dimensional latent space with an autoencoder and then training HNNs or related models in that space (the original HNNs also do this). However, these reductions have not been guaranteed to be symplectically valid. The paper resolves this by proposing an autoencoder that is a symplectomorphism.

**Strengths:**

1. The proposed method prepares an autoencoder $(\phi_Q, \rho_Q)$ for the configuration, and then designs an autoencoder for the momentum using the pullback via this mapping. If the variable were velocity, one could use a pushforward, but because it is momentum, the pullback of the inverse map is required. This seems to be the key idea of the work.

2. The proposed framework incorporates many designs that respect intrinsic geometric structures, such as using an SPD network for the mass matrix and the dissipation matrix, and adopting a symplectic integrator for non-separable systems.

**Weaknesses:**

1. At line 342, observations are listed as position, velocity, and force, while the model is defined using momentum. Is this a typo for momentum? At line 366, the observations include momentum.

2. If the observations indeed include momentum, how would this be measured? For a point mass, one could multiply by mass, but in general dynamical systems, this is not sufficient, and a Legendre transform is required. Moreover, under the situation that mass can be observed, would the system still be meaningfully unknown? I am unsure about realistic scenarios where this framework would be useful. Speeding up simulation, as in Sec. 4.3, is conceivable, but then, the paper needs to clarify how much speedup is achieved and how large error is introduced, and it should compare against dimension reduction methods for known governing equations. The same goes for the force.

3. Similarly, the most direct comparison method seems to be Friedl et al. (2025), but this comparison is missing.

4. SPD network, constrained autoencoder, and the Strang-symplectic integrator are all existing components combined. The novelty appears somewhat limited. The definition of the symplectic autoencoder is very natural from the viewpoint of symplectic geometry.

5. Both the autoencoder design and the numerical integration place strong emphasis on symplectic geometry. However, the actual latent-space dynamics is something like a port-Hamiltonian system, with dissipation and external forces. While the experiments demonstrate the usefulness of the Strang-symplectic integrator, the discussion could be strengthened.

6. Prediction errors accumulate, so as the prediction time grows, errors can become extreme. A single divergent trajectory can drastically worsen the score. In fact, the standard deviation is about the same size as, or larger than, the average error. A three-times difference could be overturned. It would be better to include plots of error growth or adopt robust metrics.

Minor comments:

7. There is some confusion in the Introduction. PINNs use governing equations to define losses. Unless extended in combination with methods like SINDy, they are not intended for learning unknown dynamical systems, and their objective differs from HNNs and Neural Operators. Since this work is an extension of HNNs, discussing it separately from Neural Operators and PINNs might reduce confusion.

8. In Eq. (8), the left-hand side takes arguments, so the right-hand side should also take arguments.

9. Line 278

> Existing HNNs enforce the symmetric positive-definiteness of the inverse mass-inertia matrix via a Euclidean network encoding its Cholesky decomposition $L$

At least the original HNNs did not implement it that way. Perhaps this refers to Dissipative SymODEN? Additional explanation would help.

10. From the perspective of learning non-canonical Hamiltonian systems and dimension reduction with neural networks, in addition to neural symplectic forms, I recommend citing the following papers.

Jin et al., "Learning Poisson systems and trajectories of autonomous systems via Poisson neural networks," IEEE Transactions on Neural Networks and Learning Systems, 2023.

Sipka et al., "Direct Poisson neural networks: Learning non-symplectic mechanical systems," Journal of Physics A: Mathematical and Theoretical, 2023.

**Questions:**

1. The proposed autoencoder is implementable with vector-Jacobian products, so the computational overhead (compared to simple autoencoders) appears to be only a few times larger. Please clarify the computational overhead.

2. At line 306, is $\rho$ a typo for $\varphi$? Notations are also hard to read. Do subscripts $p$ indicate predicted quantities, while symbols without $p$ indicate encoder outputs?

---

> ### Author Response · Authors · 2025-11-21
> **Response to Reviewer fRyk (1)**
>
> We would like to thank the reviewer for their detailed feedback! We address your concerns and suggestions below.
>
> ## Summary
>
> > This paper proposes a dimension reduction that is faithful to symplectic geometry for Hamiltonian Neural Networks. Many methods have been proposed to efficiently learn high-dimensional dynamical systems by projecting them into a low-dimensional latent space with an autoencoder and then training HNNs or related models in that space (the original HNNs also do this). However, these reductions have not been guaranteed to be symplectically valid. The paper resolves this by proposing an autoencoder that is a symplectomorphism.
> >
>
> In previous HNNs paired with AEs, despite that the observations (i.e., the images) were high-dimensional, the considered physical systems were low-dimensional (i.e., less than 10 DoFs). **The role of the AE was therefore to extract the states of the low-dimensional system from images**, and the dynamics were learned based on these low-dimensional states.
>
> We would like to point out that our approach differs from previous HNNs that use AEs **not only because our AE is a symplectomorphism** but also, and importantly, in that it does not consider high-dimensional observations (e.g., images) of low-dimensional physical systems but **systems with high-dimensional state spaces,** a.k.a high-dimensional dynamics. To do so, our model learns a low-dimensional symplectic submanifold and the dynamics of the associated Hamiltonian function via a novel symplectic AE ("non-intrusive MOR").
>
>
> ## Weaknesses
>
> > 1.At line 342, observations are listed as position, velocity, and force, while the model is defined using momentum. Is this a typo for momentum? At line 366, the observations include momentum.
>
> Thank you for the alertness! This was indeed a typo, which we now fixed by writing the observations as $( \mathbf q_i(\mathcal I_{i}), \mathbf p_i(\mathcal I_{i}), \mathbf \tau_i(\mathcal I_{I}) )$.

---

> ### Author Response · Authors · 2025-11-21
> **Response to Reviewer fRyk (2)**
>
> > 2. If the observations indeed include momentum, how would this be measured? For a point mass, one could multiply by mass, but in general dynamical systems, this is not sufficient, and a Legendre transform is required. Moreover, under the situation that mass can be observed, would the system still be meaningfully unknown? I am unsure about realistic scenarios where this framework would be useful. Speeding up simulation, as in Sec. 4.3, is conceivable, but then, the paper needs to clarify how much speedup is achieved and how large error is introduced, and it should compare against dimension reduction methods for known governing equations. The same goes for the force.
> >
>
> It is important to emphasize that the canonical momentum does not necessarily correspond to the mechanical momentum. Therefore, **for many canonical Hamiltonian system, the canonical momentum can be observed without knowing the mass**. For instance, the canonical momentum is directly observed in the particle vortex experiment of Section 4, where the canonical phase-space coordinates correspond to the location of the particle in the $(x_1,x_2)$-plane, i.e., $(q,p)=(x_1, x_2)$.  Canonical momentum can also be observed in other scenarios, e.g., for quantum mechanical systems [1] such as quantum harmonic oscillators, and optical systems [2].
>
> In our paper, **we follow the assumption commonly adopted in the HNN literature** [3-8], and assume access to sets of observations $( \mathbf q_{i}, \mathbf p_{I} )_{i=1}^{N}$ that include the canonical momentum to train our RO-HNN. It is also worth emphasizing that classical benchmarks provide canonical coordinates (i.e., position and momentum) [9].
>
> In cases where the canonical momentum cannot be measured, e.g., for mechanical systems, one could assume access to an approximate estimate of $\mathbf{M}$, e.g., from a manufacturing process or physical simulator, of the mass-inertia matrix that allows the computation of approximation of the momentum, which is then used as observations to learn a refined, accurate model including the potential via the HNN. In other cases, the momentum is set as equal to the velocity, and the HNN implicitly learns the mapping between the velocity and the canonical momentum [8]. Finally, another strategy consists in learning the canonical coordinates from images via an auxiliary AE, whose latent space is identified with the phase-space of the system [3,8].
>
> We would also like to point out that **the proposed geometrically-constrained symplectic AE introduced in Section 3.1 can be utilized in classical intrusive model-order reduction scenarios where the high-dimensional dynamical parameters are assumed to be fully known**. In such scenarios, our geometrically-constrained symplectic AE can be used to obtain a computationally cheaper yet accurate low-dimensional surrogate model of the known high-dimensional system, and the momentum can be computed via a Legendre transform. We consider such a scenario in Table 2 to evaluate our geometrically-constrained AE compared to linear and quadratic symplectic projections and to a weakly-symplectic AE, showcasing that our geometrically-constrained AE leads to the lowest prediction error.
>
> Concerning the computational speed-up, we would like to emphasize that **we are interested in scenarios where the governing equations are unknown, and should be inferred from data,** e.g., by using a HNN. In such a scenario, there is no known model to reduce in the first place and traditional dimensionality reductions methods are not applicable. In other words, **our primary objective is to predict trajectories of the system, not in reducing the computational cost of predicting them.** Nevertheless, we aim at providing an idea of the computational effort of the RO-HNN compared to the evaluation of the full-order known governing equations. To do so, we symbolically derive the Hamiltonian equations of motion with known physical quantities of the 15-DoF coupled pendulum of Sec. 4.1 and obtain the equations of motion of the 600-DoF cloth from Sec. 4.3 from Mujoco. We compare the wall-clock
> time of evaluation of these two FOMs against the respective RO-HNN averaged over 10 runs. We obtain evaluation times:
>
> - Pendulum FOM: 1.29s vs pendulum RO-HNN: **0.79s**
> - Cloth FOM: 14.58s vs cloth RO-HNN: **1.60s**
>
> We added this comparison in Appendix H.4 under “Speedup of simulation via the RO-HNN”.

---

> ### Author Response · Authors · 2025-11-21
> **Response to Reviewer fRyk (3)**
>
> > 3. Similarly, the most direct comparison method seems to be Friedl et al. (2025), but this comparison is missing.
> >
>
> Our approach builds on the Hamiltonian formulation and assumes a set of high-dimensional observations $( \mathbf q_{i}, \mathbf p_{i} )_{i=1}^{N}$. The approach of Friedl et al. (2025) instead considers a Lagrangian formulation and assumes to be given position and velocity data as observations. As the assumptions of the two approaches differ, we consider comparisons against other approaches based on the Hamiltonian formalism in Section 4. From a theoretical perspective, it is worth noticing that the Lagrangian formalism results in a second-order differential equation, while the Hamiltonian leads to a first-order differential equation.
>
> > 4. SPD network, constrained autoencoder, and the Strang-symplectic integrator are all existing components combined. The novelty appears somewhat limited. The definition of the symplectic autoencoder is very natural from the viewpoint of symplectic geometry.
> >
>
> Our RO-HNN is composed of a geometrically-constrained symplectic AE and of a latent geometric HNN whose dynamics are integrated with a Strang-symplectic integrator.
>
> We are glad that the reviewer agrees that a symplectic AE is very natural from a differential geometric perspective.
> We would like to emphasize that, while it leverages an existing constrained AE and is natural from a geometric perspective, to the best of our knowledge, **our geometrically-constrained symplectic AE is the first AE to structurally guarantee the preservation of the symplectic structure** of the high-dimensional Hamiltonian system in a low-dimensional latent space. As such, it goes noticeably beyond existing AE without hard constraints such as [10,11].
>
> While we agree that the SPD network is not novel on its own, **our paper is the first to consider the geometry of the mass-inertia and dissipation matrices in the context of Hamiltonian dynamics learning**, showing significant improvement over previous HNNs that leveraged its Cholesky decomposition [4-6].
>
> Finally, we believe that **our geometric reduced-order Hamiltonian neural network (RO-HNN) goes beyond the state of the art as a novel approach to learn unknown high-dimensional Hamiltonian dynamics**. In this regard, we would like to highlight a remark of Reviewer uBEM, who stated that “*Other components of the RO-HNN model take good advantages of methods from prior literature (e.g. the SPD network for parameterizing the HNN, symplectic integrators, constrained autoencoders)*”.

---

> ### Author Response · Authors · 2025-11-21
> **Response to Reviewer fRyk (4)**
>
> > 5. Both the autoencoder design and the numerical integration place strong emphasis on symplectic geometry. However, the actual latent-space dynamics is something like a port-Hamiltonian system, with dissipation and external forces. While the experiments demonstrate the usefulness of the Strang-symplectic integrator, the discussion could be strengthened.
>
> Thank you for this input! It is correct that the system ceases to be strictly Hamiltonian when dissipation and external forces are introduced.
>
> - **We refined Section 3.2 (”dissipative dynamics”) to precise the type of dissipative systems we assume, and how the structure of the resulting vector field is preserved by our geometrically-constrained symplectic AE**. Specifically, we updated Section 3.2 (lines 306-321) as follows, adding Proposition 3:
>
>     “ Specifically, we consider high-dimensional systems with observed external inputs $\mathbf \tau_{\text ext}$ and viscous damping following a Rayleigh dissipative function $\mathcal{D}(\mathbf{q}, \dot{\mathbf{q}}) = \frac{1}{2} \dot{\mathbf{q}}^\intercal \mathbf{D}(\mathbf{q})\dot{\mathbf{q}}$ with unknown positive semi-definite dissipation matrix $\mathbf{D}(\mathbf{q})$. The resulting force field is $\mathbf X_{\mathcal F} = \left(\begin{smallmatrix} \mathbf 0 \\\\ \mathbf \tau_{\text ext } + \mathbf \tau_{\text d } \end{smallmatrix}\right)$ with damping force $\mathbf \tau_{\text d} = \frac{\partial \mathcal{D}(\mathbf{q}, \dot{\mathbf{q}})}{\partial \dot{\mathbf{q}}} = \mathbf{D}(\mathbf{q})\dot{\mathbf{q}}$.
>
>     **Proposition 3.** The reduced vector field $\check{\mathbf{X}} = \varphi^*\mathbf{X}$ obtained via the pullback of the cotangent-lifted embedding $\varphi$ (8) preserves the structure of the total vector field $\mathbf{X} =  \mathbf X_{\mathcal H} + \mathbf X_{\mathcal F}$ with $\mathbf X_{\mathcal F} = \left(\begin{smallmatrix} \mathbf 0 \\\\ \mathbf \tau_{\text ext} + \mathbf \tau_{\text d} \end{smallmatrix}\right)$.”
>
> - In Section 3.2 (lines 327-331), we provided the following **additional discussion on structure preservation for dissipative dynamics, and on the relationship between the considered dissipative system and more general models**:
>
>    “ Note that the dissipative dynamics no longer preserve a symplectic structure. However, Proposition 3 shows that the RO-HNN latent space still preserves the structure of high-dimensional dissipative dynamical systems characterized by a Rayleigh dissipative function. It is worth noting that these systems are equivalent to Port-Hamiltonian systems with energy dissipation matrix $\mathbf{R}(\mathbf q,\mathbf p) = \left(\begin{smallmatrix}\mathbf 0 & \mathbf 0 \\\\ \mathbf 0 & \mathbf{D}(\mathbf{q}) \end{smallmatrix}\right)$ and input $\mathbf{G}(\mathbf q,\mathbf p)\mathbf u = \mathbf \tau_{\text ext}$, and to contact Hamiltonians $\mathcal H_{\text c}(\mathbf q,\mathbf p, s)$ with $\dot{s} = \mathcal{D}.$ ”
>
> - Finally, we conducted an **additional experiment to demonstrate and discuss the usefulness of the Strang-symplectic integrator**. Specifically, we compare against a non-symplectic Runge-Kutta integrator of order 4. The results are reported in **Fig. 20 in App. H.3** and discussed in **Section 4.3** (lines 521-526) as follows:
>
>    “ As the dissipative dynamics do not preserve the symplectic structure, we compare the Strang symplectic integrator, which assumes a symplectic structure, against a non-symplectic Runge-Kutta integrator of order 4. Fig. 20 in App. H.3 shows that, despite the dissipative structure, the Strang symplectic integrator outperforms the Runge-Kutta one. We hypothesize that this is due to the fact that the evolution of this dissipative system is mostly governed by its Hamiltonian function, especially over the short timesteps taken by the integrators. ”
>
> - In general, **we consider the generalization of the RO-HNN to more general dissipative dynamics as future work**. We highlighted this point in the conclusion as:
>
>    “ [In future work] we will also generalize the RO-HNN to learn more general dynamics of Port-Hamiltonian and contact Hamiltonian systems. ”

---

> ### Author Response · Authors · 2025-11-21
> **Response to Reviewer fRyk (5)**
>
> > 6. Prediction errors accumulate, so as the prediction time grows, errors can become extreme. A single divergent trajectory can drastically worsen the score. In fact, the standard deviation is about the same size as, or larger than, the average error. A three-times difference could be overturned. It would be better to include plots of error growth or adopt robust metrics.
> >
>
> We added the **following visualizations** to the manuscript to strengthen and confirm the results observed via the concise mean and standard deviations originally reported in Tables 1-4:
>
> - We updated Fig. 2-top to visualizes the median and quartiles of the relative position prediction error over a prediction time horizon $\Delta t = 5$s for different learned architectures on the 15-DoF pendulum dataset.
> - We added a new Fig. 9 that reports the median and interquartile range of the prediction errors obtained over pendulum trajectories with different intrusive symplectic MOR approaches, supporting the quantitative evaluation reported in Table 2.
> - We added a new Fig. 13 that reports the median interquartile range of the prediction errors obtained over particle vortex trajectories with different latent space dimensions and latent HNNs, supporting the quantitative evaluation reported  in Table 3.
> - We added a new Fig. 14 that visualizes the median and quartiles of latent and embedded position and momentum errors over cloth trajectories for a prediction time horizon $\Delta t = 0.5$s for different choices of learned latent damping, supporting the quantitative evaluation reported  in Table 4.
>
> > 7. There is some confusion in the Introduction. PINNs use governing equations to define losses. Unless extended in combination with methods like SINDy, they are not intended for learning unknown dynamical systems, and their objective differs from HNNs and Neural Operators. Since this work is an extension of HNNs, discussing it separately from Neural Operators and PINNs might reduce confusion.
> >
>
> We thank the reviewer for their suggestion. We **revised the first paragraph of the introduction** as follows, to broadly introduce physics-inspired models before focusing on models that learn unknown dynamics by embedding physics priors as hard constraints in neural models:
>
> “Learning the unknown governing equations of dynamical systems is of fundamental importance to model physical processes. In this context, generic neural models lack built-in physical intuition, thus resulting in limited explainability and poor generalization beyond the data support. Embedding fundamental physical properties, such as conservation laws and boundary conditions, into neural networks has been shown to drastically improve their performance. Various models incorporate physical intuition as soft constraints via penalty terms in the loss function (Saqlain et al., 2023). This often leads to suboptimal enforcement of physical properties and to stiff optimization (Wang et al., 2021), motivating the embedding of domain-specific priors as hard constraints in specialized neural architectures. This allowed recent methods to learn dynamics that preserve energy (Greydanus et al., 2019; Cranmer et al., 2020; Lutter & Peters, 2023), conserve mass and momentum (Jnini et al., 2025), and strictly enforce general conservation laws (Liu et al., 2024), thereby improving
> performances, generalization, and stability while yielding physically-consistent predictions.”
>
> > 8. In Eq. (8), the left-hand side takes arguments, so the right-hand side should also take arguments.
> >
>
> Thank you for pointing this out! We rewrote Equation 8 with arguments in the right-hand side as:
>
> $\varphi(\check{\mathbf{q}}, \check{\mathbf{p}}) = \left(\begin{matrix} \varphi_{\mathcal{Q}} (\check{\mathbf{q}}) \\\\ d\rho_{\mathcal{Q}}\vert_{\varphi_{\mathcal{Q}}(\check{\mathbf{q}})}^T \check{\mathbf{p}} \end{matrix}\right)$ and $\rho(\mathbf{q}, \mathbf{p}) = \left(\begin{matrix}
> \rho_{\mathcal{Q}} (\mathbf{q}) \\\\ d\varphi_{\mathcal{Q}}\vert_{\rho_{\mathcal{Q}}(\mathbf{q})}^T \mathbf{p}
> \end{matrix}\right)$

---

> ### Author Response · Authors · 2025-11-21
> **Response to Reviewer fRyk (6)**
>
> > 9. Line 278
> >
> >
> > > Existing HNNs enforce the symmetric positive-definiteness of the inverse mass-inertia matrix via a Euclidean network encoding its Cholesky decomposition
> > >
> >
> > At least the original HNNs did not implement it that way. Perhaps this refers to Dissipative SymODEN? Additional explanation would help.
> >
>
> It is correct that the original HNN [3] directly learns the Hamiltonian function $\mathcal{H}$. We aimed at referring to existing HNNs such as [4] and [5] (SymODEN) that model the Hamiltonian functions via two networks parametrizing the inverse-mass inertia matrix and the potential energy. We added citations to [4], [5] at this point in the text (line 291) to clarify that we refer to these specific Hamiltonian models.
>
> > 10. From the perspective of learning non-canonical Hamiltonian systems and dimension reduction with neural networks, in addition to neural symplectic forms, I recommend citing the following papers.
> >
> >
> > Jin et al., "Learning Poisson systems and trajectories of autonomous systems via Poisson neural networks," IEEE Transactions on Neural Networks and Learning Systems, 2023.
> >
> > Sipka et al., "Direct Poisson neural networks: Learning non-symplectic mechanical systems," Journal of Physics A: Mathematical and Theoretical, 2023.
> >
>
> We thank the reviewer for the literature suggestions that we did not cover in our initial submission. We agree that these works are relevant, and we have now **incorporated them into the revised paper**, at the end of the second paragraph of the introduction as follows:
>
> “While most HNNs consider Hamiltonians characterized by a canonical symplectic form — exhibited at least locally for all Hamiltonian systems — few works proposed architectures handling non-canonical forms (Chen et al., 2021) and more general Poisson systems (Jin et al., 2021; Sipka et al., 2023).”
>
> ## Questions:
>
> > 1. The proposed autoencoder is implementable with vector-Jacobian products, so the computational overhead (compared to simple autoencoders) appears to be only a few times larger. Please clarify the computational overhead.
> >
>
> As suggested by the Reviewer, the proposed autoencoder, i.e., the cotangent-lifted maps of Eq. (8), can be implemented via transposed Jacobian-vector products. For enhanced computational efficiency, we implemented the pullbacks $d\rho_{\mathcal{Q}}\vert_{\varphi_{\mathcal Q}(\check{\mathbf{q}})}^\intercal \check{\mathbf{p}}$ and $d\varphi_{\mathcal{Q}}\vert_{\rho_{\mathcal{Q}}(\mathbf{q})}^\intercal \mathbf{p}$ within the cotangent-lifted maps as a composition of analytically-computed transposed derivatives of the encoder $\rho_\mathcal{Q}$ and decoder $\varphi_{\mathcal{Q}}$. Following the Reviewer’s suggestion, **we added an Appendix D.2 which describes the associated analytical computations**. Moreover, **we added the corresponding runtimes in App. H.4**, which indicate that wall-clock time for evaluation of the analytically-computed cotangent-lifted autoencoder is significantly lower than twice that of a simple position-level autoencoder.
>
> > 2. At line 306, is $\rho$ a typo for $\varphi$ ? Notations are also hard to read. Do subscripts indicate predicted quantities, while symbols without indicate encoder outputs?
> >
>
> We thank the reviewer for spotting this typo! We corrected the sentence which now reads: “(2) decoding the obtained reduced-order position and momentum $(\check q_{p}, \check p_{p} )$ into the high-dimensional coordinates of the original system with the lifted point embedding $\varphi$ (8), i.e.,  $(\tilde q_{p}, \tilde p_{p}) = \varphi(\check q_{p},\check p_{p})$”.
>
> The subscript $\text{p}$ indicates the predicted quantity, while symbols without subscript denotes the encoder outputs. We clarified this at line 336 of the revised manuscript.

---

> > ### Author Response · Authors · 2025-11-21
> > **Response to Reviewer fRyk (7/7)**
> >
> > ## References
> >
> > [1] Peter Woit. Quantum theory, groups and representations. Springer, 2017.
> >
> > [2] Ole Steuernagel. Equivalence between focused paraxial beams and the quantum harmonic oscillator. American Journal of Physics, 2005.
> >
> > [3] Samuel Greydanus, Misko Dzamba, and Jason Yosinski. Hamiltonian neural networks. In Neural Information Processing Systems (NeurIPS), volume 32, 2019
> >
> > [4] Michael Lutter and Jan Peters. Combining physics and deep learning to learn continuous-time dynamics models. Intl. Journal of Robotics Research, 42(3):83–107, 2023.
> >
> > [5] Yaofeng Desmond Zhong, Biswadip Dey, and Amit Chakraborty. Dissipative SymODEN: Encoding Hamiltonian dynamics with dissipation and control into deep learning. In ICLR Workshop on Integration of Deep Neural Models and Differential Equations (DeepDiffEq), 2020.
> >
> > [6] Thai Duong and Nikolay Atanasov. Hamiltonian-based neural ODE networks on the SE(3) manifold for dynamics learning and control. In Robotics: Science and Systems (R:SS), 2021.
> >
> > [7] Jing-Yi Liu, Pablo Borja and Cosimo Della Santina. “Physics‐Informed Neural Networks to Model and Control Robots: A Theoretical and Experimental Investigation.” *Advanced Intelligent Systems* 6, 2023.
> >
> > [8] Yuhan Chen, Takashi Matsubara, and Takaharu Yaguchi. Neural symplectic form: Learning Hamiltonian equations on general coordinate systems. In Neural Information Processing Systems (NeurIPS), 2021
> >
> > [9] K. Otness *et al.* An Extensible Benchmark Suite for Learning to Simulate Physical Systems. In NeurIPS *Track on Datasets and Benchmarks,* 2021.
> >
> > [10] Buchfink, P., Glas, S., & Haasdonk, B. (2023). Symplectic model reduction of Hamiltonian systems on nonlinear manifolds and approximation with weakly symplectic autoencoder. SIAM Journal on Scientific Computing, 45(2), A289-A311.
> >
> > [11] Marco Lepri, Davide Bacciu, and Cosimo Della Santina. Neural autoencoder-based structure-preserving model order reduction and control design for high-dimensional physical systems. IEEE Control Systems Letters, 8:133–138, 2024.
> >
> > [12] Samuel E. Otto, Gregory R. Macchio, and Clarence W. Rowley. Learning nonlinear projections for reduced-order modeling of dynamical systems using constrained autoencoders. Chaos: An Interdisciplinary Journal of Nonlinear Science, 33(11), 2023.

---

> > > ### Comment · Reviewer_fRyk · 2025-11-22
> > >
> > > Thank you very much for your detailed response. I am now convinced by your new explanation regarding the dissipation/inputs and your choice of the Strang symplectic integrator. I also find the newly rewritten introduction appropriate.
> > >
> > > In my Q.9, I did not consider SymODEN to be an HNN but its extension, but if you prefer to refer to this line of work collectively as "HNNs", I am happy to respect that viewpoint.
> > >
> > > > it does not consider high-dimensional observations (e.g., images) of low-dimensional physical systems but systems with high-dimensional state spaces, a.k.a high-dimensional dynamics.
> > >
> > > I apologize that my sentence was misleading. I do understand this point.
> > >
> > > From the beginning, I have appreciated the elegance of your method, and I think only one question remains for me: what is the benefit of using a Hamiltonian formulation?
> > >
> > > > The approach of Friedl et al. (2025) instead considers a Lagrangian formulation and assumes to be given position and velocity data as observations.
> > >
> > > I understand the technical difference. However, the fact that your approach is based on a Hamiltonian formulation and uses momentum as observations seems to be a disadvantage rather than a strength. At the very least, Friedl et al. (2025) can tackle the experiment in Figure 5, and it feels more natural to tackle it with Friedl et al. (2025), that is, in a Lagrangian formulation, than with your proposed method.
> > >
> > > > for many canonical Hamiltonian system, the canonical momentum can be observed without knowing the mass
> > >
> > > That is true in the case of the particle vortex. However, it seems somewhat weak if the particle vortex is the only good application. Therefore, if you could compare the experiment in Figure 5 with Friedl et al. (2025) and demonstrate better performance (and provide justification for why it is better), that would constitute a very strong advantage.
> > >
> > > > In our paper, we follow the assumption commonly adopted in the HNN literature [3-8],
> > >
> > > [4][5][6][7] convert observations from (angular) velocity to momentum by learning the mass matrix. [8] also learns the coordinate system from data, so the observed quantities do not need to be the canonical momentum. In fact, these works demonstrate that the assumption in [3] that "the momentum is directly observable" is somewhat unnatural and needs to be resolved.
> > >
> > > That said, I have to admit that my first rating of 4 was undeservedly low. I would like to increase it to 6.

---

> ### Author Response · Authors · 2025-11-27
> **Answer to remaining concerns of Reviewer fRyk**
>
> Many thanks for your response and for following up! We are very glad to know that you appreciate the “elegance” of our approach. We hope to clarify the remaining concern regarding benefits in the following.
>
> We uploaded a revised manuscript with additional revisions highlighted in pink.
>
> > […] , the fact that your approach is based on a Hamiltonian formulation and uses momentum as observations seems to be a disadvantage rather than a strength. […] it seems somewhat weak if the particle vortex is the only good application. Therefore, if you could compare the experiment in Figure 5 with Friedl et al. (2025) and demonstrate better performance (and provide justification for why it is better), that would constitute a very strong advantage.
> […]
> [4][5][6][7] convert observations from (angular) velocity to momentum by learning the mass matrix. [8] also learns the coordinate system from data, so the observed quantities do not need to be the canonical momentum. In fact, these works demonstrate that the assumption in [3] that "the momentum is directly observable" is somewhat unnatural and needs to be resolved.
> >
>
> We agree that [6] indeed convert observations from velocity to momentum by learning the mass matrix. However, the experiments in [4],[5],[7] assume momentum observations. We would also like to emphasize that canonical momentum can also be observed in other scenarios than the particle vortex, e.g., for quantum mechanical systems [1] and optical systems [2], although we did not conduct experiments on such these scenarios in the present paper.
>
> Nevertheless, **as rightfully suggested by the Reviewer, we added a comparison between the RO-LNN from Friedl et al. (2025) and the proposed RO-HNN for the 600-DoFs cloth experiment of Sec. 4.3** (see App. H.3). Specifically, we trained RO-LNNs with 10-dimensional latent space and a latent LNN using a similar training procedure as for our RO-HNN, adapted from the code from Friedl et al. (2025) (see App. G.3). Note that the RO-LNN is conservative, and we provide the dissipation forces $\tau_{\text{d}}$ as ground truth in the external input, i.e., $\tau = \tau_{\text{c}} + \tau_{\text{d}}$, similarly as for the conservative RO-HNN considered in Sec. 4.3. The results are displayed in App. H3 by:
>
> - The updated Fig. 15, which now includes a fourth row featuring the RO-HNN predictions, to be compared against the ground truth (first row), and RO-HNN with 10-dimensional latent space (third row);
> - The new Fig. 21, which compares the latent prediction and reconstructed prediction errors of the RO-LNN and RO-HNN over time. Note that Fig. 21 considers (1) a RO-HNN with geometric latent HNN, (2) a RO-HNN with a latent HNN encoded as a black-box network, (3) a RO-LNN with geometric latent LNN (Friedl et al, 2025), and (4) a RO-LNN with a latent LNN encoded as a black-box network.
>
> As discussed in App. H.3., **our results show that the RO-HNN outperforms the RO-LNN, leading to more accurate short- and long-term predictions**. This is observed qualitatively in Fig. 15, especially for $t\geq 0.3$s, and quantitatively in Fig. 21. It is worth noting that the RO-HNN in Fig. 15 outperforms the RO-LNN despite that it also learns the dissipation forces via the latent damping matrix, while the ground truth dissipation forces are provided to the RO-LNN. Moreover, we observe that the RO-HNN and RO-LNN with geometric latent HNN and LNN outperform their black box counterparts, showcasing the importance of considering the quadratic energy structure of mechanical systems in both network types.
>
> We hypothesize that the improved accuracy of the RO-HNN compared to the RO-LNN can be attributed to **the first-order dynamic formulation stemming from Hamiltonian mechanics**, which is easier to learn and optimize than the second-order Lagrangian formulation. This aligns with discussion in previous works [7], which showed that, by using position and momentum observations, HNNs learn mass-inertia matrices that are close to the physical solutions, while LNNs only learn one of the solutions satisfying the Euler-Lagrange equations, i.e, via a scaled mass-inertia matrix. We also attribute the improved accuracy of the RO-HNN to **the Strang-symplectic integration** which is specifically designed for Hamiltonian systems and cannot be used for integrating Lagrangian dynamics. In contrast, more generic Runge-Kutta integrators are typically used in the continuous-time Lagrangian case.

---

> > ### Comment · Reviewer_fRyk · 2025-11-28
> >
> > Thank you for pointing this out. Reference [5] indeed uses momentum. However, references [4] and [7] use either velocity and momentum depending on the task, and, in particular, [4] states the limitations of Hamiltonian formulations as follows:
> >
> > > We only use DeLaN for the physical systems as for these systems we do not observe the momentum. Hence, only the Lagrangian physics prior is applicable.
> >
> > > For all physical systems, only the joint position is directly observed. The velocities and accelerations are computed using finite differences and low-pass filters.
> >
> > Therefore, if considering practical use rather than proof-of-concept experiments, I still cannot accept the assumption that the momentum of mechanistic systems is observable.
> >
> > However, if the proposed method is used to accelerate the simulation of known systems, then the assumption that the momentum is observable is acceptable. This is what I meant in weakness 4.
> >
> > Thank you also for the additional experiments. It is very interesting that the proposed method can accelerate the simulations of known systems, despite using neural networks. Then, how is the trade-off between the error and the speed-up when using the proposed method? When applying a lower or higher order numerical integration to the FOM, we also get a trade-off between the numerical error and the speed-up. Does your proposed method will offer a better trade-off than numerical integrations? Of course, I understand that this would strongly depend on the dataset size.
> >
> > I understand that I am now pointing out very fine details. This is just one suggestion, and I am completely fine if the advantage is demonstrated in another way. In any case, at this moment, I do not really perceive much benefit in the fact that it accelerates the cloth simulations using momentum.

---

> ### Author Response · Authors · 2025-11-30
> **Answer to further remaining concerns of Reviewer fRyk (1)**
>
> > Therefore, if considering practical use rather than proof-of-concept experiments, I still cannot accept the assumption that the momentum of mechanistic systems is observable.
>
> Thank you for following up!
>
> For physical mechanical systems where the canonical momentum cannot be measured, an estimate of the mass-inertia matrix can often be retrieved via CAD models, while external potentials and damping remain unknown. In such cases, the RO-HNN can leverage the known mass-inertia matrix to compute the momentum as $\mathbf p = \mathbf M(\mathbf q) \dot{\mathbf q}$ and obtain the reduced mass-inertia matrix through the proposed geometrically-constrained symplectic AE as $\check{\mathbf M}(\mathbf q) = d\varphi_{\mathcal Q}^\intercal \mathbf M(\mathbf q)d\varphi_{\mathcal Q}$, while learning (1) the parameters of the geometrically-constrained symplectic AE, and (2) the reduced potential $\check V_\theta$ and damping matrix  $\check{\mathbf D}_\theta$.
>
> By providing an effective solution for learning reduced-order dynamics, **our RO-HNN opens new avenues for model-based applications beyond forward simulation**. As stated in Sec. 5, this includes **model-based control strategies**, which could take advantage of available estimates of $\mathbf M(\mathbf q)$ and of the compressed representation of the RO-HNN for applications related to the control of soft robots or deformable objects [1]. The RO-HNN may also be used to **extract non-linear synergies** of high-dimensional anthropomorphic hands [2,3] or humanoid robots [4] — for which an estimate of $\mathbf M(\mathbf q)$ is typically available — in the form of a structure-preserving and physically-consistent symplectic subspace, and subsequently design associated controllers.
>
> As pointed out by the Reviewer, our geometrically-constrained symplectic AE and RO-HNN can also be used in **classical intrusive model-order reduction scenarios where the high-dimensional dynamical parameters are assumed to be fully known to speed-up forward simulations**, as discussed in our first answer to weakness 2.
>
> > Then, how is the trade-off between the error and the speed-up when using the proposed method? When applying a lower or higher order numerical integration to the FOM, we also get a trade-off between the numerical error and the speed-up. Does your proposed method will offer a better trade-off than numerical integrations? Of course, I understand that this would strongly depend on the dataset size.
> >
>
> We updated our analysis in App. H. 4 “Speedup of simulation via the RO-HNN “ to deliver a more complete picture. In the new Table 6 and Figure 22, we compare the wall-clock time of evaluation of the FOMs of the 15-DoF pendulum and 600-DoF cloth against their respective ROM obtained via the geometrically-constrained symplectic integrator. We consider several integration schemes — namely the Strang-symplectic integration of the FOMs and ROMs, and a computationally-cheaper alternative via forward Euler integration for the FOM — and several step sizes $\Delta t=\{10^{-2},10^{-1}\}$ for the pendulum, and $\Delta t=\{10^{-4},10^{-3}\}$ for the cloth. We compare the evaluation time and the reconstruction errors in all scenarios and observe a prominent reduction of the evaluation time for the RO-HNNS compared to the FOMs. This reduction is exacerbated for higher-dimensional systems, e.g., the cloth, where the evaluation time of the RO-HNN remains significantly lower than that of the FOM evaluated with the forward Euler integrator or increased step size. Moreover, the RO-HNN exhibit lower prediction errors than their FOM counterparts in addition to a reduced computational complexity. Note that significantly increasing $\Delta t$ to achieve speedup with the Euler integrator, e.g. by a factor $10$, led to instability in all observed scenarios, validating the intuition of the reviewer about the necessary trade off between the achieved reconstruction error and speed-up.

---

> > ### Author Response · Authors · 2025-11-30
> > **Answer to further remaining concerns of Reviewer fRyk (2/2)**
> >
> > ### Summary of our discussion on the Hamiltonian formulation and momentum observations
> >
> > - The RO-HNN leads to increased performances compared to its Lagrangian counterpart, which we primarily attribute to the first-order formulation stemming from Hamiltonian mechanics and to the Strang-symplectic integrator specifically designed for Hamiltonian systems.
> > - Momentum is observable in a range of dynamical systems, e.g., the particle vortex of Sec. 4.2, quantum mechanical systems, and optical systems, to name a few.
> > - For physical mechanical systems where the canonical momentum cannot be measured, an estimate of the mass-inertia matrix is often available and can be used to retrieve momentum observations and a reduced-order mass-inertia matrix.
> > - Many scenarios consider observations in the form of images, in which case the position and momentum are retrieved via an auxiliary AE.
> > - Our geometrically-constrained symplectic AE can also be used in classical intrusive model-order reduction scenarios where the high-dimensional dynamical parameters are assumed to be fully known to speed-up forward simulations.
> >
> > Our work then provides a novel and effective solution to learn guaranteed-structure-preserving symplectic latent spaces and (some of) the associated dynamic parameters, and open new avenues for model-based applications beyond forward simulation.
> >
> > ### References
> >
> > [1] Marco Lepri, Davide Bacciu, and Cosimo Della Santina. Neural autoencoder-based structure-preserving model order reduction and control design for high-dimensional physical systems. IEEE Control Systems Letters, 8:133–138, 2024.
> >
> > [2] J. Romero, T. Feix, C. H. Ek, H. Kjellström and D. Kragic. Extracting Postural Synergies for Robotic Grasping. IEEE Transactions on Robotics, 29:6, pp. 1342-1352, 2013.
> >
> > [3] F. Ficuciello, G. Palli, C. Melchiorri, B. Siciliano. Postural synergies of the UB Hand IV for human-like grasping. Robotics and Autonomous Systems, 62:4, 2014.
> >
> > [4] R. Malhotra, W. Chong, C. Cuan and O. Khatib. Humanoid Motion Scripting with Postural Synergies.  IEEE-RAS International Conference on Humanoid Robots (Humanoids), 2025.

---

### Official Review · Reviewer_cMWE · 2025-10-28

**Soundness:** 2
**Presentation:** 1
**Contribution:** 1
**Rating:** 2
**Confidence:** 4

**Summary:**

This paper presents RO-HNN (Geometric Reduced-order Hamiltonian Neural Network), which aims to extend Hamiltonian neural networks (HNNs) to high-dimensional systems. The model combines (1) a geometrically constrained symplectic autoencoder for structure-preserving dimensionality reduction and (2) a geometric HNN for learning reduced-order Hamiltonian dynamics in latent space. By enforcing symplecticity and biorthogonality constraints within the autoencoder and using SPD networks to parametrize the latent inertia and damping matrices, the authors claim to obtain stable and physically consistent dynamics. Experiments conducted on a series of physical models show improved reconstruction and long-term prediction compared to vanilla HNNs.

**Strengths:**

1. The paper systematically integrates geometric and symplectic constraints across model components, which guarantees the outperformance in preserving the physical laws during the learning process.

2. Empirical results demonstrate advantages of the proposed method over the baseline HNNs.

**Weaknesses:**

1. Lack of originality:The proposed RO-HNN essentially combines existing components. For example, using an autoencoder for latent-space projection of Hamiltonian systems is a well-established idea [R1],[R2]. The concept of symplectic or structure-preserving projections has been explored in SympNet [R3]. Modeling external forces and damping has already been discussed in [R4],[R5].
Thus, the method largely stitches together known techniques rather than introducing a new conceptual contribution.

2. The paper frames its “differential-geometric” perspective as new, but most theoretical results are restatements of standard symplectic and Riemannian geometry properties already available in textbooks or prior works [R6].

3. In the experiments, all datasets are clean and simulated, and the paper does not analyze performance under observation noise, though this is critical for real-world Hamiltonian systems (e.g., robotics, experimental physics).

4. Given that the framework aims to handle high-dimensional systems, it would be natural to compare against Koopman-based operator learning or neural operator baselines [R1],[R2],[R7]. The current paper mainly compares with vanilla HNN, which was been developed six years ago and is not the SOTA method in the related tasks.

__References__
[R1] Lusch, B., Kutz, J. N., & Brunton, S. L. (2018). Deep learning for universal linear embeddings of nonlinear dynamics. Nature communications, 9(1), 4950.

[R2] Zhang, J., Zhu, Q., & Lin, W. (2024). Learning hamiltonian neural koopman operator and simultaneously sustaining and discovering conservation laws. Physical Review Research, 6(1), L012031.

[R3] Jin, P., Zhang, Z., Zhu, A., Tang, Y., & Karniadakis, G. E. (2020). SympNets: Intrinsic structure-preserving symplectic networks for identifying Hamiltonian systems. Neural Networks, 132, 166-179.

[R4] Zhong, Y. D., Dey, B., & Chakraborty, A. (2019). Symplectic ode-net: Learning hamiltonian dynamics with control. arXiv preprint arXiv:1909.12077.

[R5] Liu, Z., Wang, B., Meng, Q., Chen, W., Tegmark, M., & Liu, T. Y. (2021). Machine-learning nonconservative dynamics for new-physics detection. Physical Review E, 104(5), 055302.

[R6] Buchfink, P., Glas, S., & Haasdonk, B. (2023). Symplectic model reduction of Hamiltonian systems on nonlinear manifolds and approximation with weakly symplectic autoencoder. SIAM Journal on Scientific Computing, 45(2), A289-A311.

[R7] Xiong, W., Huang, X., Zhang, Z., Deng, R., Sun, P., & Tian, Y. (2024). Koopman neural operator as a mesh-free solver of non-linear partial differential equations. Journal of Computational Physics, 513, 113194.

**Questions:**

Q1: How does the proposed method compare to Koopman-based latent Hamiltonian learning [R1],[R2] in terms of reconstruction accuracy and scalability when the observational data is perturbed by noise?

Q2: Can the model handle partial observations (e.g., only $q$ is available)?

Q3: Since the method combines known components, what is the main conceptual novelty that differentiates it from prior structure-preserving model-reduction approaches?

Q4: Can the proposed framework handles irregular time series data?

Q5: In line 295, should it be 'constrain' instead of 'constraint'?

Q6: In line 350, the integrals of $\check{q}$ and $\check{p}$ should solve the IVP, what the initial value in this case?

Q7: In line 423, it should be $\{6,10\}$ instead of $\{6.10\}$.

Q8: In line 481, is the title of the 5th and 6th column in Table 4 right? Does the $\check{q}_p$ and $q$ have the same dimension?

Q9: In your experiments, why do you select the hyperparameter of latent dimension as $6,10$?

---

> ### Author Response · Authors · 2025-11-21
> **Response to Reviewer cMWE (1)**
>
> Thank you for the valuable feedback! Below, we hope to reach the raised concerns and clear up differences compared to existing related works.
>
> ### Weaknesses
>
> > 1. Lack of originality: The proposed RO-HNN essentially combines existing components. For example, using an autoencoder for latent-space projection of Hamiltonian systems is a well-established idea [R1],[R2]. The concept of symplectic or structure-preserving projections has been explored in SympNet [R3]. Modeling external forces and damping has already been discussed in [R4],[R5]. Thus, the method largely stitches together known techniques rather than introducing a new conceptual contribution.
> >
>
> We would like to emphasize that the contributions of our paper go beyond combining existing components as follows:
>
> 1. **Conceptual novelty:** Our approach aims at **learning high-dimensional Hamiltonian dynamics from data by learning the low-dimensional dynamic parameters of a surrogate lower-dimensional Hamiltonian system.** Therefore, our approach goes beyond (1) previous HNNs that only handle low-dimensional physical systems, and (2) Hamiltonian model-order reduction approaches that are intrusive, i.e., assume to have full knowledge of the full-order system dynamics.
>
>
>
>   2. **Relationship to Koopman-based latent spaces learned via AEs [R1],[R2]:** Our paper introduces a novel geometrically-constrained symplectic AE that learns a low-dimensional symplectic latent space from trajectory data lying on a high-dimensional symplectic manifold. To the best of our knowledge, **our geometrically-constrained symplectic AE is the first AE that strictly guarantees the preservation of the symplectic structure in the learned latent space**. Although [R1],[R2] also use a AE, these approaches aim at learning a latent space where the dynamics of the original system become linear, i.e., a flattening latent space. Their models consist of (1) an AE that identifies a set of (potentially higher-dimensional) intrinsic coordinates spanned by a set of Koopman eigenfunctions, and (2) a linear dynamical system (represented via a Koopman matrix) learned in the latent space of the AE. **Our approach differs in that it learns a nonlinear symplectic latent space that preserves the symplectic structure of the original high-dimensional system**.
>
>
> 3. **Relationship to existing models that learn Hamiltonian dynamics [R3]:** SympNets [R3] learns Hamiltonian dynamics by parametrizing symplectomorphisms via a neural network. Similarly to HNNs that directly learn the Hamiltonian function [R11] or its components [R12], the physical systems considered by SympNets were low dimensional (up to 6 DoF (12-dimensional state space) in [R3]). Moreover, the symplectomorphisms learned by SympNets preserve the dimension of the original system. **In contrast, our approach learns the dynamics of high-dimensional Hamiltonian systems (up to 600 DoF in our experiments) via a low-dimensional structure-preserving representation of the original high-dimensional Hamiltonian system.** Moreover, SympNets model dynamic evolutions as symplectomorphisms with fixed stepsize, while our approach allows us to simulate continuous dynamics with variable step size.
> 4. **Relationship to existing models that consider external forces and damping [R4],[R5]:** Similarly to conservative HNNs [R3,R11,R12], Symplectic ODE-Net [R4] learns the dynamics of low-dimensional dissipative Hamiltonian systems (up to 6 state dimensions). Moreover, [R5] proposed to learn dissipative Lagrangian dynamics by decomposing the system into a conservative component modeled via a Lagrangian neural network (LNN) and a non-conservative component modeled via a universal approximator network. As [R4], [R5] only considers low-dimensional dissipative systems (up to 2 DoF). **In contrast to [R5] which learns low-dimensional dissipative Lagrangian dynamics, our work considers high-dimensional dissipative Hamiltonian systems**. Similarly to [R4], our approach learns a conservative Hamiltonian vector field via a HNN and a dissipative component. However, **our approach differs from [R4] in that it learns high-dimensional Hamiltonian dynamics, thereby learning the conservative and dissipative dynamic parameters in a low-dimensional structure-preserving symplectic latent space**.

---

> > ### Author Response · Authors · 2025-11-21
> > **Response to Reviewer cMWE (2)**
> >
> > 5. **Components of the proposed RO-HNN:** Our RO-HNN is composed of a geometrically-constrained symplectic AE and of a latent geometric HNN whose dynamics are integrated with a Strang-symplectic integrator. We would like to emphasize that, while it leverages an existing constrained AE, **the proposed geometrically-constrained symplectic AE is a contribution on its own**, as it is, to the best of our knowledge, **the first AE that structurally guarantees the preservation of the symplectic structure** of the high-dimensional Hamiltonian system in a low-dimensional latent space. Moreover, we would also like to point out that, **although the SPD network is not novel, our paper is the first to consider the geometry of the mass-inertia and dissipation matrices in the context of Hamiltonian dynamics learning**, showing significant improvement over previous HNNs that leveraged its Cholesky decomposition. Finally, we believe that **our geometric reduced-order Hamiltonian neural network (RO-HNN) goes beyond the state of the art as a novel approach to learn unknown high-dimensional Hamiltonian dynamics**. In this regard, we would like to highlight a remark of Reviewer uBEM, who stated that “*Other components of the RO-HNN model take good advantages of methods from prior literature (e.g. the SPD network for parameterizing the HNN, symplectic integrators, constrained autoencoders)*”.
> >
> > > 2. The paper frames its “differential-geometric” perspective as new, but most theoretical results are restatements of standard symplectic and Riemannian geometry properties already available in textbooks or prior works [R6].
> > >
> >
> > As acknowledged in Section 2 (”Background”), our paper employs the symplectic geometry underlying Hamiltonian systems commonly presented in textbooks (Section 2.1), as well as the theoretical framework from [R8] (Section 2.2) to subsequently obtain the proposed RO-HNN, which builds on symplectic and Riemannian geometric properties, in Section 3. As such, **our paper takes a differential-geometric perspective to tackle the problem of learning high-dimensional Hamiltonian dynamics**, thus contrasting with existing approaches learning low-dimensional Hamiltonian dynamics [R1,R2,R3,R4,R5,R11,R12].
> >
> > As highlighted in Section 1 (lines 74-77) and in Section 2.2, we take inspiration from several MOR approaches [R6,R8] that also build upon differential geometric properties. While our work builds on the theoretical framework from [R8], it is important to emphasize that it differs from [R6,R8] in that **the full-order dynamic parameters are unknown**, while the models [R6,R8] are intrusive, i.e., they assume to have access to the full-order Hamiltonian $\mathcal{H}$ (see also our answer to Q3 below). Moreover, **our geometrically-constrained symplectic AE differs from the weakly-symplectic AE from [R6] as it guarantees the preservation of the symplectic structure via hard structural constraints**. Instead, the weakly-symplectic AE from [R6] only encourages the preservation of the symplectic structure via an auxiliary loss function.
> >
> > > 3. In the experiments, all datasets are clean and simulated, and the paper does not analyze performance under observation noise, though this is critical for real-world Hamiltonian systems (e.g., robotics, experimental physics).
> > >
> >
> > > 4. Given that the framework aims to handle high-dimensional systems, it would be natural to compare against Koopman-based operator learning or neural operator baselines [R1],[R2],[R7]. The current paper mainly compares with vanilla HNN, which was been developed six years ago and is not the SOTA method in the related tasks.
> > >
> >
> > Thank you for raising up these points. We addressed them by adding new experiments with observation noise and new comparisons against Koopman-based approaches in the paper, which we detail in our answer to Q1 below.

---

> > > ### Author Response · Authors · 2025-11-21
> > > **Response to Reviewer cMWE (3)**
> > >
> > > ### Questions
> > >
> > > > Q1: How does the proposed method compare to Koopman-based latent Hamiltonian learning [R1],[R2] in terms of reconstruction accuracy and scalability when the observational data is perturbed by noise?
> > > >
> > >
> > > Koopman-based methods such as [R1,R2] aim at learning a linear dynamical system in a lifted, higher-dimensional latent space and dynamic model. Our proposed RO-HNN differs in that it learns a nonlinear Hamiltonian system in a lower-dimensional symplectic latent space. We extended our state of the art to include Koopman-based approaches as follows (line 68-71): “A different line of works leverage Koopman operator theory to model nonlinear, e.g., Hamiltonian, dynamics via a learned surrogate linear dynamic model embedded in a higher-dimensional latent space (Lusch et al., 2018; Zhang et al., 2024).”
> > >
> > > Moreovoer, we implemented HKNO from [R2] and considered it as an additional baseline in Section 4.1. The results on the 15-DoF pendulum are reported in Table 1, Fig. 2, and Fig. 7. We observe that the HNKO shows significantly higher prediction errors (one order of magnitude higher) than our RO-HNN.
> > >
> > > We also conducted an additional experiment to assess the robustness of the RO-HNNs under noisy observations, compared to the HNKO baseline, which is reported to be robust to noise in high-dimensional systems. We generate noisy training data of the 15-DoF coupled pendulum with different level of noise (0, 5, 10%). The results are reported in App. H. 1 (”Training under noisy observations”) and Table 5. We observe that the performance of both models decreases with increasing noise magnitude. In each scenario, the RO-HNN outperforms the HNKO baseline. Note that the RO-HNN trained at a level noise of 10% outperforms the HNKO trained without noise, demonstrating the enhanced accuracy and robustness of the RO-HNN to noisy observations.
> > >
> > > > Q2: Can the model handle partial observations (e.g., only $q$ is available)?
> > > >
> > >
> > > In this paper, we consider observations of the full state $(\mathbf{q}, \mathbf{p})$ to be available, and did not consider the problem of partial observability.
> > >
> > > > Q3: Since the method combines known components, what is the main conceptual novelty that differentiates it from prior structure-preserving model-reduction approaches?
> > > >
> > > 1. Prior structure-preserving model-order reduction (MOR) approaches are intrusive, i.e., they assume to have full knowledge of the full-order system dynamics [R6, R8-10]. In other words, classical MOR approaches do not learn dynamics, but reduce the dimension of known high-dimensional dynamics. Instead, **our method tackles scenarios where the high-dimensional dynamic parameters are unknown:** As opposed to the classical MOR framework, we do not have access to the dynamic parameters and aim at learning them.
> > > 2. Several MOR approaches consider linear or quadratic reductions mappings [R6, R8], and/or linear configuration manifolds, i.e., with fixed inverse mass-inertia matrix $\mathbf{M}^{-1}=\mathbf{I}$ [R6, R8, R9]. In contrast, our approach **is (1) more general and more expressive in that it can deal with systems that are not linearly reducible and (2) does not assume a fixed inverse mass-inertia matrix,** but learns it as a function of the reduced-order position as $\check{\mathbf{M}}^{-1}(\mathbf{q})$.
> > > 3. Finally, [R6] is close to our work as it considers non-linear reduction mappings learned via an AE. However, the preservation of the symplectic structure is only softly enforced via a loss function, i.e., the symplectic structure is not guaranteed to be preserved. Moreover, [R6] only considers the intrusive scenario. Instead, **our work guarantees the preservation of the symplectic structure of the FOM in the ROM by geometrically constraining the AE**. Moreover, **our paper considers the non-intrusive case**, where the high-dimensional dynamic parameters are unknown.
> > >
> > > > Q4: Can the proposed framework handles irregular time series data?
> > > >
> > >
> > > In our experiments, we used sets of observation collected over intervals with a constant timestep $\Delta t$. However, the framework could handle irregular time series data by employing varying integration time constants and/or time horizons $H$ in the training loss (12).
> > >
> > > > Q5: In line 295, should it be 'constrain' instead of 'constraint'?
> > > >
> > >
> > > Thank you for pointing this typo out! It is fixed in the updated version of the manuscript.

---

> > > > ### Author Response · Authors · 2025-11-21
> > > > **Response to Reviewer cMWE (4)**
> > > >
> > > > > Q6: In line 350, the integrals of $\check{q}$ and $\check{p}$ should solve the IVP, what the initial value in this case?
> > > > >
> > > >
> > > > For every of the $i=1,...,N$ observations over time intervals $\mathcal I_{i} = \left[t_{i}, t_{i}+H\Delta t\right]$, the IVP $\check{\mathbf{q}} _{\text p ,i}(t _{i,j}) = \int _{t _{i}} ^{t _{i,j}} \dot{\check{\mathbf q}} _{\text p ,i} \mathrm dt$
> > > > uses the initial condition $\check{\mathbf q} _{\text p ,i}(t _{i}) := \check{\mathbf q} _{i}$,
> > > > i.e., the reduced observation at the initial timestep $t _{i}$.
> > > >
> > > > Equivalently, the momentum integra $\mathcal I_{i} = \left[t_{i}, t_{i}+H\Delta t\right]$, the IVP $\check{\mathbf{p}} _{\text p ,i}(t _{i,j}) = \int _{t _{i}} ^{t _{i,j}} \dot{\check{\mathbf p}} _{\text p ,i} \mathrm dt$
> > > >  uses the initial condition $\check{\mathbf p} _{\text p ,i}(t _{i}) := \check{\mathbf p} _{i}$. We added this information at the end of Section 3.4 (model training).
> > > >
> > > > > Q7: In line 423, it should be {6,10} instead of 6.10.
> > > > >
> > > >
> > > > We updated the manuscript accordingly.
> > > >
> > > > > Q8: In line 481, is the title of the 5th and 6th column in Table 4 right? Does the $\check{q}$ and $q$ have the same dimension?
> > > > >
> > > >
> > > > This is correct, the dimension of $\check{\mathbf q}$ is lower than the dimension of $\mathbf q$, and $\check{\mathbf{q}} _{\text{p}}$ has the same dimension as $\check{\mathbf{q}}$.
> > > > We fixed the typo in the title of the 5th and 6th columns, which are now $\|\|\check{\mathbf{q}} _{\text{p}} - \check{\mathbf q}\|\| / \|\|\check{\mathbf q}\|\|$ and $\|\|\check{\mathbf{p}} _{\text{p}} - \check{\mathbf p}\|\| / \|\|\check{\mathbf p}\|\|$
> > > >
> > > > > Q9: In your experiments, why do you select the hyperparameter of latent dimension as 6, 10?
> > > > >
> > > >
> > > > In our experiments for the pendulum (Section 4.1 and Appendix H.1) we select our latent space dimension $d=3$, as we expect the latent representation to be mostly driven by the underlying 3-DoF pendulum dynamics, with only little influence from the 12-DoF mesh dynamics.
> > > >
> > > > For the particle vortex and cloth experiments (Sections 4.2 and 4.3, respectively), we empirically selected the latent space dimensions as 6 or 10. These latent dimensions are in line with the literature, showing many high-dimensional systems can be well represented by only few dimensions [R8-10]. For the cloth experiment, we additionally ablate the latent space dimensions ($d={2,6,10}$) in Appendix H.3, showing that with increasing dimension, that prediction errors decrease as the latent dimension increases, suggesting that higher-dimensional latent spaces better capture the original high-dimensional dynamics. However, in general, the prediction errors increase beyond a certain latent dimension, indicating that the latent HNN becomes harder to train. All in all, **the choice of latent dimension must trade off between the expressivity of the latent space and the limitations of HNNs in higher dimensions**.

---

> > > > > ### Author Response · Authors · 2025-11-21
> > > > > **Response to Reviewer cMWE (5/5)**
> > > > >
> > > > > ### References
> > > > >
> > > > > [R1] Lusch, B., Kutz, J. N., & Brunton, S. L. (2018). Deep learning for universal linear embeddings of nonlinear dynamics. Nature communications, 9(1), 4950.
> > > > >
> > > > > [R2] Zhang, J., Zhu, Q., & Lin, W. (2024). Learning Hamiltonian neural koopman operator and simultaneously sustaining and discovering conservation laws. Physical Review Research, 6(1), L012031.
> > > > >
> > > > > [R3] Jin, P., Zhang, Z., Zhu, A., Tang, Y., & Karniadakis, G. E. (2020). SympNets: Intrinsic structure-preserving symplectic networks for identifying Hamiltonian systems. Neural Networks, 132, 166-179.
> > > > >
> > > > > [R4] Zhong, Y. D., Dey, B., & Chakraborty, A. (2019). Symplectic ode-net: Learning hamiltonian dynamics with control. arXiv preprint arXiv:1909.12077.
> > > > >
> > > > > [R5] Liu, Z., Wang, B., Meng, Q., Chen, W., Tegmark, M., & Liu, T. Y. (2021). Machine-learning nonconservative dynamics for new-physics detection. Physical Review E, 104(5), 055302.
> > > > >
> > > > > [R6] Buchfink, P., Glas, S., & Haasdonk, B. (2023). Symplectic model reduction of Hamiltonian systems on nonlinear manifolds and approximation with weakly symplectic autoencoder. SIAM Journal on Scientific Computing, 45(2), A289-A311.
> > > > >
> > > > > [R7] Xiong, W., Huang, X., Zhang, Z., Deng, R., Sun, P., & Tian, Y. (2024). Koopman neural operator as a mesh-free solver of non-linear partial differential equations. Journal of Computational Physics, 513, 113194.
> > > > >
> > > > > [R8] Liqian Peng and Kamran Mohseni. Symplectic model reduction of Hamiltonian systems. SIAM Journal on Scientific Computing, 38(1):A1–A27, 2016.
> > > > >
> > > > > [R9] Harsh Sharma, Hongliang Mu, Patrick Buchfink, Rudy Geelen, Silke Glas, and Boris Kramer. Symplectic model reduction of Hamiltonian systems using data-driven quadratic manifolds. Computer Methods in Applied Mechanics and Engineering, 417:116402, 2023.
> > > > >
> > > > > [R10] Patrick Buchfink, Silke Glas, Bernard Haasdonk, and Benjamin Unger. Model reduction on manifolds: A differential geometric framework. Physica D: Nonlinear Phenomena, 468:134299, 2024.
> > > > >
> > > > > [R11] Samuel Greydanus, Misko Dzamba, and Jason Yosinski. Hamiltonian neural networks. In Neural Information Processing Systems (NeurIPS), volume 32, 2019.
> > > > >
> > > > > [R12] Michael Lutter and Jan Peters. Combining physics and deep learning to learn continuous-time dynamics models. Intl. Journal of Robotics Research, 42(3):83–107, 2023.

---

### Official Review · Reviewer_D54U · 2025-10-31

**Soundness:** 3
**Presentation:** 3
**Contribution:** 3
**Rating:** 4
**Confidence:** 3

**Summary:**

This paper proposes a geometric framework for learning Hamiltonian dynamics via a Reduced-Order Hamiltonian Neural Network (RO-HNN). The method combines a symplectic autoencoder—constructed through cotangent-lift mappings that enforce exact symplecticity and projection consistency—with a latent-space Hamiltonian network that parameterizes the inverse mass, damping, and potential functions using SPD-aware and MLP components. A Strang-type symplectic integrator is employed to evolve the latent dynamics, and the framework is trained end-to-end. Experiments cover systems up to 600 degrees of freedom, including pendula, vortex particles, and damped cloth simulations.

**Strengths:**

The paper introduces a cotangent-lift–based symplectic autoencoder that enforces exact symplecticity and projection consistency through hard constraints rather than soft penalties. This ensures that the learned encoder–decoder pair defines a true symplectic submanifold in the ambient phase space, maintaining the canonical two-form and the projection property.

The inverse mass matrix and damping operator in the latent Hamiltonian are parameterized using SPD-aware neural networks instead of Euclidean Cholesky factors. This geometrically consistent formulation improves numerical stability and preserves positive definiteness during long-time integration.

The method is demonstrated on large-scale systems with 15-, 90-, and 600-degree-of-freedom (DoF) dynamics—including multi-pendulum, vortex particle, and damped cloth systems—showing strong stability, energy preservation, and generalization across regimes.

The paper’s separation of geometry learning (symplectic autoencoder) and dynamics learning (latent-space Hamiltonian) is conceptually clean, interpretable, and aligns with the theoretical principles of symplectic reduction.

**Weaknesses:**

1. The definition in Equation (8) is not clear to me. It is stated that ( M = T^*Q ), but is every symplectic manifold assumed to possess such a cotangent-bundle structure?

Moreover, in the definitions of ( \varphi_Q ) and ( \rho_Q ), the inputs appear to depend only on the configuration variable ( q ). What is the role of the momentum variable ( p ) in these mappings?  Given that the paper’s primary focus is on constructing a learnable, structure-preserving dimensionality-reduction network rather than developing new differential-geometric theory, I would strongly suggest abandoning the abstract manifold language and instead expressing the model directly in coordinate form. A clear coordinate-level presentation of how the encoder and decoder act on (q,p) would make the construction more transparent and easier to verify mathematically and computationally.

2. When dissipation and external forces are introduced, the system ceases to be Hamiltonian. The paper embeds damping and external forces via a learnable SPD operator, but it does not clarify the resulting geometric consistency. In particular, once dissipation is present, the underlying dynamics no longer preserve a symplectic structure. It is therefore unclear what the meaning or advantage of enforcing a symplectic reduction remains in this non-Hamiltonian setting.

If the proposed framework is intended to generalize to dissipative dynamics (e.g., via contact geometry or GENERIC formulations), this connection should be explicitly established. Otherwise, the rationale for retaining a symplectic encoder–decoder becomes questionable. Moreover, the numerical experiments involving dissipative systems do not appear to demonstrate a clear benefit of using a symplectic reduction compared with standard autoencoders or non-geometric models. The authors should clarify whether any theoretical or empirical advantage persists once energy dissipation breaks the canonical symplectic structure.

**Questions:**

pleasse see weakness above

---

> ### Author Response · Authors · 2025-11-21
> **Response to Reviewer D54U (1)**
>
> Many thanks for the insightful feedback! We are glad to read that “the paper’s separation of geometry learning (symplectic autoencoder) and dynamics learning (latent-space Hamiltonian) is conceptually clean, interpretable, and aligns with the theoretical principles of symplectic reduction”!
>
> We address the raised concerns below.
>
> > 1. The definition in Equation (8) is not clear to me. It is stated that $M = T^*Q$, but is every symplectic manifold assumed to possess such a cotangent-bundle structure?
> Moreover, in the definitions of $\varphi_Q$ and $\rho_Q$ , the inputs appear to depend only on the configuration variable ( $q$ ). What is the role of the momentum variable ( $p$ ) in these mappings?
> Given that the paper’s primary focus is on constructing a learnable, structure-preserving dimensionality-reduction network rather than developing new differential-geometric theory, I would strongly suggest abandoning the abstract manifold language and instead expressing the model directly in coordinate form. A clear coordinate-level presentation of how the encoder and decoder act on (q,p) would make the construction more transparent and easier to verify mathematically and computationally.
>
> Thank you for pointing this out! While the embedding $\varphi_{\mathcal{Q}}$ and point reduction $\rho_{\mathcal{Q}}$ depend only on the (reduced) configuration variable, the cotangent-lifted embedding $\varphi$ and point reduction $\rho$ defined in Equation (8) also depend on the momentum through the derivative of the embedding $\varphi_{\mathcal{Q}}$ and point reduction $\rho_{\mathcal{Q}}$. This dependence was not explicitly indicated in Equation (8). We adjusted Equation (8) in the updated version of the manuscript as follows, showing the dependence on $q$ and $p$ explicitly as,
>
> $\varphi(\check{\mathbf{q}}, \check{\mathbf{p}}) = \left(\begin{matrix} \varphi_{\mathcal{Q}} (\check{\mathbf{q}}) \\\\ d\rho_{\mathcal{Q}}\vert_{\varphi_{\mathcal{Q}}(\check{\mathbf{q}})}^T \check{\mathbf{p}} \end{matrix}\right)$ and $\rho(\mathbf{q}, \mathbf{p}) = \left(\begin{matrix}
> \rho_{\mathcal{Q}} (\mathbf{q}) \\\\ d\varphi_{\mathcal{Q}}\vert_{\rho_{\mathcal{Q}}(\mathbf{q})}^T \mathbf{p}
> \end{matrix}\right)$
>
> We intended to write all equations in coordinate-level notations and would be glad to add coordinates if any are still missing. Moreover, we added the following sentence right after (8) in line 221: “the pullbacks $d\rho_{\mathcal{Q}}\vert_{\varphi_{\mathcal Q}(\check{\mathbf{q}})}^\intercal \check{\mathbf{p}}$ and $d\varphi_\mathcal{Q}\vert_{\rho_{\mathcal{Q}}(\mathbf{q})}^\intercal \mathbf{p}$  are computed using the Jacobian of $\rho_{\mathcal{Q}}$ and $\varphi_{\mathcal{Q}}$.” to clarify the computation of the pullbacks.
>
> Concerning the cotangent-bundle structure, not every symplectic manifold possess such a cotangent structure globally. However, every symplectic manifold locally possesses a cotangent bundle structure [1, Sec. 6]. In this paper, we consider Hamiltonian systems $(\mathcal{M}, \mathbf{\omega}, \mathcal{H})$ with globally valid canonical symplectic form $\mathbf{\omega}=\mathbb{J}_{2n}^\intercal$. From [1], this guarantees that the manifold locally has the structure of a cotangent bundle. Moreover, for Legendre-transformable systems, $\mathcal{M}=\mathcal{T}^*\mathcal{Q}$ also holds globally. Note that the converse also holds, i.e., every cotangent bundle possess a canonical symplectic structure. It is important to emphasize that many symplectic manifolds do possess such a structure: This includes most mechanical systems, quantum mechanical systems such as quantum harmonic oscillators, to name a few.

---

> ### Author Response · Authors · 2025-11-21
> **Response to Reviewer D54U (2)**
>
> > 2. When dissipation and external forces are introduced, the system ceases to be Hamiltonian. The paper embeds damping and external forces via a learnable SPD operator, but it does not clarify the resulting geometric consistency. In particular, once dissipation is present, the underlying dynamics no longer preserve a symplectic structure. It is therefore unclear what the meaning or advantage of enforcing a symplectic reduction remains in this non-Hamiltonian setting.
> If the proposed framework is intended to generalize to dissipative dynamics (e.g., via contact geometry or GENERIC formulations), this connection should be explicitly established. Otherwise, the rationale for retaining a symplectic encoder–decoder becomes questionable. Moreover, the numerical experiments involving dissipative systems do not appear to demonstrate a clear benefit of using a symplectic reduction compared with standard autoencoders or non-geometric models. The authors should clarify whether any theoretical or empirical advantage persists once energy dissipation breaks the canonical symplectic structure.
>
> Thank you for this input! It is correct that the system ceases to be Hamiltonian when dissipation and external forces are introduced. However, in our paper, **we consider a specific type of dissipative systems (characterized by a Rayleigh dissipation function), whose vector field structure is preserved by our geometrically-constrained symplectic autoencoder**. We updated Section 3.2 as follows:
>
> -  In lines 306-310, we clarify our assumption as follows:
>
>     “ Specifically, we consider high-dimensional systems with observed external inputs $\mathbf \tau_{\text ext}$ and viscous damping following a Rayleigh dissipative function $\mathcal{D}(\mathbf{q}, \dot{\mathbf{q}}) = \frac{1}{2} \dot{\mathbf{q}}^\intercal \mathbf{D}(\mathbf{q})\dot{\mathbf{q}}$ with unknown positive semi-definite dissipation matrix $\mathbf{D}(\mathbf{q})$. The resulting force field is $\mathbf X_{\mathcal F } = \left(\begin{smallmatrix} \mathbf 0 \\\\ \mathbf \tau_{\text ext} + \mathbf \tau_{\text d} \end{smallmatrix}\right)$ with damping force $\mathbf \tau_{\text d} = \frac{\partial \mathcal{D}(\mathbf{q}, \dot{\mathbf{q}})}{\partial \dot{\mathbf{q}}} = \mathbf{D}(\mathbf{q})\dot{\mathbf{q}}$.”
>
> - We **added the following Proposition 3 and its proof** (lines 311-321) to clarify the preservation of structure of the vector field:
>
>     “**Proposition 3.** The reduced vector field $\check{\mathbf{X}}=\varphi^*\mathbf{X}$ obtained via the pullback of the cotangent-lifted embedding $\varphi$ (8) preserves the structure of the total vector field $\mathbf X = \mathbf X_{\mathcal H} + \mathbf X_{\mathcal F}$ with $\mathbf X_{\mathcal F} = \left(\begin{smallmatrix} \mathbf 0 \\\\ \mathbf \tau_{\text ext} + \mathbf \tau_{\text d } \end{smallmatrix}\right)$.”
>
> - In lines 327-331, we provided an **additional discussion on structure preservation for dissipative dynamics, and on the relationship between the considered dissipative system and more general models**:
>
>     “Note that the dissipative dynamics no longer preserve a symplectic structure. However, Proposition 3 shows that the RO-HNN latent space still preserves the structure of high-dimensional dissipative dynamical systems characterized by a Rayleigh dissipative function. It is worth noting that these systems are equivalent to Port-Hamiltonian systems with energy dissipation matrix $\mathbf{R}(\mathbf{q},\mathbf{p})=\left(\begin{smallmatrix}\mathbf 0 & \mathbf 0 \\\\ \mathbf 0 & \mathbf D(\mathbf q)\end{smallmatrix}\right)$ and input $\mathbf G(\mathbf q ,\mathbf p)\mathbf u = \mathbf \tau_{\text ext}$, and to contact Hamiltonians $\mathcal H_{\text c}(\mathbf q,\mathbf p, s)$ with $\dot s = \mathcal D.$”
>
> In general, **we consider the generalization of the RO-HNN to more general dissipative dynamics as future work**. We highlighted this point in the conclusion as “[In future work] we will also generalize the RO-HNN to learn more general dynamics of Port-Hamiltonian and contact Hamiltonian systems.”
>
> Finally, we would like to highlight that the numerical experiments involving dissipative systems originally presented in Section 3.3) did not compare the proposed geometrically-constrained symplectic AE with a non-symplectic AE. We added an **additional experiment in App. H.3, where we compare our geometrically-constrained symplectic AE against a non-symplectic constrained AE**. Our results presented in Figure 20 show that our geometrically-constrained symplectic AE leads to significantly lower errors than non-symplectic constrained AE, showcasing the benefit of preserving the structure of dissipative vector field in the latent space.
>
> ### References
>
> [1] Alan Weinstein , Symplectic manifolds and their lagrangian submanifolds, Advances in Mathematics, 1971.

---

### Official Review · Reviewer_uBEM · 2025-10-31

**Soundness:** 3
**Presentation:** 3
**Contribution:** 3
**Rating:** 6
**Confidence:** 2

**Summary:**

In this work, the authors propose a new neural network architecture for learning reduced-order Hamiltonian dynamics, consisting mainly of an dimensionality-reducing autoencoder with a specific parameterization that ensures a symplecticity property, and a Hamiltonian NN architecture on the lower-dimensional symplectic manifold. The method is demonstrated to perform well on three physical systems with degrees-of-freedom ranging from 15 to 600.

**Strengths:**

1. The "geometrically-constrained symplectic autoencoder" achieves theoretically-guaranteed symplecticity property through the novel architecture design while demonstrating good expressivity in the experiments, which is a commendable contribution in my opinion.

2. Other components of the RO-HNN model take good advantages of methods from prior literature (e.g. the SPD network for parameterizing the HNN, symplectic integrators, constrained autoencoders)

3. The experiments cover several Hamiltonian systems ranging from simple to higher-dimensional ones and also covering damped systems.

4. The paper is well-presented overall.

**Weaknesses:**

The method assumes that the system has a canonical symplectic form that is valid globally, as well as the existence of a low-dimensional latent Hamiltonian system that captures the high-dimensional dynamics well.

**Questions:**

In the experiments with the coupled pendulum system, is it the case that the learned latent system actually coincide with the 3 degrees of freedom of the underlying pendulum? For the more complex systems (particle vortex and cloth), do the latent dimensions have physical correspondences at all?

---

> ### Author Response · Authors · 2025-11-21
> **Response to Reviewer uBEM**
>
> Thank you very much for the constructive feedback! We are glad that the guaranteed-symplectic reduced space constitutes a “commendable contribution”, that our model “takes good advantage of methods from prior literature”, and that our selection of experiments and results was well-received.
>
> ## Weaknesses
>
> > The method assumes that the system has a canonical symplectic form that is valid globally, as well as the existence of a low-dimensional latent Hamiltonian system that captures the high-dimensional dynamics well.
> >
>
> We agree with the Reviewer that our RO-HNN approach relies on these two assumptions.
>
> However, we would like to highlight that the former (a canonical symplectic form that is valid globally) is a common assumption in HNNs, see e.g., [1-3], which is fulfilled by many real-world systems, e.g., most mechanical systems, quantum mechanical systems such as quantum harmonic oscillators, and systems obtained via a Legendre transform. Moreover, the second assumption (the existence of a surrogate low-dimensional latent Hamiltonian system) is commonly adopted in the structure-preserving MOR literature, see e.g. [4-6].
>
> Despite these assumptions, we believe that our RO-HNN goes beyond the state of the art as a novel approach to learn unknown high-dimensional Hamiltonian dynamics. As stated in our conclusion (lines 535-537), we consider the extension of the RO-HNN to  Hamiltonian systems with non-canonical symplectic form as future work. We believe that this is an interesting research direction, which may be tackled by leveraging Darboux’s theorem to learn local RO-HNNs.
>
> ## Questions
>
> > In the experiments with the coupled pendulum system, is it the case that the learned latent system actually coincide with the 3 degrees of freedom of the underlying pendulum?
> >
>
> This synthetic system was designed primarily to study the performance of our approach on a well-reducible system. Therefore, we expect the latent representation to be mostly driven by the underlying pendulum dynamics, with only little influence from the mesh dynamics. In other words, we expect the RO-HNN to learn latent dynamics that are similar to those of the pendulum, i.e. comparable relative frequencies and amplitudes. However, due to the nonlinear dimensionality reduction conducted via the AE, the latent dynamics are not expected to be exactly equal to the ground truth pendulum dynamics.
>
> To illustrate this, we added a new Figure 9 in App. H.1, which shows that the trajectories of the original 15-DoF pendulum projected into the latent space of the RO-HNN are similar but do not coincide directly with the trajectories of the underlying 3-DoF pendulum representation.
>
> > For the more complex systems (particle vortex and cloth), do the latent dimensions have physical correspondences at all?
> >
>
> In these two cases, there is no direct physical correspondence to the degrees of freedom (DoF) of the original system, and the latent dimensions are selected empirically. Instead our geometrically-constrained symplectic AE extracts the system’s dominant dynamics, which may be shared across several DoF of the original system. As showed for the 600-DoF cloth in Figure 14 in Appendix H.3, a 6-DoF latent space only captures the dominant cloth dynamics, while the 10-DoF latent spaces allows the model to capture more fine-grained dynamics, thus modeling more details within the cloth.
>
> ## References
>
> [1] Michael Lutter and Jan Peters. Combining physics and deep learning to learn continuous-time dynamics models. Intl. Journal of Robotics Research, 42(3):83–107, 2023.
>
> [2] Yaofeng Desmond Zhong, Biswadip Dey, and Amit Chakraborty. Symplectic ODE-net: Learning Hamiltonian dynamics with control. In Intl. Conf. on Learning Representations (ICLR), 2020.
>
> [3] Pengzhan Jin, Zhen Zhang, Aiqing Zhu, Yifa Tang, and George Em Karniadakis. Sympnets: Intrinsic structure-preserving symplectic networks for identifying Hamiltonian systems. Neural Networks, 132:166–179, 2020.
>
> [4] Liqian Peng and Kamran Mohseni. Symplectic model reduction of Hamiltonian systems. SIAM Journal on Scientific Computing, 38(1):A1–A27, 2016.
>
> [5] Harsh Sharma, Hongliang Mu, Patrick Buchfink, Rudy Geelen, Silke Glas, and Boris Kramer. Symplectic model reduction of Hamiltonian systems using data-driven quadratic manifolds. Computer Methods in Applied Mechanics and Engineering, 417:116402, 2023.
>
> [6] Buchfink, P., Glas, S., & Haasdonk, B. (2023). Symplectic model reduction of Hamiltonian systems on nonlinear manifolds and approximation with weakly symplectic autoencoder. SIAM Journal on Scientific Computing, 45(2), A289-A311.

---

### Author Response · Authors · 2025-11-21
**general response by the authors**

First, we thank all reviewers for the valuable feedback regarding our work, and committing their valuable time to the reviews! With this general response, we address the main concerns that were shared among multiple reviewers. Individual questions and comments will be contained in the replies to the specific reviewers. We uploaded a revised manuscript with changes (beyond non-technical typos) highlighted in blue. We hope to reach each individual reviewers’ attention and are looking forward to a valuable discussion!

## Novelty

**Our aim is to learn high-dimensional Hamiltonian dynamics from data.** To do so, our approach **learns the low-dimensional dynamic parameters of a surrogate lower-dimensional Hamiltonian system.**

Our contributions are as follows:

- We propose a geometrically-constrained AE that, by construction, preserves the symplecticity of the high-dimensional manifold onto the latent space;
- We propose a geometric HNN that models conservative dynamics and dissipative dynamics characterized by a Rayleigh dissipation function. Our geometric HNN leverages SPD networks and a Strang-symplectic integrator, leading to more accurate (latent) dynamic modeling;
- We propose a **geometric reduced-order Hamiltonian neural network (RO-HNN)** to learn a low-dimensional symplectic submanifold and the dynamics of the associated Hamiltonian function. We would like to emphasize that the **RO-HNN goes beyond the state of the art as a novel approach to learn unknown high-dimensional Hamiltonian dynamics**, specifically goes beyond (1) previous HNNs that only handle low-dimensional physical systems (considering direct low-dimensional or high-dimensional observations (e.g., images) thereof), and (2) Hamiltonian model-order reduction approaches that are intrusive, i.e., assume to have full knowledge of the full-order system dynamics.

We address the specific questions of the reviewers regarding those points in the individual responses.

## Summary of main changes in the paper

### Clarifications

- In Section 1, we discuss general Poisson systems and Koopman-based approaches;
- We revised Section 3.2 (”dissipative dynamics”) as follows:
    - We precise our assumptions on the type of high-dimensional dissipative systems we consider.
    - We added Proposition 3, which formally shows that the reduced vector field obtained via the pullback of the proposed cotangent-lifted mapping preserves the structure of the high-dimensional dissipative vector field $ \mathbf X = \mathbf X_{\mathcal H} + \mathbf X_{\mathcal F}$
    - We added a discussion on the relationship of the considered dissipative Hamiltonian systems with Port-Hamiltonian and contact Hamiltonian systems.
- In Sec 4.3, we discuss the performance of the Strang-symplectic integrator for dissipative systems.
- In App. D.2, we discuss the computation of the vector-jacobian product of the proposed cotangent lift via analytic derivatives.
- We adjusted Fig. 2 and added Fig.9, 13, and 14 to confirm our results according to metrics that are more robust to error accumulations.

### Additional experiments and comparisons

- In Section 4.1, we compare our model against an additional baseline, the Hamiltonian neural Koopman operator (HNKO) method [1], showing that the HNKO shows significantly higher prediction errors (one order of magnitude higher) than our RO-HNN (see Table 1 and Fig. 2).


    [1] Zhang, J., Zhu, Q., & Lin, W. (2024). Learning hamiltonian neural koopman operator and simultaneously sustaining and discovering conservation laws. Physical Review Research, 6(1), L012031.


- In App. H.1, we investigated the robustness of the RO-HNN to noisy training observation and compared it to the HNKO (see Table 5).
- in App. H.1, added a plot of the latent space learned by the RO-HNN on the pendulum experiment, which we compare with the 3-DoF representation underlying the dynamics of the 15-DoF pendulum (Fig 8).
- In App. H.3, we added an ablation of (1) the geometrically-constrained symplectic AE, and (2) the Strang-symplectic integrator when learning dissipative dynamics (see Fig. 20).
- In App. H.4, we added:
    - A comparison of the measured runtimes of the RO-HNNs and of the numerical ODE solution of the analytic FOM, showing the significant simulation speedup by the RO-HNNs leads to lower evaluation times.
    - The runtimes of different AE, which indicate that wall-clock time for evaluation of the analytically-computed cotangent-lifted autoencoder is significantly lower than twice that of a simple position-level autoencoder.

---

### Author Response · Authors · 2025-11-27
**open to feedback on our rebuttal**

We gladly thank Reviewer fRyk for the engagement in the discussion period and valuable feedback!

We would like to kindly invite Reviewers uBEM, D54U, and cMWE to confirm if our initial response addressed their concerns and would be happy to discuss further or remaining questions.

---

### Meta-Review · Area_Chair_iF5Z · 2026-01-07

**Summary:**

This paper considers the problem of learning a reduced order low-dimensional Hamiltonian system from time trajectories in higher dimensional space. It does so by the introduction of a `symplectic' autoencoder that learns a low-dimensional symplectic submanifold and a structure preserving neural network that models the dynamics on the submanifold. Both reviewers and I agree this is an important problem and this paper proposes some interesting idea. However, there are also major concerns about novelty, technical details (for example, reviewer fRyk raised their rating but in fact still had concerns), practicality of experiments, theoretical guarantee, and missing important relevant references. Unfortunately, these concerns seem to remain largely unresolved after the rebuttal discussion. Recognizing the potential of the idea, I encourage the authors to take the discussions into consideration and re-submit a revised version.

**Reviewer Concerns:**

Just a guess: they probably won't be fully convinced.

**Reviewer Scores:**

Just a guess: they probably won't increase by too much.

---

### Decision · Program_Chairs · 2026-01-26

Reject